# Beyond Additive Decompositions: Interpretability through Separability

**Jinyang Liu** [1]  **Munir Eberhardt Hiabu** [1]

## Abstract

Interpretable machine learning requires models that are accurate and structurally faithful to the data. Existing explainability methods rely heavily on additive representations (e.g., Generalized Additive Models (GAMs), SHapley Additive exPlanations (SHAP), functional ANOVA), which can suffer from signal cancellation and off-support extrapolation in the presence of strong interactions. We propose Tensor Separation Learning (TSL), a regression model that learns a sum of rank-1 products of univariate per-feature functions via a stagewise greedy procedure with orthogonal refitting. By enforcing separability, TSL avoids the information loss inherent in additive projections caused by marginalizing higher-order interactions. The learned TSL model can be fully reconstructed from first-order partial dependence functions, up to constant factors. This stage-wise correspondence ensures that the resulting visualizations are faithful to the fitted components. We establish approximation-rate guarantees for functions with bounded mixed $p$-th order partial derivatives and demonstrate that TSL competes with black-box models on regression benchmarks. [†]

## 1. Introduction

Interpretable machine learning faces a fundamental tension: practitioners need both accurate predictions and stable, readable model structure. Deep neural networks and gradient boosting ensembles achieve state-of-the-art accuracy but operate as "black boxes," making it difficult to understand predictions or assess robustness (Rudin, 2019; Murdoch et al., 2019). Glass-box models like Generalized Additive Models (GAMs) (Hastie & Tibshirani, 1990) provide read-

---
[1]Department of Mathematical Sciences, University of Copenhagen, Denmark. Correspondence to: Jinyang Liu <jl@math.ku.dk>.

*Proceedings of the 43$^{rd}$ International Conference on Machine Learning*, Seoul, South Korea. PMLR 306, 2026. Copyright 2026 by the author(s).

able structure but often miss complex interactions or rely on parametric assumptions that may be misspecified.

The field of interpretable machine learning – that includes glass-box models and post-hoc interpretations – has largely addressed interpretability through *additive decompositions*, where models are expressed as sums of main effects and explicit interaction terms. However, additive decompositions suffer from fundamental limitations in the presence of strong interactions, which we detail in Section 3. Three failure modes recur: (i) local SHAP explanations can cancel across main and interaction effects; (ii) partial dependence plots ignore higher-order interactions; (iii) spurious patterns appear in low-density regions (Hooker, 2007).

We propose *Tensor Separation Learning* (TSL), a glass-box regression model of the form

$$\hat{m}(\mathbf{x}) = \sum_{\ell=1}^{R}\Big(\lambda_+^{(\ell)}\prod_{j=1}^{p}\hat{m}_{+,j}^{(\ell)}(x_j) - \lambda_-^{(\ell)}\prod_{j=1}^{p}\hat{m}_{-,j}^{(\ell)}(x_j)\Big),$$

where $R \geq 1$ is the number of TSL stages and each stage $\ell$ is the difference of two positive separable products, and every component function $\hat{m}_{\pm,j}^{(\ell)} : \mathbb{R} \rightarrow \mathbb{R}_{>0}$ is a univariate positive function of a single feature. By enforcing positive components $\hat{m}_{\pm,j}^{(\ell)} > 0$, each component unambiguously amplifies or suppresses its product. TSL fits the model greedily. At each stage $\ell$, the products $(\prod_{j=1}^{p}\hat{m}_{+,j}^{(\ell)}(x_j), \prod_{j=1}^{p}\hat{m}_{-,j}^{(\ell)}(x_j))$ are fit against the current residuals, then all current and previous stage coefficients $\{\lambda_+^{(k)}, \lambda_-^{(k)}\}_{k=1}^{\ell}$ are jointly optimized via least squares. Because each stage is a difference of two products of 1D functions, marginalizing over the other features rescales each side by a constant, so the partial dependence functions are directly proportional to the fitted components.

Our contributions are as follows:

- **Tensor Separation Learning (TSL)**: A regression algorithm that estimates the conditional mean as a sum of differences of positive rank-1 separable products.

- **Model-Native Interpretability**: Each separable stage is recoverable from its branch-wise 1D partial dependence curves, up to one scalar normalizer per stage, so the plots faithfully represent fitted components rather

than post-hoc surrogates. We further introduce a back-bone/tilt representation that separates stage activity from signed branch imbalance. Exemplary visualizations are found at: Sections I.1 and I.2

- **Approximation Theory**: We establish $O(1/\sqrt{r})$ approximation rates for the TSL hypothesis class on functions with bounded mixed $p$-th order partial derivatives, via the orthogonal greedy algorithm (OGA) framework of Barron et al. (2008).

**Conflict of Interest Disclosure.** The authors declare no financial conflicts of interest.

## 2. Related Work

Interpretability methods in machine learning can be broadly categorized into two approaches: *post-hoc explanations* that analyze black-box models after training, and *glass-box models* that are interpretable by design (Murdoch et al., 2019; Rudin, 2019). Despite this distinction, both categories share a fundamental characteristic: they rely on *additive decompositions* to provide interpretable views of the predictor.

### 2.1. Additive Post-Hoc Explanations and Glass-Box Models

Post-hoc explanation methods apply additive decompositions to fixed black-box predictors. Examples include SHAP (Lundberg & Lee, 2017), which decomposes a prediction into feature contributions at a point; Integrated Gradients (Sundararajan et al., 2017), whose Completeness axiom forces an additive decomposition of the prediction across features; and the Functional ANOVA decomposition (Hooker, 2007), which provides a global additive view.

Glass-box models enforce interpretability by constraining the predictor itself to an additive structure. Generalized Additive Models (GAMs) represent the predictor as a sum of univariate "shape functions" (Hastie & Tibshirani, 1990), achieving transparency by restricting components to 1D terms. GAMs can be extended to include pairwise interaction surfaces (GA²M) that remain visualizable in one or two dimensions (Lou et al., 2013), decomposing the predictive model as $m(\mathbf{x}) = m_0 + \sum_j m_j(x_j) + \sum_{i<j} m_{ij}(x_i, x_j)$. Random Planted Forests (Hiabu et al., 2020) generalizes this idea by allowing interaction components up to a pre-specified order, each represented by a regression tree and fitted in parallel. Neural Additive Models (NAMs) replace the parametric 1D shape functions with flexible neural components (Agarwal et al., 2021); Neural Basis Models share a low-rank shape-function basis across features (Radenovic et al., 2022); NODE-GAM instantiates GAM and GA²M variants via differentiable oblivious decision trees (Chang et al., 2022); and models such as GAMI-Net explicitly in-

corporate structured interaction terms under heredity-type constraints (Yang et al., 2021).

Across both perspectives, the central challenge is not only to obtain an additive representation that is interpretable, but to do so without sacrificing predictive performance. The tension is two-sided: black-box models may achieve high predictive accuracy because they are not constrained to additively decompose into low-dimensional components. This flexibility renders faithful post-hoc explanation difficult (see Section 3), whereas glass-box models are interpretable by construction but may sacrifice accuracy due to their structural restrictions.

### 2.2. Tensor Decomposition Methods

A closely related line of work is classical tensor decomposition. In CP/PARAFAC, one models an observed $p$-way tensor $\mathcal{X} \in \mathbb{R}^{I_1 \times \cdots \times I_p}$, where $I_k \in \{1, 2, \dots\}$, by a sum of outer-products $\mathcal{X} \approx \sum_{\ell=1}^{r} \lambda_\ell \, a_\ell^{(1)} \circ \cdots \circ a_\ell^{(p)}$ with factor vectors $a_\ell^{(k)} \in \mathbb{R}^{I_k}$ and weights $\lambda_\ell \in \mathbb{R}$ (Harshman, 1970; Carroll & Chang, 1970; Kolda & Bader, 2009; Sidiropoulos et al., 2017). Equivalently, in entrywise form, $X_{i_1,\dots,i_p} \approx \sum_{\ell=1}^{r} \lambda_\ell \prod_{k=1}^{p} a_{i_k\ell}^{(k)}$, which is an additive superposition of separable rank-1 components. The key difference is the inferential objective: classical tensor decomposition typically seeks recovery of ground-truth latent factors (up to the usual scaling/permutation ambiguities), whereas our method uses separable structure to learn an interpretable supervised predictor from scattered covariate–response samples.

### 2.3. Separation Learning

We define the class of *separation models* as functions that can be represented as a sum of separable rank-1 products:

$$m(\mathbf{x}) = \sum_{\ell=1}^{r} s_\ell \prod_{j=1}^{p} g_j^{(\ell)}(x_j), \quad s_\ell \in \mathbb{R}, \ g_j^{(\ell)} : \mathbb{R} \to \mathbb{R}. \tag{1}$$

Although (1) algebraically resembles a tensor decomposition discussed in the previous section, the goal here is different. Separation learning considers the problem of supervised prediction with a separable model (see Section 4 for setup). The functional form was first introduced by Beylkin & Mohlenkamp (2005) to approximate a *fixed* high-dimensional function. Adapting this form to the supervised setting, previous work on separation learning reduces estimation to optimizing the coefficients of a fixed univariate basis, differing mainly in the choice of basis and optimizer: (Beylkin et al., 2009; Garcke, 2010; Chevreuil et al., 2015), spline bases under Gauss–Newton updates (Govindarajan et al., 2022), and a truncated Fourier basis fit via tensor decomposition of the coefficient tensor (Kargas & Sidiropoulos, 2021). In a related functional-ANOVA setting, ANOVA-

TPNN (Park et al., 2025) models the ANOVA components as tensor products of learnable centered neural bases, fitting main and interaction effects jointly under sum-to-zero identifiability constraints. TSL differs from these approaches in two significant ways. (a) Estimation is done stagewise, by learning on the residuals of the previous stage rather than solving a fixed-rank joint optimization problem over a predetermined basis. (b) Instead of fitting unconstrained rank-1 tensors, we fit a difference of two positive rank-1 tensors at every stage aiding both interpretability and stability of the fit (discussed in Sections 4.1 and 4.2.3).

### 2.3.1. OTHER RELATED MODELS

Separation learning is distinct from tensor regression with tensor-valued covariates (Zhou et al., 2013; Lock, 2018), tensor-network classifiers such as tensor trains (Stoudenmire & Schwab, 2016), and Kolmogorov–Arnold networks (Liu et al., 2025b).

### 2.4. Greedy Approximation and Boosting Lineage

Forward stagewise procedures (including boosting-style fitting) iteratively add weak learners that reduce residual error, yielding an additive estimator with controlled complexity via early stopping (Hastie et al., 2009). Rule ensembles (Friedman & Popescu, 2008) and modern gradient-boosted trees such as XGBoost (Chen & Guestrin, 2016) and LightGBM (Ke et al., 2017) are strong baselines, but their accuracy comes from aggregating many weak learners, yielding a sum of hundreds of terms that is effectively opaque. TSL shares the stagewise structure but differs in that it is a low-rank sum of interpretable strong learners.

## 3. Limitations of Post-Hoc Additive Decompositions

Post-hoc additive decompositions face two well-known limitations when strong interactions are present. We now discuss two popular methods of additive explanation — SHAP and partial dependence plots — and show how each can obscure the true effect of individual features.

**Shapley cancellation.** SHAP (Lundberg & Lee, 2017) assigns each feature $k$ a single local attribution $\phi_k(\mathbf{x})$. There exists a functional decomposition $m(\mathbf{x}) = \sum_{S \subseteq [p]} m_S(\mathbf{x}_S)$, where $\mathbf{x}_S := (x_j)_{j \in S}$ is the subvector indexed by $S$, $m_S$ is the corresponding ANOVA component, and $m_\emptyset$ is a constant intercept, such that the Shapley value distributes each functional component equally among the features it contains (Rota, 1964; Harsanyi, 1963; Hiabu et al., 2023; Bordt & von Luxburg, 2023): $\phi_k(\mathbf{x}) = m_k(x_k) + \frac{1}{2} \sum_{j \neq k} m_{kj}(x_j, x_k) + \cdots$ Thus SHAP reports only the net contribution of feature $k$: main and interaction effects with opposite signs can cancel, so $\phi_k(\mathbf{x}) = 0$ does

not imply that feature $k$ is irrelevant or inactive at $\mathbf{x}$.

**Partial dependence discards interactions.** The partial dependence function of feature $j$ is defined as the marginal expectation $\mathrm{PD}_j(x_j) := \mathbb{E}_{X_{(-j)}}[m(x_j, X_{(-j)})]$, where $X_{(-j)}$ denotes the random subvector over all coordinates except $j$; it recovers only the main effect of feature $j$ and is blind to the higher-order interactions in which the feature participates. Under the functional decomposition $m(\mathbf{x}) = \sum_{S \subseteq [p]} m_S(\mathbf{x}_S)$, identified with the marginal constraint (Hiabu et al., 2023)

$$\sum_{\substack{T \subseteq [p]: \\ T \cap S \neq \emptyset}} \mathbb{E}[m_T(\mathbf{x}_{T \setminus S}, X_{T \cap S})] = 0,$$

the partial dependence function of $x_1$ equals the intercept plus its main effect, $\mathrm{PD}_1(x_1) = m_\emptyset + m_1(x_1)$, and therefore carries no information about any interaction term $m_S$ with $1 \in S$ and $|S| \geq 2$. Furthermore, when features are correlated, marginal averaging can suffer from extrapolation in low-density regions (Liu et al., 2025a; Aas et al., 2021). We demonstrate the interaction-masking failure mode in a synthetic example (Section 7.2).

**Partial dependence faithfulness for separable models.** Separable models do not suffer from the aforementioned limitations. For a separable product, the interaction structure lives in the product form itself, $h(\mathbf{x}) = \prod_{j=1}^{p} h_j(x_j)$. Partial dependence functions factorize as $\mathrm{PD}_j[h](x_j) = c_j h_j(x_j)$, with $c_j = \mathbb{E}[\prod_{k \neq j} h_k(X_k)]$ constant in $x_j$, so $\mathrm{PD}_j$ recovers the factor shape exactly rather than collapsing to a main effect.

## 4. Method

We define the TSL estimator in Section 4.1 and its stagewise fitting procedure in Section 4.2, including grid-tensor refinement, bagged aggregation, and identifiability. We consider the supervised regression problem with features $\mathbf{x} \in \mathcal{X} \subseteq \mathbb{R}^p$, with $\mathcal{X}$ possibly non-rectangular, and targets $y \in \mathbb{R}$. Throughout, for a positive integer $k$, we write $[k] := \{1, \ldots, k\}$; upper-case $X_j$ denotes the random variable for coordinate $j$ and lower-case $x_j$ its realized value, with $X_{(-j)}$ the random subvector over all coordinates except $j$. We aim to estimate the conditional mean $m(\mathbf{x}) = \mathbb{E}[Y|X = \mathbf{x}]$ from i.i.d. observations $\mathcal{D}_n = \{(y^{(i)}, \mathbf{x}^{(i)})\}_{i=1}^{n} \subseteq \mathbb{R} \times \mathcal{X}$.

### 4.1. Tensor Separation Learning

We introduce the *Tensor Separation Learning* (TSL) estimator $\hat{m}$ that learns $R$ stages for learning $m$ from data $\mathcal{D}_n$:

$$\hat{m}(\mathbf{x}) = \sum_{\ell=1}^{R} \left( \lambda_+^{(\ell)} \prod_{j=1}^{p} \hat{m}_{+,j}^{(\ell)}(x_j) - \lambda_-^{(\ell)} \prod_{j=1}^{p} \hat{m}_{-,j}^{(\ell)}(x_j) \right),$$

$$(2)$$

subject to positive univariate components $\hat{m}_{+,j}^{(\ell)}(x_j) > 0$, $\hat{m}_{-,j}^{(\ell)}(x_j) > 0$ for all $j \in [p]$. We define the signed products $\hat{m}_{\pm}^{(\ell)}(\mathbf{x}) := \lambda_{\pm}^{(\ell)} \prod_{j=1}^{p} \hat{m}_{\pm,j}^{(\ell)}(x_j)$ and stage predictor $\hat{m}^{(\ell)}(\mathbf{x}) := \hat{m}_{+}^{(\ell)}(\mathbf{x}) - \hat{m}_{-}^{(\ell)}(\mathbf{x})$ for brevity.

Beylkin et al. (2009) defined the *separation rank* as the number of separable products in the model; since each stage in our model contributes at most two separable products if no product is zero, the total separation rank is at most $2R$. A low separation rank aids interpretability: fewer products are needed to represent the model.

TSL differs from the separable model (1) considered by others in one key respect: it constrains all components to be positive. The design is motivated by two considerations. First, positivity promotes stability during fitting (discussed in Section 4.2.3). Second, it improves interpretability. In an unconstrained product $\prod_j h_j(x_j)$, $h_j \in \mathbb{R}$, the sign of the entire product is determined by the product of all component signs — so a positive $x_2$ component does not imply a positive contribution from the product if the component for $x_1$ can be negative. Consider a pair of features $(x_1, x_2)$ that is predominantly observed on a set $D \subset \mathbb{R}^2$: if $h_1 < 0$ and $h_2 > 0$ on $D$, the product's contribution on $D$ is negative, despite $h_2$ being positive. Outside $D$, where $h_1$ may be positive, the same product contributes positively. Positivity removes this ambiguity: when all components are positive, each component unambiguously amplifies its product's contribution, and the sign of a component faithfully reflects its directional role.

### 4.1.1. BACKBONE AND EXPONENTIAL-TILT PARAMETRIZATION

Each TSL stage is the difference of two positive separable products, and we observe empirically that the component pair $(\hat{m}_{+,j}^{(\ell)}, \hat{m}_{-,j}^{(\ell)})$ on a given feature is often close to balanced ($\hat{m}_{+,j}^{(\ell)} = \hat{m}_j^{(\ell)} = \hat{m}_{-,j}^{(\ell)}$) for some features but strongly imbalanced for others. When feature $x_j$ is balanced, it factors out of the difference as a common multiplier $\hat{m}_j^{(\ell)} \cdot (\prod_{k \neq j} \hat{m}_{+,k}^{(\ell)} - \prod_{k \neq j} \hat{m}_{-,k}^{(\ell)})$ and acts purely as a scale knob; when imbalanced, it tilts the stage toward the $(+)$ or $(-)$ product. To make these two roles explicit, we reparameterize each pair via a positive *backbone* $b_j^{(\ell)}(x_j) > 0$ encoding the shared magnitude and a *tilt* $d_j^{(\ell)}(x_j) \in \mathbb{R}$ encoding the signed imbalance:

$$\hat{m}_{\pm,j}^{(\ell)}(x_j) = b_j^{(\ell)}(x_j) e^{\pm d_j^{(\ell)}(x_j)}, \qquad (3)$$

The map (3) is a bijection with inverse $b_j^{(\ell)} = \sqrt{\hat{m}_{+,j}^{(\ell)} \hat{m}_{-,j}^{(\ell)}}$ and $d_j^{(\ell)} = \frac{1}{2} \log(\hat{m}_{+,j}^{(\ell)} / \hat{m}_{-,j}^{(\ell)})$ (see Section F for proof). The same parametrization absorbs the stage scalars $\lambda_{\pm}^{(\ell)}$ into a stage-level backbone scale, $b_0^{(\ell)} = \sqrt{\lambda_+^{(\ell)} \lambda_-^{(\ell)}}$, and tilt

intercept, $d_0^{(\ell)} = \frac{1}{2} \log(\lambda_+^{(\ell)} / \lambda_-^{(\ell)})$, so that the per-feature backbones and tilts aggregate cleanly at the stage level as $b^{(\ell)}(\mathbf{x}) = b_0^{(\ell)} \prod_{j=1}^{p} b_j^{(\ell)}(x_j)$ and $d^{(\ell)}(\mathbf{x}) = d_0^{(\ell)} + \sum_{j=1}^{p} d_j^{(\ell)}(x_j)$. Substituting (3) into the stage predictor $\hat{m}^{(\ell)}$ yields

$$\hat{m}(\mathbf{x}) = 2 \sum_{\ell=1}^{R} b^{(\ell)}(\mathbf{x}) \sinh(d^{(\ell)}(\mathbf{x})). \qquad (4)$$

Equation (4) cleanly separates stage activity from signed imbalance. The backbone $b^{(\ell)}(\mathbf{x}) > 0$ is an activity gate: $b_0^{(\ell)}$ sets the stage's overall scale, and the per-feature backbones $b_j^{(\ell)}(x_j)$ multiply to determine where the stage is active—a near-zero factor in any feature switches the stage off in that region. The tilt $d^{(\ell)}(\mathbf{x}) \in \mathbb{R}$ controls the signed direction through the strictly increasing odd function $\sinh$: $d_0^{(\ell)}$ is the baseline $(+)/(-)$ imbalance before features contribute, and each $d_j^{(\ell)}(x_j)$ adds onto this intercept. Thus, holding the backbone and the other tilts fixed, increasing $d_j^{(\ell)}(x_j)$ always increases the stage contribution; positive values tilt the stage toward the $(+)$ product, negative values tilt it toward the $(-)$ product, and zero corresponds to local branch balance. In this sense, $d_j^{(\ell)}$ has a clean interpretation as an additive imbalance score, with $\sinh$ only providing a monotone response-scale transformation of the total tilt.

### 4.1.2. NORMALIZATION

The backbone–tilt parametrization in (4) is invariant to feature-wise rescalings of $b_j^{(\ell)}$ and constant shifts of $d_j^{(\ell)}$. We use the normalized representative satisfying

$$\mathbb{E}\left[\log b_j^{(\ell)}(X_j)\right] = 0, \quad \mathbb{E}\left[d_j^{(\ell)}(X_j)\right] = 0, \qquad j \in [p].$$

Equivalently, since $\hat{m}_{\pm,j}^{(\ell)} = b_j^{(\ell)} \exp(\pm d_j^{(\ell)})$, this is the same as imposing

$$\mathbb{E}\left[\log \hat{m}_{+,j}^{(\ell)}(X_j)\right] = \mathbb{E}\left[\log \hat{m}_{-,j}^{(\ell)}(X_j)\right] = 0.$$

Thus each univariate branch factor has geometric mean one, while the branch-level scales are carried by $\lambda_+^{(\ell)}$ and $\lambda_-^{(\ell)}$. The empirical version of this normalization is used before averaging bagged grids in Section D.3. This fixes a within-stage representative, but does not remove the support-induced non-identifiability discussed in Section 4.2.3.

## 4.2. Fitting

The TSL fitting procedure combines forward stagewise additive modeling with boosting-style residual fitting and CART-style partition refinement (Breiman et al., 1984; Friedman, 2001; Hastie et al., 2009); pseudocode is in Section C and

the end-to-end pipeline is illustrated in the flowchart in Figure 4. At stage $\ell$, the previously selected stages are held fixed and a new stage predictor $\hat{m}^{(\ell)}$ is fit to the residuals $R_i^{(\ell-1)} := y_i - \sum_{k=1}^{\ell-1} \hat{m}^{(k)}(\mathbf{x}^{(i)})$ (empty sum at $\ell = 1$, so the first stage is fit directly to $y_i$) via grid-tensor refinement and bagged aggregation. After fitting stage $\ell$, we *refit* all stage scalars $\{\lambda_\pm^{(k)}\}_{k=1}^{\ell}$ up to the current stage jointly by least squares against $y$.

### 4.2.1. GRID TENSOR REFINEMENT

We assume for now that $\lambda_\pm^{(\ell)} = 1$; the stage scalars are initialized and re-fit in the outer loop (see Section C). The current stage is fit by learning the positive $(+)$ and negative $(-)$ separable products on a shared adaptive partition. Formally, we define a *grid tensor* as the tuple $\mathcal{G} = \left( \prod_{j=1}^p \mathcal{I}_j, \{\hat{v}_{\pm,j,I}\}_{I \in \mathcal{I}_j, j \in [p]} \right)$, where $\mathcal{I}_j = \{I_{j,1}, \ldots, I_{j,L_j}\}$ partitions the domain of $x_j$ and $\hat{v}_{\pm,j,I} > 0$ are the $(+)/(-)$ values attained on interval $I$. We then model each separable term as a product of component functions $\hat{m}_{\pm,j}(x_j)$ that are piecewise-constant on $\mathcal{I}_j$, namely $\hat{m}_{\pm,j}(x_j) = \sum_{I \in \mathcal{I}_j} \hat{v}_{\pm,j,I} \mathbb{1}_I(x_j)$ so that each $(\pm)$ product evaluates to the corresponding cell-value of its rank-1 tensor. The grid tensor is iteratively learned by refining the intervals $\mathcal{I}_j$ on each feature axis $j$ via a greedy split-scoring procedure, inspired by CART-style decision trees (Breiman et al., 1984). At each iteration, we evaluate candidate splits on feature axis $j$ at threshold $s$ by partitioning an existing interval $I = [a, b) \in \mathcal{I}_j$ into left and right subintervals $I_L = [a, s)$ and $I_R = [s, b)$, then solve the following regularized least-squares objective for each region $S \in \{L, R\}$:

$$\mathcal{L}_S(u_+^S, u_-^S) = \sum_{x_j^{(i)} \in I_S} w_i (R_i^{(\ell-1)} - (u_+^S \hat{m}_+^{(i)} - u_-^S \hat{m}_-^{(i)}))^2$$
$$+ \alpha((u_+^S - 1)^2 + (u_-^S - 1)^2), \tag{5}$$

where $\hat{m}_\pm^{(i)} := \hat{m}_\pm^{(\ell)}(\mathbf{x}^{(i)})$ represents the predictions of the current $(+)$ and $(-)$ products on $\mathbf{x}^{(i)}$ as defined in (2), $w_i \geq 0$ are stabilizing weights and $\alpha \geq 0$ is ridge regularization. To see how (5) arises, observe that for the interval $I = [a, b) \in \mathcal{I}_j$ that is being refined, the predictions of the current $(+)$ and $(-)$ products on the samples in $I$ are given by $\hat{m}_\pm^{(i)} = \lambda_\pm^{(\ell)} \hat{v}_{\pm,j,I} \prod_{k \neq j} \hat{m}_{\pm,k}^{(\ell)}(x_k^{(i)})$ for $i$ with $x_j^{(i)} \in I$, where $\hat{v}_{\pm,j,I}$ is the value attained by $\hat{m}_{\pm,j}^{(\ell)}$ on $I$. Upon splitting $I$ into $I_L = [a, s)$ and $I_R = [s, b)$, the prediction made by the other components $\prod_{k \neq j} \hat{m}_{\pm,k}^{(i)}$ remains unchanged. In order to refine the predictions on $I_L$ and $I_R$, the updated values are parametrized as $\hat{v}_{\pm,j,I_S} = \hat{v}_{\pm,j,I} u_\pm^S$ for $S \in \{L, R\}$, and the multiplicative updates $u_\pm^S$ are optimized. The post-split prediction for any sample $i$ with $x_j^{(i)} \in I_S$ then becomes $u_\pm^S \hat{m}_\pm^{(i)}$, thus motivating the objective (5).

The minimizer of (5) is obtained from a closed-form $2 \times 2$ linear system of sufficient statistics (see Section B.1). The split that maximizes the total reduction $\Delta_{\text{split}} = \Delta_L + \Delta_R$ where $\Delta_S := \mathcal{L}_S(1, 1) - \mathcal{L}_S(u_+^S, u_-^S)$ for $S \in \{L, R\}$ is selected.

### 4.2.2. BAGGING AND AGGREGATION

Borrowing from bagging and Random Forests (Breiman, 1996; 2001), we reduce variance by fitting $n_{\text{grids}}$ grid tensors $\mathcal{G}^{(\ell,1)}, \ldots, \mathcal{G}^{(\ell,n_{\text{grids}})}$ on bootstrap samples in parallel; let $\hat{m}_\pm^{(\ell,c)}$ for $c = 1, \ldots, n_{\text{grids}}$ denote the resulting stage predictors. An arithmetic mean of products is not a product, so we aggregate the per-feature components $\hat{m}_{\pm,j}^{(\ell,c)}$ rather than the stage predictors. Factor-wise averaging is fragile because separable representations are non-identifiable (Section 4.2.3), so independently fitted factors need not lie in a comparable gauge.

To aggregate the bagged grids we work in the backbone/tilt-parametrization. Our similarity-based aggregation scheme proceeds in four steps. *Normalize:* center each $\log b_j^{(\ell,c)}$ and $d_j^{(\ell,c)}$ on every axis and absorb the removed constants into $\lambda_\pm^{(\ell)}$. *Anchor:* each candidate $c$ contributes a point $(\lambda_+^{(\ell,c)}, \lambda_-^{(\ell,c)})$ in the $(\lambda_+, \lambda_-)$ plane; we choose as reference $\mathcal{G}^\star$ the candidate closest to their centroid,

$$\mathcal{G}^\star = \underset{\mathcal{G}^{(\ell,c)}}{\arg\min} \sum_{c'=1}^{n_{\text{grids}}} \left[ (\lambda_+^{(c)} - \lambda_+^{(c')})^2 + (\lambda_-^{(c)} - \lambda_-^{(c')})^2 \right].$$

*Filter:* discard candidates whose normalized components are too dissimilar to the reference under cosine similarity, controlled by a tuned threshold $\xi$. *Average:* combine the survivors by taking the average of $\log b_j^{(\ell,c)}$ and tilts $d_j^{(\ell,c)}$ for every $j$. Figure 14 illustrates an example where bagged grids converge to two distinct backbone representations of the same fitted stage, showing how averaging without filtering can cause identifiability issues as discussed in the next section.

The overall training cost for TSL is $\mathcal{O}(R\,n_{\text{grids}}\,T n p)$ for $R$ stages, $n_{\text{grids}}$ bagged grids per stage, $T$ split iterations per grid, $n$ samples, and $p$ features. The bagged grids are embarrassingly parallel, and histogram binning can reduce the per-iteration dependence from $n$ to the number of histogram bins. Full details are given in Sections A to C and D.1.

### 4.2.3. IDENTIFIABILITY AND STABILITY

Separable models are not uniquely identified. There are three sources of ambiguity which we discuss below.

**Classical scaling and permutation ambiguities.** In classical tensor decomposition, a fully observed multi-way array is identifiable up to permutation and scaling ambiguities

only (Kruskal, 1977; Kolda & Bader, 2009; Sidiropoulos et al., 2017). TSL inherits the same ambiguities but can be remedied via normalization (Section 4.1.2).

**Non-identifiability from non-rectangular support.** In supervised learning, the feature-support $\mathcal{X}$ may not be rectangular and can introduce nontrivial ambiguities beyond global scaling and sign. For a two-dimensional example, let $\mathcal{X} = A \cup B$, where $A = [-1, 0] \times [-1, 0]$ and $B = [0, 1] \times [0, 1]$, and define

$$m(x_1, x_2) = \mathbb{1}_A(x_1, x_2)a_1(x_1)a_2(x_2)$$
$$+ \mathbb{1}_B(x_1, x_2)b_1(x_1)b_2(x_2),$$

for arbitrary functions $a_1, a_2, b_1, b_2$. The following rank-1 factorizations all agree with $m$ on the observed support $\mathcal{X}$:

$$m^{(1)} = \big(\mathbb{1}_{[-1,0]}a_1 + \mathbb{1}_{[0,1]}b_1\big)\big(\mathbb{1}_{[-1,0]}a_2 + \mathbb{1}_{[0,1]}b_2\big),$$
$$m^{(2)} = \big(-\mathbb{1}_{[-1,0]}a_1 + \mathbb{1}_{[0,1]}b_1\big)\big(-\mathbb{1}_{[-1,0]}a_2 + \mathbb{1}_{[0,1]}b_2\big),$$
$$m^{(3)} = \big(-\mathbb{1}_{[-1,0]}a_1 - \mathbb{1}_{[0,1]}b_1\big)\big(-\mathbb{1}_{[-1,0]}a_2 - \mathbb{1}_{[0,1]}b_2\big).$$

When fitting $n_{\text{grids}}$ in parallel, the fitted grid tensors may adopt different representations such as above, that all estimate $m$ equally well. By imposing positivity, TSL stabilizes the aggregation of independently fitted components, such that averaging the components will not result in cancellation, which may happen in the $A \cup B$ example above if the $x_1$ component of $m^{(1)}$ was averaged with the $x_1$ component of $m^{(3)}$. Regardless, positivity does not fully remove non-identifiability. Even after the signs are fixed, each positive component can still have region-dependent scalings. For example, on region $A$ above, replacing $a_1$ with $a_1/c$ and $a_2$ with $ca_2$ for $c > 0$ leaves the prediction unchanged on $A$ and therefore on all of $\mathcal{X}$. This is a consequence of the non-rectangular support: were the model evaluated on the full rectangular span $[-1, 1]^2$, the rescaling would alter the predictions on $[-1, 1]^2 \setminus \mathcal{X}$, making it an inadmissible reparameterization and resolving the ambiguity.

**Non-identifiability from noisy observations.** Finally, in the supervised setting, the fitting criterion evaluates only predictive error against noisy responses, $y^{(i)}$, so distinct separable representations can attain essentially the same error. Consequently, the latent factors should not in general be interpreted as uniquely recoverable objects, but rather one of many admissible representations selected by the stochastic fitting procedure. Figure 14 in Section I.4 shows a concrete instance, where bagged grid tensors converge to two distinct backbone representations of the same fitted stage, populating opposite ends of the $(\lambda_+, \lambda_-)$ spectrum. The align-then-filter procedure of Section 4.2.2 resolves the ambiguity by selecting one canonical representative for averaging.

## 5. Model-Native Partial Dependence

The key advantage of TSL's separable structure is that each stage is reconstructed—up to per-stage, per-feature, per-branch scalar constants $C_{\pm,j}^{(\ell)}$—from its one-dimensional partial dependence functions. Global interpretability diagnostics computed from partial dependence are therefore exact representations of the fitted factor *shapes*, not post-hoc approximations.

**Proposition 5.1** (Partial Dependence Decomposition). *Consider a TSL estimator $\hat{m}$ as defined in (2). For any stage $\ell$ and coordinate $j$, define $c_{\pm,j}^{(\ell)} := \mathbb{E}\Big[\prod_{k \neq j} \hat{m}_{\pm,k}^{(\ell)}(X_k)\Big]$ and $C_{\pm,j}^{(\ell)} := c_{\pm,j}^{(\ell)} \lambda_{\pm}^{(\ell)}$. Then the 1D partial dependence function of $\hat{m}_{\pm}^{(\ell)}(\mathbf{x}) = \lambda_{\pm}^{(\ell)} \prod_{j=1}^{p} \hat{m}_{\pm,j}^{(\ell)}(x_j)$ on coordinate $j$ satisfies*

$$\text{PD}_{\pm,j}^{(\ell)}(x_j) := \mathbb{E}\Big[\hat{m}_{\pm}^{(\ell)}(x_j, X_{(-j)})\Big] = C_{\pm,j}^{(\ell)} \hat{m}_{\pm,j}^{(\ell)}(x_j),$$
$$(6)$$

*i.e., each 1D partial dependence curve is the 1D factor shape times the constant $C_{\pm,j}^{(\ell)}$. Moreover, letting $\bar{m}_{\pm}^{(\ell)} := \mathbb{E}\Big[\hat{m}_{\pm}^{(\ell)}(X)\Big], Z_{\pm}^{(\ell)} := \mathbb{E}\Big[\prod_{j=1}^{p} \text{PD}_{\pm,j}^{(\ell)}(X_j)\Big]$, the stage admits the following exact reconstruction*

$$\hat{m}_{\pm}^{(\ell)}(\mathbf{x}) = \frac{\bar{m}_{\pm}^{(\ell)}}{Z_{\pm}^{(\ell)}} \prod_{j=1}^{p} \text{PD}_{\pm,j}^{(\ell)}(x_j). \qquad (7)$$

*Consequently, each stage (and thus any downstream explanation primitive built from the stage factors) is computable from 1D partial dependence summaries up to a single scalar normalizer per stage and sign branch.*

The proof is in Section E. TSL's 1D partial dependence plots therefore recover the fitted factor shapes up to the per-stage scalars $C_{\pm,j}^{(\ell)}$, without a post-hoc surrogate.

An analogous reconstruction theorem recovers the backbone/tilt parametrization of Equation (3) directly from the 1D partial dependence curves: $b_j^{(\ell)}(x_j) = (C_{+,j}^{(\ell)} C_{-,j}^{(\ell)})^{-1/2} \sqrt{\text{PD}_{+,j}^{(\ell)}(x_j) \text{PD}_{-,j}^{(\ell)}(x_j)}$ and $d_j^{(\ell)}(x_j) = \frac{1}{2} \log(\text{PD}_{+,j}^{(\ell)}(x_j)/\text{PD}_{-,j}^{(\ell)}(x_j)) + \gamma_j^{(\ell)}$ with $\gamma_j^{(\ell)} = \frac{1}{2} \log(C_{-,j}^{(\ell)}/C_{+,j}^{(\ell)})$, in the normalized gauge fixed by Equation (3). The proof and statement is in Section F. The backbone is therefore a *magnitude* summary that cannot cancel even when the signed stage partial dependence function $\text{PD}_j^{(\ell)} = \text{PD}_{+,j}^{(\ell)} - \text{PD}_{-,j}^{(\ell)}$ is near zero, while the tilt captures the signed imbalance and provides directionality. The reconstruction shows that only the $(+)/(-)$ partial dependence functions for each feature need to be visualized in order to faithfully explain the model.

## 6. Theoretical Analysis

We analyze TSL via the Orthogonal Greedy Approximation (OGA) framework (Barron et al., 2008). In doing so, we obtain an $O(1/\sqrt{r})$ approximation-rate for functions in the Sobolev space of dominant mixed smoothness $\mathcal{W}_{\text{mix}}^{(1,1)}$. The class consists of functions $f : \mathcal{X} \to \mathbb{R}$ such that every mixed partial derivative $D_{\boldsymbol{\alpha}} f$ with $\boldsymbol{\alpha} \in \{0,1\}^p$ (i.e., $\|\boldsymbol{\alpha}\|_\infty \leq 1$) is $L^1$-integrable: $D_{\boldsymbol{\alpha}} f \in L^1(\mathcal{X})$. For functions anchored at zero, this reduces to requiring only the highest-order mixed derivative $D_{(1,\dots,1)} f$ to be in $L^1$ (Bungartz & Griebel, 2004, Sec. 2).

**Proposition 6.1.** *Let $\mathcal{X} = [0,1]^p$, assume the data distribution $P_X$ is absolutely continuous with respect to Lebesgue measure on $\mathcal{X}$, and consider the positive dictionary $\mathcal{D}_p^+$. Let $f_r$ denote the OGA approximation to $f$ after $r$ greedy steps over $\mathcal{D}_p^+$ (Section H.1). For any $f \in \mathcal{W}_{\text{mix}}^{(1,1)}(\mathcal{X})$ that is anchored at zero (i.e., $f(\mathbf{x}) = 0$ whenever $x_j = 0$ for some $j$),*

$$\|f - f_r\|_{L^2(P_X)} \leq \frac{2\|D_{(1,\dots,1)} f\|_{L^1(\mathcal{X})}}{\sqrt{r}}. \qquad (8)$$

Barron et al. (2008) show that for any dictionary $\mathcal{D}$ in a Hilbert space, the $r$-term OGA approximation $f_r$ of a target $f$ in the *variation class* $\mathcal{V}_1(\mathcal{D}) = \{\sum_k \lambda_k g_k : g_k \in \mathcal{D}, \sum_k |\lambda_k| < \infty\}$ achieves an approximation rate of $\|f - f_r\|_{L^2(P_X)} \leq 2\|f\|_{\mathcal{V}_1(\mathcal{D})}/\sqrt{r}$. We apply this result with $\mathcal{D} = \mathcal{D}_p^+$, which is the dictionary of normalized non-negative rank-1 products that TSL fits. We then show that $\mathcal{W}_{\text{mix}}^{(1,1)}(\mathcal{X}) \subset \mathcal{V}_1(\mathcal{D}_p^+)$, with the variation norm controlled by the Lebesgue $L^1$-norm of the highest-order mixed derivative, so Barron's rate transfers and yields the bound. The full proof is in Section H.1. Note that the rate achieved in Theorem 6.1 does not depend on the dimension, but the target class tightens with $p$: with increasing dimension the requirement of mixed differentiability up to order $p$ becomes more restrictive.

## 7. Experiments

We evaluate TSL on the OpenML CTR 23 regression suite (Fischer et al., 2023) and compare against state-of-the-art tree-ensemble baselines: XGBoost (Chen & Guestrin, 2016), LightGBM (Ke et al., 2017), and Random Forest (Breiman, 2001; Pedregosa et al., 2011), alongside Explainable Boosting Machines (EBM) (Lou et al., 2013; Nori et al., 2019). We also include the separable model of Beylkin et al. (2009) as *SepALS* in two regimes, SepALS ($r \leq 2$) and SepALS ($r \leq 10$); its implementation, basis, and ALS sweep are described in detail in Section J. TSL was capped at $R \leq 2$ and $R \leq 10$ stages; we additionally include a TSL (1-product) ablation with one unconstrained separable product per stage and $R \leq 4$ (total separation rank $r = R \leq 4$) to isolate

the contribution of the positivity and ordered-difference structure. The tree baselines were capped at depth 2 in the interpretable condition and depth 10 (XGB, LGBM) or 20 (RF) in the black-box condition; EBM was left unchanged as a GA²M-style additive model already constrained to low-order interactions.

Due to computational constraints, we restricted attention to CTR datasets with sample–feature product $n \times p \leq 480{,}000$, yielding 27 regression benchmarks (all datasets in Table 11 up to and including `kings_county`, excluding `Moneyball` as it contained NAs). Following a performance-based selection strategy, we report results for 10 representative datasets spanning the performance spectrum: 4 cases where TSL performs strongly, 3 competitive cases, and 3 challenging cases that demonstrate limitations. Full results on the completed datasets are provided in Section L.

**Protocol.** We randomly split each dataset 80%/20% train/test. Hyperparameters were tuned on the training data via Optuna's Tree-structured Parzen Estimator (TPE) sampler (Akiba et al., 2019; Bergstra et al., 2011), 200 trials per model, each scored by 10-fold cross-validated MSE; the best configuration was then refit on the full training set and evaluated once on the held-out test set. Full search spaces are in Section K.

**Compute.** All experiments ran under a fixed global seed on a shared compute node with dual-socket AMD EPYC 7501 CPUs ($2 \times 32$ cores @ 2.0 GHz) and 256 GB RAM (see Section M for code/models).

Table 1 shows that TSL achieves competitive predictive performance overall. Across the 27 evaluated datasets, TSL ($R \leq 2$ or $R \leq 10$) ranks top-3 of the interpretable group on 17 and is the best interpretable model on 5; SepALS (combined) is best on 13 and EBM on 5 (full per-dataset results in Table 11). TSL excels when the signal has low-rank separable structure. On `socmob`, TSL outperforms the tree-ensemble and EBM baselines by a wide margin (RMSE 9.48 at $R \leq 10$ vs. 17.65–21.71 across XGB, LGBM, RF, and EBM); SepALS ($r \leq 2$) is also strong here (RMSE 7.73), consistent with the known log-additive structure of `socmob` (Biblarz & Raftery, 1993), which matches a separable/multiplicative model well on the original scale.

At matched total separation rank ($\leq 4$: the TSL (1-product) runs at $R \leq 4$ with one product per stage, against TSL ($R \leq 2$) at two ordered-difference products per stage), TSL (1-product) performs noticeably worse on several datasets (e.g., `socmob`: 10.58 vs. 9.87; `naval_propulsion_plant`: 0.0027 vs. 0.0013; `auction_verification`: 1336.51 vs. 1135.80), confirming that at a fixed rank budget the positivity constraint together with the ordered-difference structure $\hat{m}_+^{(\ell)} - \hat{m}_-^{(\ell)}$ is what gives TSL its representational flex-

*Table 1.* Selected benchmark results from the OpenML CTR 23 suite (10 datasets). RMSE on a single 80/20 split; lower is better. (\*\*\*)/(\*\*)/(\*) = best/2nd/3rd within displayed Interpretable or Black-box group, bold = group winner. *These are point estimates without confidence intervals, so the rank marks for near-tied entries are only indicative and may reorder on a different split.* SepALS rank $r$ follows Beylkin et al. (2009) (Section J); for TSL, $R$ stages correspond to total separation rank $\leq 2R$. TSL (1-product) is an ablation with one unconstrained separable component per stage (no positivity constraint). Datasets are grouped by where TSL ranks against the interpretable baselines (Best/Near-Best, Competitive, Challenging). Full benchmark results are available in Section L.

| | Interpretable | | | | | | | | Black-box | | |
|---|---|---|---|---|---|---|---|---|---|---|---|
| Dataset | EBM | LGBM | XGB | SepALS ($r \leq 2$) | SepALS ($r \leq 10$) | TSL (1-product) | TSL ($R \leq 2$) | TSL ($R \leq 10$) | LGBM | RF | XGB |
| *Best/Near-Best* | | | | | | | | | | | |
| brazilian_houses | 3327.29 | 6567.66 | 2528.95 (*) | 3582.69 | 3996.05 | 3194.80 | 2473.98 (**) | **2398.68 (\*\*\*)** | 3200.36 (**) | **2302.72 (\*\*\*)** | 4289.10 (*) |
| auction_verification | 1738.17 | 2155.10 | 1972.82 | 997.08 (*) | 682.20 (**) | 1336.61 | 1135.80 | **624.36 (\*\*\*)** | 409.60 (**) | 1223.57 (*) | **369.34 (\*\*\*)** |
| cpu_activity | 2.3546 (**) | 2.4960 | 2.5281 | 2.5303 | 2.8475 | 2.3826 (*) | 2.5686 | **2.3076 (\*\*\*)** | 2.2102 (**) | 2.4391 (*) | **2.1945 (\*\*\*)** |
| miami_housing | 91777.25 (**) | 100598.20 | 101205.68 | 95686.67 | 99426.86 | 94382.52 (*) | 94466.71 | **89692.96 (\*\*\*)** | 83011.80 (**) | 89215.72 (*) | **82325.09 (\*\*\*)** |
| *Competitive* | | | | | | | | | | | |
| socmob | 20.21 | 21.71 | 19.48 | **7.73 (\*\*\*)** | 22.88 | 10.58 | 9.87 (*) | 9.48 (**) | 20.45 (**) | 20.75 (*) | **17.65 (\*\*\*)** |
| california_housing | **48866.28 (\*\*\*)** | 54495.49 (*) | 55235.15 | 195728.08 | 62162.22 | 55376.04 | 54557.89 | 49376.09 (**) | 45123.03 (**) | 49047.19 (*) | **44971.31 (\*\*\*)** |
| naval_propulsion_plant | 0.0015 | 0.0032 | 0.0075 | 0.0004 (**) | **0.0000 (\*\*\*)** | 0.0027 | 0.0013 | 0.0006 (*) | **0.0009 (\*\*\*)** | 0.0010 (*) | 0.0033 (*) |
| *Challenging* | | | | | | | | | | | |
| QSAR_fish_toxicity | **0.9466 (\*\*\*)** | 0.9970 | 1.0199 | 0.9960 (*) | 0.9704 (**) | 1.0835 | 1.1662 | 1.1143 | 0.9913 (**) | **0.9795 (\*\*\*)** | 1.0047 (*) |
| red_wine | **0.5991 (\*\*\*)** | 0.6093 (*) | 0.6068 (**) | 0.6135 | 0.6107 | 0.6177 | 0.6210 | 0.6438 | 0.5690 (*) | **0.5410 (\*\*\*)** | 0.5585 (**) |
| abalone | 2.2007 (*) | 2.2301 | 2.2221 | **2.1513 (\*\*\*)** | 2.1561 (**) | 2.2598 | 2.2415 | 2.2392 | 2.2400 (*) | **2.1836 (\*\*\*)** | 2.2129 (**) |

**Stage 1: Backbone and 2D partial dependence function**   **Stage 2: Backbone and 2D partial dependence function**

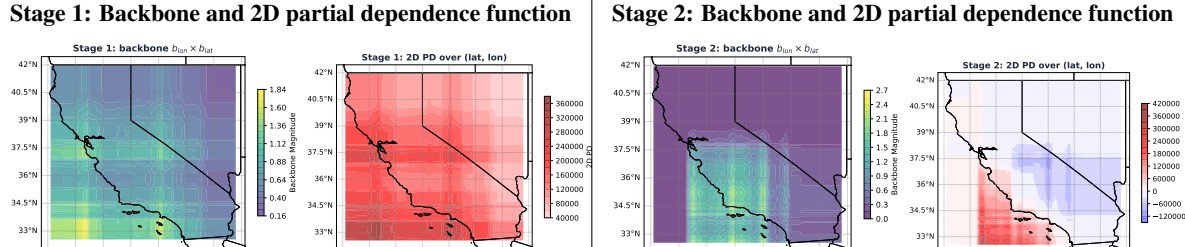

*Figure 1.* Spatial backbone evolution on California housing. Each column is a TSL stage; within each, the left sub-panel shows the learned backbone product $b_j^{(\ell)}(x_j)$ for $j \in \{\text{lat}, \text{lon}\}$ (positive, unitless gating) and the right sub-panel shows the stage's 2D partial dependence surface for (lat, lon) in dollars. Axes are longitude ($x$) and latitude ($y$) in degrees. Stage 2's backbone is high along separable longitude $\times$ latitude bands that include the inland over-prediction region; its 2D PD supplies a negative correction over the desert that offsets Stage 1's over-prediction there.

ibility while preserving stable, interpretable components.

The comparison with SepALS is mixed. SepALS attains lower RMSE on `forest_fires`, `naval_propulsion_plant`, `kin8nm`, `energy_efficiency`, and `grid_stability`; TSL wins `california_housing` at matched rank budgets (49,376 vs. 62,162, a $\approx$ 21% reduction), `auction_verification`, `brazilian_houses`, `cpu_activity`, `miami_housing`, and `kings_county`. A possible explanation for the performance difference is target smoothness: SepALS excels when the data is smooth but may oversmooth sharper features, as Figure 2 shows, whereas TSL's adaptive grid splitting lets it capture more abrupt variations.

### 7.1. California Housing: Interpretability Case Study

We use the California Housing benchmark, with geography (latitude/longitude) and socioeconomic covariates. The dataset features a dominant *spatial interaction* (coastal proximity), making it an ideal interpretability test. We take a representative Optuna-tuned TSL model with $R = 2$ stages from our interpretable setting (the $R \leq 2$ column of Ta-

ble 1).

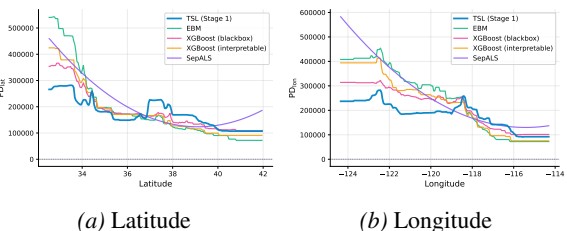

*(a)* Latitude   *(b)* Longitude

*Figure 2.* 1D partial dependence on latitude (left) and longitude (right) across TSL, SepALS, EBM, and XGBoost. Each panel: $x$-axis = coordinate in degrees, $y$-axis = predicted value in dollars (the response); one line per model. SepALS's smooth basis washes out localized location effects; TSL retains spikes near Los Angeles and the Bay Area.

Figure 1 visualizes stagewise backbone and 2D partial dependence for (longitude, latitude). The multiplicative backbone acts as *spatial gating*: a near-zero backbone switches the stage off. Stage 1 is active along the coast, capturing the coastal premium. Its separable form puts longitude peaks at $\approx -118.5, -122.5$ (LA, SF) and a latitude peak at $\approx 37.5$ (Bay Area), so their product over-predicts at points that fall in *both* peak intervals but lie in low-density inland re-

gions (e.g., desert). Stage 2's backbone gates onto exactly these regions and supplies a negative correction that cancels the artifact. Figure 2 compares 1D partial dependence for latitude and longitude across TSL, SepALS, EBM, and XGBoost. TSL captures localized, non-monotone effects (spikes near LA, San Francisco, and a Bay Area latitude bump), whereas EBM and XGBoost yield largely monotone trends. EBM/XGBoost express the location signal through joint (latitude, longitude) structure, so averaging over one coordinate smooths the other; TSL's separable factors instead yield model-native 1D summaries retaining localized structure. Additional figures are in Section I.1; an analogous interpretability case study on Bike Sharing (1D and 2D PD, hour × workingday interaction) appears in Section I.2.

### 7.2. Synthetic Case Study: Masked Interactions

We illustrate the interaction-masking failure mode of Section 3 with a synthetic regression problem with independent features $X_1 \sim \mathcal{N}(0, 1^2)$, $X_2 \sim \mathcal{N}(-1, 1.5^2)$, $X_3 \sim \mathcal{N}(-1, 0.8^2)$ and noise $\varepsilon \sim \mathcal{N}(0, 0.25^2)$. The response is $Y = x_1^2 x_2 (1 + x_3) + \varepsilon$. For hyperparameter search, we run Optuna for 20,000 trials, each training on 10,000 freshly sampled points and evaluating on an independent 50,000-point test sample; after tuning, we refit each model on 100,000 fresh samples and then produce the PD/ICE visualizations (for XGBoost we fix max_depth to 3 and tune the number of trees in [10, 100]). Since $\mathbb{E}[1 + X_3] = 0$, the population $\mathrm{PD}_1(x_1) = x_1^2 \, \mathbb{E}[X_2] \, \mathbb{E}[1 + X_3]$ is identically zero: the 1D marginalization integrates the third-order interaction away, so a strong effect through $x_1$ leaves no signature on the 1D PD. Figure 3 (protocol in Section G) shows that *all fitted models* produce near-zero 1D PD plots, consistent with this population identity, whereas TSL's per-product curves $\mathrm{PD}_{+,j}$ and $\mathrm{PD}_{-,j}$ expose stable per-stage magnitude: the backbone $b_j^{(\ell)}(x_j)$ recovers the quadratic effect of $x_1$ with the tilt remaining small.

## 8. Discussion

*Limitations.* Our approximation-rate guarantee covers a mixed-smoothness class but does not address statistical learning rates or consistency. TSL's principal limitation is the non-identifiability of separable representations: bagged grids can differ across seeds and aggregation can exhibit high variance. The separable structure suits interaction-rich regimes; on largely additive problems EBM remains the stronger interpretable baseline.

*Future work.* Our $O(1/\sqrt{r})$ bound is an approximation rate for the mixed-smoothness function class; extending it to finite-sample learning rates and consistency for the fitted stagewise estimator is open. Aggregation currently averages only the bagged grids that survive a similarity filter and discards the rest; aligning *all* bags on the positive rank-1

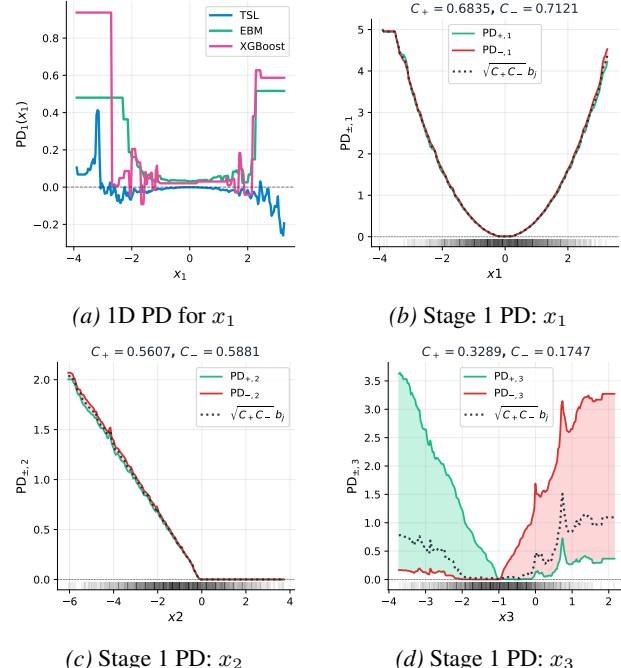

*(a)* 1D PD for $x_1$

*(b)* Stage 1 PD: $x_1$

*(c)* Stage 1 PD: $x_2$

*(d)* Stage 1 PD: $x_3$

*Figure 3.* Masked interaction. (a) All models yield near-zero signed partial dependence for $x_1$ despite a strong quadratic effect through higher-order interactions, because 1D marginalization integrates the interaction signal to zero. (b–d) Stage 1 scaled first-order partial dependence (TSL only) for $\hat{m}_+^{(\ell)}$ and $\hat{m}_-^{(\ell)}$: the backbone retains magnitude even when the signed partial dependence (shaded: green +, red −) is close to zero. $C_\pm := C_{\pm,j}^{(1)}$ for the displayed feature $j$, i.e., the per-stage, per-feature scaling constants from Theorem 5.1. Small tilt on $x_1$, $x_2$ means they mainly amplify the signed signal of $x_3$.

manifold (e.g., Riemannian optimization (Liu & Boumal, 2020)) would cut across-seed variance and use every fit. Finally, the backbone–tilt form (4) is a general representation of a separable model: the factors $b^{(\ell)}$ and $d^{(\ell)}$ need not be fit by the CART-style piecewise-constant estimator used here, but could instead be parametrized by neural networks or other flexible function approximators while retaining the separable interpretable structure.

## 9. Conclusion

We introduced Tensor Separation Learning (TSL), a glass-box regression model that fits a sum of differences of positive rank-1 separable products via a stagewise greedy procedure with orthogonal refitting. By baking separability into the predictor, TSL mitigates the interaction-masking and contamination problems that plague additive explanations in interaction-rich settings while remaining competitive with strong tree-ensemble baselines.

## Impact Statement

TSL is a methodological contribution to interpretable machine learning. By producing a glass-box regression model whose partial dependence plots recover the fitted factor shapes up to per-stage scalars (rather than post-hoc summaries), it can help practitioners inspect interaction structure, identify which features act as multipliers versus directional drivers, and detect regions where the model is active versus balanced. That said, interpretability does not by itself guarantee correctness, fairness, or safety. Like any regression model, TSL should be paired with domain expertise, robustness checks, and fairness analyses before informing consequential decisions.

## Acknowledgments

J. Liu would like to thank Otto G. Roepstorff for the helpful discussions regarding life and work. M. E. Hiabu has carried out this research in association with the project framework InterAct.

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

## A. Grid Tensor Representation Details

For each stage $\ell \leq R$, the $(+)$ and $(-)$ products $\hat{m}_{\pm}^{(\ell)}$ of the learner $\hat{m}^{(\ell)}$ (in the stage model from the main paper) are each a product of univariate piecewise constant functions. This structure implies that each is piecewise constant on a grid $\mathcal{G}^{(\ell)}$ covering the domain $\mathcal{X}$. The two positive products share the same form but each carries its own grid structure; the description below applies to either of them independently.

Formally, $\mathcal{G}^{(\ell)}$ is the Cartesian product of interval partitions $\mathcal{I}_1^{(\ell)}, \ldots, \mathcal{I}_p^{(\ell)}$, where $\mathcal{I}_j^{(\ell)} = \{I_{j,1}^{(\ell)}, \ldots, I_{j,L_j^{(\ell)}}^{(\ell)}\}$ and $L_j^{(\ell)} :=$ $\#\mathcal{I}_j^{(\ell)}$ represents the intervals on which the univariate factor $\hat{m}_{\pm,j}^{(\ell)}$ is constant. In particular:

$$\hat{m}_{\pm,j}^{(\ell)}(x_j) = \sum_{I \in \mathcal{I}_j^{(\ell)}} \hat{v}_{\pm,j,I}^{(\ell)} \mathbb{1}_I(x_j), \tag{9}$$

where $\hat{v}_{\pm,j,I}^{(\ell)}$ is the value attained by $\hat{m}_{\pm,j}^{(\ell)}$ on interval $I$. We refer to the grid $\mathcal{G}^{(\ell)}$ together with the piecewise constant function on this grid as the grid tensor.

In particular, each positive grid tensor is *exactly* a separable piecewise-constant function (a product of univariate piecewise-constant factors). Therefore, after normalization to unit $L^2(P_X)$ norm, each positive grid tensor is an element of the dictionary $\mathcal{D}_p^+$ defined in Section H.1.

The grid tensor representation allows us to refine the learner iteratively via operations similar to tree splitting in CART (Breiman et al., 1984). Unlike CART where a single leaf rectangle is split, our grid splits partition all intervals along axis $j$ simultaneously, creating $\prod_{k \neq j} L_k^{(\ell)}$ new grid cells. This global splitting strategy enables efficient exploration of the function space while maintaining the separable structure.

## B. Forward Additive Approach Details

We learn the estimator stagewise. At stage $\ell$, we define the *outer residuals*

$$R_i^{(\ell-1)} := y^{(i)} - \sum_{k=1}^{\ell-1} \left[\hat{m}_+^{(k)}(\mathbf{x}^{(i)}) - \hat{m}_-^{(k)}(\mathbf{x}^{(i)})\right], \tag{10}$$

where $\hat{m}_{\pm}^{(k)}$ are the $(+)$ and $(-)$ products of stage $k$ as defined in main eq. (2) (each $\hat{m}_{\pm}^{(k)}(\mathbf{x}) = \lambda_{\pm}^{(k)} \prod_{j=1}^p \hat{m}_{\pm,j}^{(k)}(x_j)$ absorbs its stage scalar). We fit a new stage estimator $\hat{m}^{(\ell)}$ to these residuals using grid refinement. In practice, we fit $n_{\text{grids}}$ learners $\hat{m}^{(\ell,1)}, \ldots, \hat{m}^{(\ell,n_{\text{grids}})}$ to the residuals via bagging, and then combine them to obtain the stage estimator $\hat{m}^{(\ell)}$.

Within each stage, we maintain *within-stage residuals* $r_i := R_i - \hat{m}^{(\ell)}(\mathbf{x}^{(i)})$, where $R_i$ is the outer residual and $\hat{m}^{(\ell)}$ is the current (in-progress) stage prediction. At stage start, $\hat{m}^{(\ell)}$ is initialized from the outer residuals (via $\lambda_+^{(\ell)}, \lambda_-^{(\ell)}$ as described in Section C), so $r_i$ reflects the residual after the initial constant fit. We use $i \in I$ to mean that the sample $\mathbf{x}^{(i)}$ falls into interval $I$.

The estimator $\hat{m}^{(\ell)}$ is initialized with both $(+)$ and $(-)$ products as constant functions (all univariate factors set to 1, with $\lambda_{\pm}^{(\ell)}$ initialized from the residual signs as described in Section C), and then iteratively refined via operations similar to tree splitting in CART. After refinement, the stage coefficients $\lambda_+^{(\ell)}$ and $\lambda_-^{(\ell)}$ are refit by least-squares regression of the outer residuals onto the ordered difference $\hat{m}^{(\ell)} = \hat{m}_+^{(\ell)} - \hat{m}_-^{(\ell)}$, with $\hat{m}_{\pm}^{(\ell)}$ as in main eq. (2).

**Notation convention for algorithms.** In the pseudocode below, the per-sample algorithm variables $\hat{m}_+^{(i)}$ and $\hat{m}_-^{(i)}$ denote the *unscaled* per-sample products

$$\hat{m}_{\pm}^{(i)} := \prod_{j=1}^p \hat{m}_{\pm,j}^{(\ell)}(x_j^{(i)}),$$

so that the full stage prediction at $\mathbf{x}^{(i)}$ is $\hat{m}_+^{(\ell)}(\mathbf{x}^{(i)}) - \hat{m}_-^{(\ell)}(\mathbf{x}^{(i)}) = \lambda_+^{(\ell)} \hat{m}_+^{(i)} - \lambda_-^{(\ell)} \hat{m}_-^{(i)}$. The collection of univariate factors $\{\hat{m}_{\pm,j}^{(\ell)}\}_{j=1}^p$ together with the bin-value structure is passed in algorithm pseudocode under the name $\hat{m}_+^{(\ell)}, \hat{m}_-^{(\ell)}$ (the product components), and is used to evaluate the unscaled per-sample $\hat{m}_{\pm}^{(i)}$.

## B.1. Split Scoring Details

At each refinement iteration, we evaluate candidate splits via a closed-form $2 \times 2$ regularized least-squares system. For each candidate split on feature axis $j$ at threshold $s$, we partition the samples into left and right regions $L$ and $R$ based on whether $x_j^{(i)} < s$ or $x_j^{(i)} \geq s$.

**Local refinement loss.** For a region $S \in \{L, R\}$, we score the candidate by how much the regularized within-stage loss can be reduced if both products on $S$ are perturbed by additive increments $\hat{u}_\pm \hat{m}_\pm^{(i)}$ (an additive delta in unscaled-product space; cf. main eq. (5)). Concretely, we minimise

$$\mathcal{L}_S(\hat{u}_+, \hat{u}_-) := \sum_{i \in S} w_i \big(r_i - (\hat{u}_+ \hat{m}_+^{(i)} - \hat{u}_- \hat{m}_-^{(i)})\big)^2 + \alpha\big(\hat{u}_+^2 + \hat{u}_-^2\big), \tag{11}$$

where $r_i$ is the within-stage residual and $\hat{m}_\pm^{(i)} = \prod_{j=1}^{p} \hat{m}_{\pm,j}^{(\ell)}(x_j^{(i)})$ is the unscaled per-sample $(\pm)$ product. Relative to main eq. (5), we parametrise by the delta $\hat{u}_\pm = u_\pm - 1$, so the ridge penalty $\alpha(u_\pm - 1)^2$ in main becomes $\alpha\hat{u}_\pm^2$ here, and the baseline "no-update" point is $\hat{u}_\pm = 0$ (rather than $u_\pm = 1$). The minimiser $(\hat{u}_+^\star, \hat{u}_-^\star)$ has the closed form below; the post-update multiplicative factor applied to the bin value $\hat{v}_{\pm,j,I_S}$ is then $v_\pm^S = 1 + \hat{u}_\pm^\star$ (clamped to $[v_{\min}, v_{\max}]$).

For the left region $L$, we compute sufficient statistics:

$$S_{11}^L = \sum_{i \in L} w_i\big(\hat{m}_+^{(i)}\big)^2, \tag{12}$$

$$S_{22}^L = \sum_{i \in L} w_i\big(\hat{m}_-^{(i)}\big)^2, \tag{13}$$

$$S_{12}^L = -\sum_{i \in L} w_i \hat{m}_+^{(i)} \hat{m}_-^{(i)}, \tag{14}$$

$$t_1^L = \sum_{i \in L} w_i r_i \hat{m}_+^{(i)}, \tag{15}$$

$$t_2^L = -\sum_{i \in L} w_i r_i \hat{m}_-^{(i)}, \tag{16}$$

where $w_i$ are stabilizing weights, $\hat{m}_\pm^{(i)} := \prod_{k=1}^{p} \hat{m}_{\pm,k}^{(\ell)}(x_k^{(i)})$ are the current $(+)$ and $(-)$ product predictions at the data, and $r_i$ are the within-stage residuals.

We then solve the $2 \times 2$ regularized least-squares system:

$$A^L = \begin{pmatrix} S_{11}^L + \alpha & S_{12}^L \\ S_{12}^L & S_{22}^L + \alpha \end{pmatrix}, \quad t^L = \begin{pmatrix} t_1^L \\ t_2^L \end{pmatrix}, \tag{17}$$

where $\alpha$ is a ridge regularization parameter. The solution is $\hat{u}^L = (A^L)^{-1} t^L$, and we compute the multiplicative updates:

$$v_+^L = \text{clamp}(1 + \hat{u}_+^L, v_{\min}, v_{\max}), \quad v_-^L = \text{clamp}(1 + \hat{u}_-^L, v_{\min}, v_{\max}), \tag{18}$$

where $\text{clamp}(x, a, b) = \max(a, \min(b, x))$ ensures positivity and stability. Crucially, we score each candidate at the *clamped* update that is actually applied, not at the unconstrained optimum $\hat{u}^L$: writing $\tilde{u}_\pm^L = v_\pm^L - 1$ for the clamped deltas, the error reduction for the left region is

$$\Delta_L = \mathcal{L}_L(0, 0) - \mathcal{L}_L(\tilde{u}_+^L, \tilde{u}_-^L), \tag{19}$$

where $\mathcal{L}_L$ is the regularised loss (11) restricted to region $L$. The clamped and unconstrained scores coincide whenever $1 + \hat{u}_\pm^L$ already lies in $[v_{\min}, v_{\max}]$. We repeat the same procedure for the right region $R$ and compute the total error reduction $\Delta_{\text{split}} = \Delta_L + \Delta_R$. The split with maximum error reduction is selected.

## C. Complete Algorithm

We provide the complete grid refinement algorithm for fitting a single grid tensor at stage $\ell$. The flowchart in Figure 4 below summarizes the end-to-end fitting loop discussed in Section 4.2: at each stage, outer residuals are fed into a bagged

grid-tensor fit, candidate grids are gauge-aligned and similarity-filtered, the survivors are averaged in log-space, all stage scalars $\{\lambda_\pm^{(k)}\}_{k=1}^\ell$ up to the current stage are jointly refit by least squares against $y$, the outer residuals are updated, and the procedure loops until all $R$ stages are fit. The numbered nodes correspond directly to the steps in Algorithms 10 and 11.

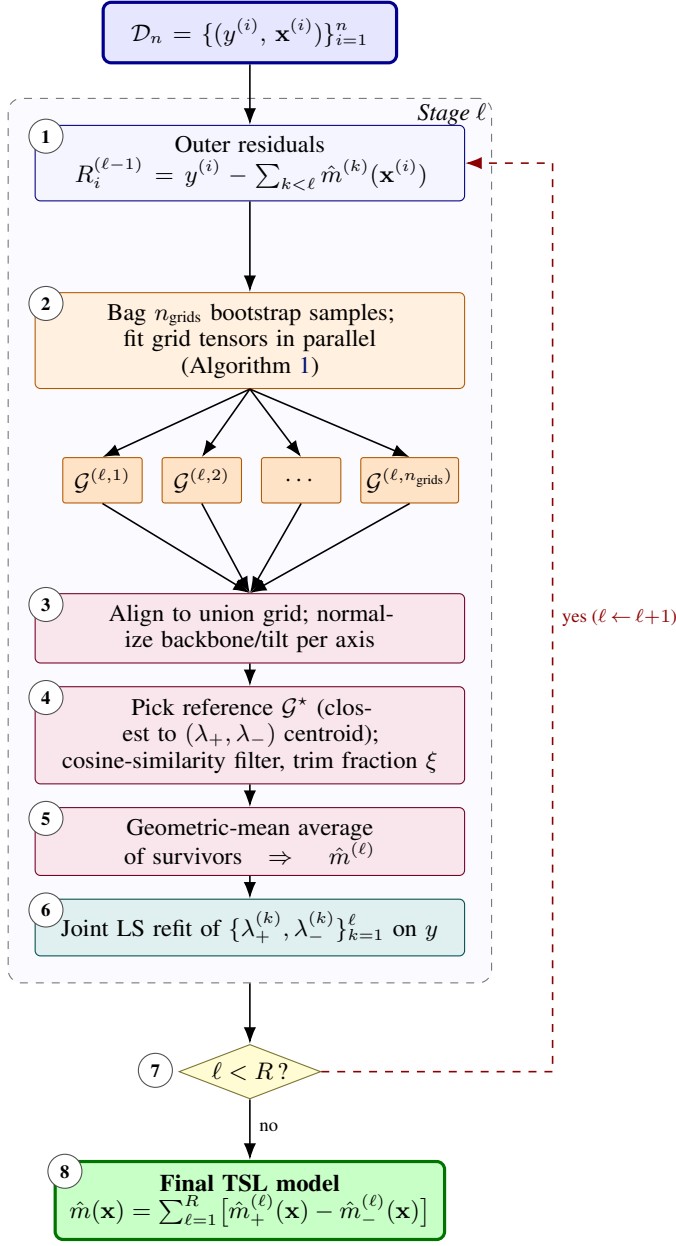

*Figure 4.* TSL fitting pipeline. Within stage $\ell$ (dashed box): outer residuals (blue) feed bagged parallel grid-tensor fits (orange), whose outputs are gauge-aligned, similarity-filtered, and averaged (purple), then all stage scalars $\{\lambda_\pm^{(k)}\}_{k=1}^\ell$ up to the current stage are jointly refit by least squares against $y$ (teal, orthogonal-greedy backfitting). The decision diamond loops back for $\ell < R$ (dashed red); otherwise the procedure exits and returns the assembled model (green). Steps 2–6 implement Algorithm 10 and Algorithm 11.

## C.1. Initialization

At the start of stage $\ell$, we initialize both the $(+)$ and $(-)$ products: $\hat{m}_{+,j}^{(\ell)} = 1$ and $\hat{m}_{-,j}^{(\ell)} = 1$ for all $j$. This corresponds to a uniform grid with a single interval per dimension for each product. The initial intervals are $\mathcal{I}_j = [\min(x_j^{(i)}), \max(x_j^{(i)})]$ where the minimum and maximum are taken over all observations in $\mathcal{D}_n$. The grid tensor is $\mathcal{G} = \{\prod_{j=1}^p \mathcal{I}_j\}$, containing

one rectangle in dimension $p$. We then initialize the stage coefficients $\lambda_+^{(\ell)}$ and $\lambda_-^{(\ell)}$ from the outer residuals $R$ so the stage starts with a constant that matches the signed mass of the residuals. **We set $\lambda_-^{(\ell)} = 0$ only when all outer residuals are non-negative** ($R_i \geq 0$ for all $i$), typically at Stage 1 with non-negative targets, but more generally whenever the outer residuals are all non-negative (positive-only mode); then $\lambda_+^{(\ell)} = 1$. If any residual is negative, we use the full two-tensor initialization: $\lambda_+^{(\ell)} = \max(\epsilon_\lambda, \sum_i w_i R_i^+ / \sum_i w_i)$ and $\lambda_-^{(\ell)} = \max(\epsilon_\lambda, \sum_i w_i R_i^- / \sum_i w_i)$, where $R_i^+ = \max(R_i, 0)$, $R_i^- = \max(-R_i, 0)$, $w_i$ are weights (e.g., $w_i = 1$ or Huber weights), and $\epsilon_\lambda > 0$ is a small floor. The initial stage prediction is $\hat{m}^{(\ell)}(\mathbf{x}) = \hat{m}_+^{(\ell)}(\mathbf{x}) - \hat{m}_-^{(\ell)}(\mathbf{x})$ in the convention of main eq. (2), which with constant univariate factors $\hat{m}_{\pm,j}^{(\ell)} \equiv 1$ equals $\lambda_+^{(\ell)} - \lambda_-^{(\ell)}$ everywhere.

## C.2. Grid Refinement Operations

We consider two types of grid refinement operations: **splits** and **resplits**. The operations apply independently to both the $(+)$ and $(-)$ products, though they share the same grid structure.

**Splits:** Partition an interval $I = [a, b)$ into $I_1 = [a, s)$ and $I_2 = [s, b)$ at threshold $s$ along axis $x_j$, and estimate optimal multipliers for both products on the two new intervals.

**Resplits:** Consider an existing pair of adjacent intervals $I_1, I_2 \in \mathcal{I}_j$ and re-estimate the optimal multipliers for both products on the two intervals.

## C.3. Algorithm Pseudocode

---

**Algorithm 1** Fit Single Grid Tensor (Stage $\ell$)

---

**Input:** outer residuals $R$, data $X$, ridge $\alpha$, clamp bounds $v_{\min}, v_{\max}$, candidate budget split_try, column-subsample rate colsample, min_interval_samples, max refinement iterations n_iter, early-stop tolerance $\varepsilon_\Delta \geq 0$

**Initialize state**

$(\hat{m}_+^{(\ell)}, \hat{m}_-^{(\ell)}, \lambda_+^{(\ell)}, \lambda_-^{(\ell)}) \leftarrow \text{INITIALIZESTATE}(p, R)$

$(\hat{m}_+, \hat{m}_-, r) \leftarrow \text{COMPUTEPREDICTIONS}(\hat{m}_+^{(\ell)}, \hat{m}_-^{(\ell)}, \lambda_+^{(\ell)}, \lambda_-^{(\ell)}, R, X)$

**Grid refinement loop**

**for** $t = 1$ to n_iter **do**

$\quad \Delta_{\text{best}} \leftarrow 0$

$\quad$ action$_{\text{best}} \leftarrow$ None

$\quad$ **Generate candidate splits: sample columns, then intervals, then split points (see Alg. 5)**

$\quad \mathcal{C} \leftarrow \text{SAMPLECANDIDATES}(X, \{\mathcal{I}_j\}_{j=1}^p, \text{split\_try}, \text{colsample}, \text{min\_interval\_samples})$ $\quad$ *($\mathcal{I}_j$ = current intervals on axis j)*

$\quad$ **for** each candidate $(j, I, s) \in \mathcal{C}$ **do**

$\quad\quad \Delta_{\text{split}} \leftarrow \text{EVALUATESPLITCANDIDATE}(j, I, s, \hat{m}_+, \hat{m}_-, r, \alpha, v_{\min}, v_{\max})$

$\quad\quad$ **if** $\Delta_{\text{split}} > \Delta_{\text{best}}$ **then**

$\quad\quad\quad \Delta_{\text{best}} \leftarrow \Delta_{\text{split}}$

$\quad\quad\quad$ action$_{\text{best}} \leftarrow (\text{split}, j, I, s)$

$\quad\quad$ **end if**

$\quad$ **end for**

$\quad$ **Early-stop if no candidate yields enough improvement**

$\quad$ **if** $\Delta_{\text{best}} \leq \varepsilon_\Delta$ **then**

$\quad\quad$ **break**

$\quad$ **end if**

$\quad$ **Apply best split action**

$\quad$ **if** action$_{\text{best}} \neq$ None **then**

$\quad\quad (\hat{m}_+^{(\ell)}, \hat{m}_-^{(\ell)}, \hat{m}_+, \hat{m}_-, r) \leftarrow \text{APPLYSPLIT}(\text{action}_{\text{best}}, \hat{m}_+^{(\ell)}, \hat{m}_-^{(\ell)}, \lambda_+^{(\ell)}, \lambda_-^{(\ell)}, \hat{m}_+, \hat{m}_-, r, R, X, \alpha, v_{\min}, v_{\max})$

$\quad$ **end if**

**end for**

**Refit stage scalars** $(\lambda_+^{(\ell)}, \lambda_-^{(\ell)})$ **on the outer residuals (least-squares projection onto the ordered difference** $\hat{m}_+^{(\ell)} - \hat{m}_-^{(\ell)}$**)**

**return** $\hat{m}^{(\ell)}$

---

---

**Algorithm 2** InitializeState$(p, R)$

---

**Input:** $p$ (number of features), outer residuals $R$

**Output:** $(\hat{m}_+^{(\ell)}, \hat{m}_-^{(\ell)}, \lambda_+^{(\ell)}, \lambda_-^{(\ell)})$

**for** $j = 1$ to $p$ **do**

    $\hat{m}_{+,j}^{(\ell)} \leftarrow 1$                                                        *(uniform grid: single interval per dimension)*

    $\hat{m}_{-,j}^{(\ell)} \leftarrow 1$

**end for**

**Initialize lambdas from residuals. Set $\lambda_-^{(\ell)} = 0$ only if all $R_i \geq 0$ (positive-only mode); else use weighted positive/negative mass.**

**if** all outer residuals non-negative: $R_i \geq 0$ for all $i$ **then**

    $\lambda_+^{(\ell)} \leftarrow 1, \quad \lambda_-^{(\ell)} \leftarrow 0$                                                     *(positive-only mode)*

**else**

    $\lambda_+^{(\ell)} \leftarrow \max\big(\epsilon_\lambda, \ \sum_i w_i R_i^+ / \sum_i w_i\big), \quad R_i^+ := \max(R_i, 0)$

    $\lambda_-^{(\ell)} \leftarrow \max\big(\epsilon_\lambda, \ \sum_i w_i R_i^- / \sum_i w_i\big), \quad R_i^- := \max(-R_i, 0)$

**end if**

**return** $(\hat{m}_+^{(\ell)}, \hat{m}_-^{(\ell)}, \lambda_+^{(\ell)}, \lambda_-^{(\ell)})$

---

**Algorithm 3** ComputePredictions$(\hat{m}_+^{(\ell)}, \hat{m}_-^{(\ell)}, \lambda_+^{(\ell)}, \lambda_-^{(\ell)}, R, X)$

---

**Input:** Component functions $\hat{m}_+^{(\ell)}, \hat{m}_-^{(\ell)}$, coefficients $\lambda_+^{(\ell)}, \lambda_-^{(\ell)}$, outer residuals $R$, data $X$

**Output:** $(\hat{m}_+, \hat{m}_-, r)$ where $\hat{m}_+^{(i)} = \prod_{j=1}^p \hat{m}_{+,j}^{(\ell)}(x_j^{(i)})$ and $\hat{m}_-^{(i)} = \prod_{j=1}^p \hat{m}_{-,j}^{(\ell)}(x_j^{(i)})$ are the per-sample unscaled $(+)/(-)$ products at $\mathbf{x}^{(i)}$, and $r$ are within-stage residuals

**for** $i = 1$ to $n$ **do**

    $\hat{m}_+^{(i)} \leftarrow \prod_{j=1}^p \hat{m}_{+,j}^{(\ell)}(x_j^{(i)})$

    $\hat{m}_-^{(i)} \leftarrow \prod_{j=1}^p \hat{m}_{-,j}^{(\ell)}(x_j^{(i)})$

    $\hat{m}_i^{(\ell)} \leftarrow \lambda_+^{(\ell)} \hat{m}_+^{(i)} - \lambda_-^{(\ell)} \hat{m}_-^{(i)}$                                       *(stage prediction)*

    $r_i \leftarrow R_i - \hat{m}_i^{(\ell)}$                                                 *(within-stage residual)*

**end for**

**return** $(\hat{m}_+, \hat{m}_-, r)$

---

**Algorithm 4** EvaluateSplitCandidate$(j, I, s, \hat{m}_+, \hat{m}_-, r, \alpha, v_{\min}, v_{\max})$

---

**Input:** Feature $j$, interval $I$, split point $s$, unscaled per-sample product values $\hat{m}_+, \hat{m}_-$ (i.e. $\hat{m}_\pm^{(i)} = \prod_{j=1}^p \hat{m}_{\pm,j}^{(\ell)}(x_j^{(i)})$ at the data), residuals $r$, regularization $\alpha$, bounds $v_{\min}, v_{\max}$

**Output:** Error reduction $\Delta_{\text{split}}$

**Partition observations into left $L$ and right $\mathcal{R}$ sides**

$L \leftarrow \{i : x_j^{(i)} < s, i \in I\}$

$\mathcal{R} \leftarrow \{i : x_j^{(i)} \geq s, i \in I\}$

**Evaluate left side**

$(S_{11}^L, S_{22}^L, S_{12}^L, t_1^L, t_2^L) \leftarrow \text{COMPUTESUFFICIENTSTATISTICS}(L, \hat{m}_+, \hat{m}_-, r)$

$(\tilde{u}_+^L, \tilde{u}_-^L, \Delta_L) \leftarrow \text{SOLVETWOTENSOR}(S_{11}^L, S_{22}^L, S_{12}^L, t_1^L, t_2^L, \alpha, v_{\min}, v_{\max})$

**Evaluate right side**

$(S_{11}^R, S_{22}^R, S_{12}^R, t_1^R, t_2^R) \leftarrow \text{COMPUTESUFFICIENTSTATISTICS}(\mathcal{R}, \hat{m}_+, \hat{m}_-, r)$

$(\tilde{u}_+^R, \tilde{u}_-^R, \Delta_R) \leftarrow \text{SOLVETWOTENSOR}(S_{11}^R, S_{22}^R, S_{12}^R, t_1^R, t_2^R, \alpha, v_{\min}, v_{\max})$

$\Delta_{\text{split}} \leftarrow \Delta_L + \Delta_R$

**return** $\Delta_{\text{split}}$

---

---

**Algorithm 5** SampleCandidates($X, \{\mathcal{I}_j\}$, split_try, colsample, min_interval_samples)

---

**Input:** Data $X$, current intervals $\{\mathcal{I}_j\}_{j=1}^p$ per axis (from the grid), split_try (max candidate split points per $(j, I)$), colsample $\in [0, 1]$ (fraction of features to sample), min_interval_samples (minimum observations per resulting sub-interval)

**Output:** Set of candidates $\mathcal{C} \subseteq \{(\text{feature, interval, split point})\}$

**Sample a subset of feature indices**

$J \leftarrow$ random subset of $[p]$ of size $\max(1, \lfloor \text{colsample} \cdot p \rfloor)$

$\mathcal{C} \leftarrow \emptyset$

**For each sampled feature and each interval: build valid positions (min_interval_samples on both sides), then sample up to split_try**

**for** each $j \in J$ **do**

  **for** each interval $I \in \mathcal{I}_j$ **do**

    **Valid positions: splits $s$ with $\geq$ min_interval_samples observations on both sides**

    $\mathcal{V} \leftarrow \{s \in \text{data values in } I : |\{i : x_j^{(i)} < s, i \in I\}| \geq \text{min\_interval\_samples and } |\{i : x_j^{(i)} \geq s, i \in I\}| \geq \text{min\_interval\_samples}\}$

    **Sample up to split_try from $\mathcal{V}$; if $|\mathcal{V}| \leq$ split_try, use all**

    **if** $|\mathcal{V}| \leq$ split_try **then**

      $\mathcal{S} \leftarrow \mathcal{V}$

    **else**

      $\mathcal{S} \leftarrow$ random sample of split_try positions from $\mathcal{V}$

    **end if**

    **for** each $s \in \mathcal{S}$ **do**

      Add $(j, I, s)$ to $\mathcal{C}$

    **end for**

  **end for**

**end for**

**return** $\mathcal{C}$

---

**Algorithm 6** ComputeSufficientStatistics($S, \hat{m}_+, \hat{m}_-, r$)

---

**Input:** Set of observation indices $S$, unscaled per-sample product values $\hat{m}_+, \hat{m}_-$ (i.e. $\hat{m}_\pm^{(i)} = \prod_{j=1}^p \hat{m}_{\pm,j}^{(\ell)}(x_j^{(i)})$ at the data), residuals $r$

**Output:** Sufficient statistics $(S_{11}, S_{22}, S_{12}, t_1, t_2)$

$S_{11} \leftarrow \sum_{i \in S} w_i (\hat{m}_+^{(i)})^2$

$S_{22} \leftarrow \sum_{i \in S} w_i (\hat{m}_-^{(i)})^2$

$S_{12} \leftarrow -\sum_{i \in S} w_i \hat{m}_+^{(i)} \hat{m}_-^{(i)}$                            *(note: negative sign)*

$t_1 \leftarrow \sum_{i \in S} w_i r_i \hat{m}_+^{(i)}$

$t_2 \leftarrow -\sum_{i \in S} w_i r_i \hat{m}_-^{(i)}$                            *(note: negative sign)*

**return** $(S_{11}, S_{22}, S_{12}, t_1, t_2)$

---

---

**Algorithm 7** SolveTwoTensor($S_{11}, S_{22}, S_{12}, t_1, t_2, \alpha, v_{\min}, v_{\max}$)

---

**Input:** Sufficient statistics $S_{11}, S_{22}, S_{12}, t_1, t_2$, regularization $\alpha$, bounds $v_{\min}, v_{\max}$
**Output:** Clamped updates $(\tilde{u}_+, \tilde{u}_-)$ and error reduction $\Delta$
**Build $2 \times 2$ system matrix**
$$A \leftarrow \begin{pmatrix} S_{11} + \alpha & S_{12} \\ S_{12} & S_{22} + \alpha \end{pmatrix}$$
$$t \leftarrow \begin{pmatrix} t_1 \\ t_2 \end{pmatrix}$$
**Solve linear system**
$\hat{u} \leftarrow A^{-1}t$        *(unconstrained optimum $\hat{u} = (\hat{u}_+, \hat{u}_-)^T$)*
**Clamp multipliers to valid range and recompute the applied deltas**
$v_+ \leftarrow \text{clamp}(1 + \hat{u}_+, v_{\min}, v_{\max})$
$v_- \leftarrow \text{clamp}(1 + \hat{u}_-, v_{\min}, v_{\max})$
$\tilde{u}_+ \leftarrow v_+ - 1, \quad \tilde{u}_- \leftarrow v_- - 1$        *(clamped deltas — the values actually applied)*
**Compute error reduction at the clamped update: $\Delta = \mathcal{L}_S(0,0) - \mathcal{L}_S(\tilde{u}_+, \tilde{u}_-)$, where $\mathcal{L}_S$ is the regularised loss (11) on region $S$**
$\Delta \leftarrow 2(t_1\tilde{u}_+ + t_2\tilde{u}_-) - (S_{11}\tilde{u}_+^2 + 2S_{12}\tilde{u}_+\tilde{u}_- + S_{22}\tilde{u}_-^2) - \alpha(\tilde{u}_+^2 + \tilde{u}_-^2)$
**return** $(\tilde{u}_+, \tilde{u}_-, \Delta)$

---

**Algorithm 8** ApplySplit(action, $\hat{m}_+^{(\ell)}, \hat{m}_-^{(\ell)}, \lambda_+^{(\ell)}, \lambda_-^{(\ell)}, \hat{m}_+, \hat{m}_-, r, R, X, \alpha, v_{\min}, v_{\max}$)

---

**Input:** Split action $(j, I, s)$, component functions $\hat{m}_+^{(\ell)}, \hat{m}_-^{(\ell)}$, coefficients $\lambda_+^{(\ell)}, \lambda_-^{(\ell)}$, evaluated product values $\hat{m}_+, \hat{m}_-$, within-stage residuals $r$, outer residuals $R$, data, hyperparameters
**Output:** Updated component functions, evaluated product values, and residuals
**Extract split parameters**
$(j, I, s) \leftarrow$ action
**Partition interval $I$ on axis $j$ at threshold $s$**
$I_L \leftarrow [a, s), I_R \leftarrow [s, b)$ where $I = [a, b)$
**Compute optimal updates for left and right intervals**
$L \leftarrow \{i : x_j^{(i)} \in I_L\}, \mathcal{R} \leftarrow \{i : x_j^{(i)} \in I_R\}$
$(S_{11}^L, S_{22}^L, S_{12}^L, t_1^L, t_2^L) \leftarrow$ COMPUTESUFFICIENTSTATISTICS$(L, \hat{m}_+, \hat{m}_-, r)$
$(\hat{u}_+^L, \hat{u}_-^L, \_) \leftarrow$ SOLVETWOTENSOR$(S_{11}^L, S_{22}^L, S_{12}^L, t_1^L, t_2^L, \alpha, v_{\min}, v_{\max})$
$(S_{11}^R, S_{22}^R, S_{12}^R, t_1^R, t_2^R) \leftarrow$ COMPUTESUFFICIENTSTATISTICS$(\mathcal{R}, \hat{m}_+, \hat{m}_-, r)$
$(\hat{u}_+^R, \hat{u}_-^R, \_) \leftarrow$ SOLVETWOTENSOR$(S_{11}^R, S_{22}^R, S_{12}^R, t_1^R, t_2^R, \alpha, v_{\min}, v_{\max})$
**Update component functions on new intervals**
For $x_j \in I_L$: $\hat{m}_{+,j}^{(\ell)}(x_j) \leftarrow \hat{m}_{+,j}^{(\ell)}(x_j) \cdot (1 + \hat{u}_+^L)$
For $x_j \in I_L$: $\hat{m}_{-,j}^{(\ell)}(x_j) \leftarrow \hat{m}_{-,j}^{(\ell)}(x_j) \cdot (1 + \hat{u}_-^L)$
For $x_j \in I_R$: $\hat{m}_{+,j}^{(\ell)}(x_j) \leftarrow \hat{m}_{+,j}^{(\ell)}(x_j) \cdot (1 + \hat{u}_+^R)$
For $x_j \in I_R$: $\hat{m}_{-,j}^{(\ell)}(x_j) \leftarrow \hat{m}_{-,j}^{(\ell)}(x_j) \cdot (1 + \hat{u}_-^R)$
**Update evaluated product values and residuals**
$(\hat{m}_+, \hat{m}_-, r) \leftarrow$ COMPUTEPREDICTIONS$(\hat{m}_+^{(\ell)}, \hat{m}_-^{(\ell)}, \lambda_+^{(\ell)}, \lambda_-^{(\ell)}, R, X)$
**Update prefix-sum caches for efficient future queries**
**return** $(\hat{m}_+^{(\ell)}, \hat{m}_-^{(\ell)}, \hat{m}_+, \hat{m}_-, r)$

---

---

**Algorithm 9** Backfit($\{\hat{m}_+^{(k)}, \hat{m}_-^{(k)}\}_{k=1}^{\ell}, y, X$)

---

**Input:** All stage components $\{\hat{m}_+^{(k)}, \hat{m}_-^{(k)}\}_{k=1}^{\ell}$, labels $y$, data $X$

**Output:** Optimal stage coefficients $\{\lambda_+^{(k)}, \lambda_-^{(k)}\}_{k=1}^{\ell}$

**Build design matrix** $M \in \mathbb{R}^{n \times 2\ell}$ with entries

$\quad M_{i,k} = \prod_{j=1}^{p} \hat{m}_{+,j}^{(k)}(x_j^{(i)}) \quad$ for $k = 1, \ldots, \ell$

$\quad M_{i,\ell+k} = -\prod_{j=1}^{p} \hat{m}_{-,j}^{(k)}(x_j^{(i)}) \quad$ for $k = 1, \ldots, \ell$

**Solve least squares:** minimize $\|\mathbf{y} - M\boldsymbol{\lambda}\|^2$

$\boldsymbol{\lambda} \leftarrow (M^T M)^{-1} M^T \mathbf{y}$ $\hfill (\boldsymbol{\lambda} = [\lambda_+^{(1)}, \ldots, \lambda_+^{(\ell)}, \lambda_-^{(1)}, \ldots, \lambda_-^{(\ell)}]^T)$

**return** $\{\lambda_+^{(k)}, \lambda_-^{(k)}\}_{k=1}^{\ell}$

---

### C.4. Implementation Details

**Positive-only special case:** We use the simplified (1D) formulation only when *all outer residuals are non-negative* ($R_i \geq 0$ for all $i$), e.g. at stage 1 with non-negative targets. In that case we set $\lambda_-^{(\ell)} = 0$ (the $(-)$ product components $\hat{m}_{-,j}^{(\ell)}$ remain 1, so $\hat{m}_-^{(\ell)}$ contributes zero). The objective then simplifies to 1D ridge-regularized least squares on the $(+)$ product only: $\mathcal{L}_S(\hat{u}_+^S) = \sum_{i \in S} w_i(r_i - \hat{u}_+^S \hat{m}_+^{(i)})^2 + \alpha(\hat{u}_+^S)^2$, with closed-form solution $\hat{u}_+^S = (\sum_{i \in S} w_i r_i \hat{m}_+^{(i)})/(\sum_{i \in S} w_i(\hat{m}_+^{(i)})^2 + \alpha)$, where $\hat{m}_+^{(i)} = \prod_{j=1}^{p} \hat{m}_{+,j}^{(i)}(x_j^{(i)})$ is the unscaled $(+)$ product at $\mathbf{x}^{(i)}$.

**Stabilizing Weights:** The weights $w_i$ are chosen to stabilize the update computation. If $\alpha$ is small and the per-sample unscaled product values $\hat{m}_+^{(i)}$ or $\hat{m}_-^{(i)}$ are close to zero, the system matrix may be near-singular. For standard least squares, we use constant weights $w_i = 1$. For robust Huber loss, we use residual-dependent weights $w_i = \min(1, c/|r_i|)$ where $c \approx 1.345$.

**Splitting Algorithm:** We use two tuning parameters: split_try $\in \mathbb{N}$ is the maximum number of split positions to sample per (feature, interval), and colsample $\in [0, 1]$ determines the fraction of columns to sample. For each $(j, I)$, we first restrict to *valid* positions (split points with at least min_interval_samples observations on both sides); we then sample up to split_try from that set (or use all if there are fewer). Among the resulting candidates, we select the split that maximizes error reduction. We do not terminate when fewer than split_try valid positions exist; we only fail to propose a split when there are zero valid positions.

**Stopping Criteria:** Grid refinement continues until one of: (1) the maximum number of refinement iterations n_iter is reached, (2) no candidate split yields an error reduction $\Delta_{\text{split}}$ exceeding a small threshold, (3) the minimum observations per interval (min_interval_samples) constraint is violated, or (4) no valid splits remain (all violate positivity constraints).

**Efficient Candidate Evaluation:** We maintain prefix-sum caches for each feature axis, enabling $O(1)$ queries for sufficient statistics. The computational complexity for evaluating all candidate splits on a feature is linear in the number of samples, with updates to prefix sums costing $O(pn \log n)$ per split (using Fenwick trees or segment trees for dynamic updates).

**Backfitting:** After completing grid refinement for stage $\ell$, we refit all stage coefficients $\{\lambda_+^{(k)}, \lambda_-^{(k)}\}_{k=1}^{\ell}$ by projecting outer residuals onto the span of all selected stages via least squares. This ensures coefficients are optimal given the current set of stages, improving approximation quality compared to frozen-coefficient greedy methods.

## D. Bagging and Aggregation Details

After fitting $n_{\text{grids}}$ grid tensors at the current stage via bagging, we combine them to obtain the final stage component $\hat{m}^{(\ell)}$. The procedure is applied separately to the $(+)$ and $(-)$ products.

### D.1. Pseudocode: Bagging then Combine

The main paper gives Algorithm 1 for fitting one grid tensor. Below we give the full stage procedure: (i) fit $n_{\text{grids}}$ tensors via bagging, then (ii) combine them into a single stage tensor.

---

**Algorithm 10** Fit Stage $\ell$ with Bagging + Combination

---

**Input:** outer residuals $R^{(\ell-1)}$, data $X$, $n_{\text{grids}}$ bagged grids, bagging scheme $\mathcal{S}$, combine trim $\xi \in [0,1]$, single-grid hyperparameters $(\alpha, v_{\min}, v_{\max}, \text{split\_try}, \text{colsample}, \dots)$

**Output:** combined stage tensor $\hat{m}^{(\ell)}$ (both $(+)$ and $(-)$ products plus $\lambda_+^{(\ell)}, \lambda_-^{(\ell)}$)

**Bagging loop (fit $n_{\text{grids}}$ candidate tensors)**

**for** $c = 1$ to $n_{\text{grids}}$ **do**

    $I_c \leftarrow \text{SAMPLEINDICES}(\mathcal{S}, n)$                                 *(e.g., bootstrap: sample $n$ indices with replacement)*

    $(R_c, X_c) \leftarrow (R^{(\ell-1)}|_{I_c}, X|_{I_c})$

    $\mathcal{G}^{(\ell,c)} \leftarrow \text{FITSINGLEGRIDTENSOR}(R_c, X_c; \alpha, v_{\min}, v_{\max}, \text{split\_try}, \text{colsample}, \dots)$         *(Algorithm 1)*

**end for**

**Combine bagged candidates into a single stage tensor**

$\hat{m}^{(\ell)} \leftarrow \text{COMBINEBAGGEDTWOTENSOR}(\{\mathcal{G}^{(\ell,c)}\}_{c=1}^{n_{\text{grids}}}, X, \xi)$                             *(Algorithm 11)*

**Refit $(\lambda_+^{(\ell)}, \lambda_-^{(\ell)})$ on full data**

$(\lambda_+^{(\ell)}, \lambda_-^{(\ell)}) \leftarrow \text{REFITSTAGECOEFFICIENTS}(\hat{m}^{(\ell)}, R^{(\ell-1)}, X)$

**return** $\hat{m}^{(\ell)}$

---

**Algorithm 11** Combine Bagged Two-Tensor Grids (Reference + Similarity + Averaging)

---

**Input:** candidate grids $\{\mathcal{G}^{(c)}\}_{c=1}^{n_{\text{grids}}}$ (each is two-tensor: $(\{b_j^{(c),k}\}, \{d_j^{(c),k}\}, \lambda_+^{(c)}, \lambda_-^{(c)})$), full data $X$, trim $\xi \in [0,1]$

**Output:** combined grid $\bar{\mathcal{G}}$ (two-tensor representation)

**1. Align grids to a common union grid**

$\{\tilde{\mathcal{G}}^{(c)}\}_{c=1}^{n_{\text{grids}}} \leftarrow \text{REFINETOUNIONGRID}(\{\mathcal{G}^{(c)}\}_{c=1}^{n_{\text{grids}}})$                       *(union over all split points per axis)*

**2. Normalize each bag (gauge-fixing so similarities compare *shapes*)**

**for** $c = 1$ to $n_{\text{grids}}$ **do**

    $\tilde{\mathcal{G}}^{(c)} \leftarrow \text{NORMALIZEPERAXIS}(\tilde{\mathcal{G}}^{(c)}; X)$ {per axis $j$, subtract empirical mean of $\log b_j$ and of $d_j$ over $X$}

**end for**

**3. Choose a reference grid (closest to the $(\lambda_+, \lambda_-)$ centroid)**

$\mathcal{G}^\star \leftarrow \arg\min_{c \in \{1,\dots,n_{\text{grids}}\}} \sum_{c'=1}^{n_{\text{grids}}} \left[ (\lambda_+^{(c)} - \lambda_+^{(c')})^2 + (\lambda_-^{(c)} - \lambda_-^{(c')})^2 \right]$

**4. Score candidates by similarity to the reference**

**for** $c = 1$ to $n_{\text{grids}}$ **do**

    Compute per-point backbone and tilt summaries:

    $\mathbf{b}_c \leftarrow \left[ \prod_{j=1}^p b_j^{(c),k_j(i)} \right]_{i=1}^n$ and $\mathbf{d}_c \leftarrow \left[ \sum_{j=1}^p d_j^{(c),k_j(i)} \right]_{i=1}^n$

    $\text{sim}_b \leftarrow \frac{\mathbf{b}^\star \cdot \mathbf{b}_c}{\|\mathbf{b}^\star\| \|\mathbf{b}_c\|}, \quad \text{sim}_d \leftarrow \frac{\mathbf{d}^\star \cdot \mathbf{d}_c}{\|\mathbf{d}^\star\| \|\mathbf{d}_c\|}$

    $\text{score}(c) \leftarrow \frac{(\text{sim}_b + 1)(\text{sim}_d + 1)}{4} \in [0,1]$                                 *(as in Eq. (23))*

**end for**

**5. Trim and keep top candidates**

Keep the top $K = \lceil (1 - \xi) n_{\text{grids}} \rceil$ indices by $\text{score}(c)$; call this kept set $\mathcal{K}$

**6. Average normalized components (geometric mean in log-space of $a_\pm$), reconstruct $(b, d)$**

$\bar{a}_{\pm,j}^k \leftarrow \exp\left( \frac{1}{|\mathcal{K}|} \sum_{c \in \mathcal{K}} \log a_{\pm,j}^{(c),k} \right)$ where $a_{\pm,j}^k = b_j^k e^{\pm d_j^k}$

Reconstruct $(\bar{b}, \bar{d})$ from $\bar{a}_+, \bar{a}_-$ (e.g. $\bar{b}_j^k = \sqrt{\bar{a}_{+,j}^k \bar{a}_{-,j}^k}$, $\bar{d}_j^k = \frac{1}{2} \log(\bar{a}_{+,j}^k / \bar{a}_{-,j}^k)$)

**7. Combined lambdas:** $\lambda_\pm^{\text{combined}} \leftarrow \exp\left( \frac{1}{|\mathcal{K}|} \sum_{c \in \mathcal{K}} \log \lambda_\pm^{(c)} \right)$

**return** $\bar{\mathcal{G}} := (\{\hat{m}_{+,j}^{\text{combined}}\}_{j=1}^p, \{\hat{m}_{-,j}^{\text{combined}}\}_{j=1}^p, \lambda_+^{\text{combined}}, \lambda_-^{\text{combined}})$

---

## D.2. Candidate Selection via Similarity

We select a reference grid $\mathcal{G}^\star$ as the candidate closest to the $(\lambda_+, \lambda_-)$ centroid, i.e. the one minimizing the sum of squared $\lambda$-distances to all other candidates:

$$\mathcal{G}^\star = \underset{c \in \{1,\ldots,n_{\text{grids}}\}}{\arg\min} \sum_{c'=1}^{n_{\text{grids}}} [(\lambda_+^{(c)} - \lambda_+^{(c')})^2 + (\lambda_-^{(c)} - \lambda_-^{(c')})^2]. \tag{20}$$

For each candidate grid $\mathcal{G}_c$ and the reference $\mathcal{G}^\star$, we compute per-point backbone and tilt contributions:

$$\mathbf{b}_c = \left[ \prod_{j=1}^{p} b_j^{k_j(i)} \right]_{i=1}^{n}, \quad \mathbf{d}_c = \left[ \sum_{j=1}^{p} d_j^{k_j(i)} \right]_{i=1}^{n}, \tag{21}$$

where $k_j(i)$ is the interval index for $x_j^{(i)}$ on axis $j$. We compute cosine similarities:

$$\text{sim}_b(\mathcal{G}^\star, \mathcal{G}_c) = \frac{\mathbf{b}^\star \cdot \mathbf{b}_c}{\|\mathbf{b}^\star\| \|\mathbf{b}_c\|}, \quad \text{sim}_d(\mathcal{G}^\star, \mathcal{G}_c) = \frac{\mathbf{d}^\star \cdot \mathbf{d}_c}{\|\mathbf{d}^\star\| \|\mathbf{d}_c\|}. \tag{22}$$

The combined similarity score is the rescaled product of the two cosine similarities (each shifted into $[0, 2]$ and divided by $4$ so the score lies in $[0, 1]$):

$$\text{sim}_{\text{combined}}(\mathcal{G}_c) = \frac{(\text{sim}_b(\mathcal{G}^\star, \mathcal{G}_c) + 1)(\text{sim}_d(\mathcal{G}^\star, \mathcal{G}_c) + 1)}{4} \in [0, 1]. \tag{23}$$

We select the top $K = \lceil (1 - \xi) n_{\text{grids}} \rceil$ candidates by combined similarity, where $\xi \in [0, 1]$ is a trimming threshold (default: $\xi = 0$ keeps all candidates). Section I.4 (Figure 14) gives a worked illustration of why this filtering step is necessary: bagged grids can converge to qualitatively distinct backbone representations of the same fitted stage, and filtering against the reference removes the competing branch before averaging.

## D.3. Normalization and Averaging

We work in the backbone/tilt parametrization: $\hat{m}_{\pm,j}(x_j) = b_j(x_j) e^{\pm d_j(x_j)}$ with $b_j = \sqrt{\hat{m}_{+,j} \hat{m}_{-,j}}$ and $d_j = \frac{1}{2} \log(\hat{m}_{+,j} / \hat{m}_{-,j})$.

**Normalization (before similarity and averaging).** For each bag we apply a gauge-fixing step so that similarities compare *shapes* rather than scale or constant offsets. For each axis $j$, we center $\log b_j$ and $d_j$ by subtracting their empirical means over the data: $\log b_j \leftarrow \log b_j - \frac{1}{n} \sum_{i=1}^{n} \log b_j(x_j^{(i)})$ and $d_j \leftarrow d_j - \frac{1}{n} \sum_{i=1}^{n} d_j(x_j^{(i)})$. Lambdas are not modified in this step.

**Averaging.** We average the normalized $(a_+, a_-)$ factors (where $a_{\pm,j}^k = b_j^k e^{\pm d_j^k}$) using geometric mean in log-space across the selected bags, then reconstruct the combined backbone and tilt $(b, d)$. Combined stage scalars are set by geometric mean: $\lambda_\pm^{\text{combined}} = \exp\left( \frac{1}{|\mathcal{K}|} \sum_{c \in \mathcal{K}} \log \lambda_\pm^{(c)} \right)$.

The combined estimator at stage $\ell$ is:

$$\hat{m}^{(\ell)}(\mathbf{x}) = \lambda_+^{\text{combined}} \cdot \prod_{j=1}^{p} \hat{m}_{+,j}^{\text{combined}}(x_j) - \lambda_-^{\text{combined}} \cdot \prod_{j=1}^{p} \hat{m}_{-,j}^{\text{combined}}(x_j). \tag{24}$$

After combination, we refit the stage coefficients via least squares to ensure optimal scaling.

## E. Proof of Partial Dependence Identity

**Proposition E.1** (Partial Dependence Decomposition, restated from Theorem 5.1). *Consider a TSL estimator $\hat{m}$ as defined in (2). For any stage $\ell$ and coordinate $j$, define the stage-wise constant $c_{\pm,j}^{(\ell)} := \mathbb{E}\left[ \prod_{k \neq j} \hat{m}_{\pm,k}^{(\ell)}(X_k) \right]$. Then the 1D partial dependence function of $\hat{m}_\pm^{(\ell)}(\mathbf{x}) = \lambda_\pm^{(\ell)} \prod_{j=1}^{p} \hat{m}_{\pm,j}^{(\ell)}(x_j)$ on coordinate $j$ satisfies*

$$\text{PD}_{\pm,j}^{(\ell)}(x_j) := \mathbb{E}\left[ \hat{m}_\pm^{(\ell)}(x_j, X_{(-j)}) \right] = C_{\pm,j}^{(\ell)} \hat{m}_{\pm,j}^{(\ell)}(x_j), \tag{25}$$

*i.e., each 1D partial dependence curve is the 1D factor shape times a constant $C_{\pm,j}^{(\ell)} = c_{\pm,j}^{(\ell)} \lambda_{\pm}^{(\ell)}$. Moreover, letting $\bar{m}_{\pm}^{(\ell)} := \mathbb{E}\left[\hat{m}_{\pm}^{(\ell)}(X)\right], \quad Z_{\pm}^{(\ell)} := \mathbb{E}\left[\prod_{j=1}^{p} \mathrm{PD}_{\pm,j}^{(\ell)}(X_j)\right]$, the stage admits the following exact reconstruction*

$$\hat{m}_{\pm}^{(\ell)}(\mathbf{x}) = \frac{\bar{m}_{\pm}^{(\ell)}}{Z_{\pm}^{(\ell)}} \prod_{j=1}^{p} \mathrm{PD}_{\pm,j}^{(\ell)}(x_j). \tag{26}$$

*Consequently, each stage (and thus any downstream explanation primitive built from the stage factors) is computable from 1D partial dependence summaries up to a single scalar normalizer per stage and sign branch.*

*Proof of Theorem 5.1.* For the forward direction, by definition of partial dependence and the separable structure:

$$\mathrm{PD}_{\pm,j}^{(\ell)}(x_j) := \mathbb{E}\left[\hat{m}_{\pm}^{(\ell)}(x_j, X_{(-j)})\right] \tag{27}$$

$$= \mathbb{E}\left[\lambda_{\pm}^{(\ell)} \hat{m}_{\pm,j}^{(\ell)}(x_j) \prod_{k \neq j} \hat{m}_{\pm,k}^{(\ell)}(X_k)\right] \tag{28}$$

$$= \lambda_{\pm}^{(\ell)} \hat{m}_{\pm,j}^{(\ell)}(x_j) \cdot \mathbb{E}\left[\prod_{k \neq j} \hat{m}_{\pm,k}^{(\ell)}(X_k)\right] \tag{29}$$

$$=: \lambda_{\pm}^{(\ell)} \hat{m}_{\pm,j}^{(\ell)}(x_j) \cdot c_{\pm,j}^{(\ell)} \tag{30}$$

$$= C_{\pm,j}^{(\ell)} \hat{m}_{\pm,j}^{(\ell)}(x_j), \tag{31}$$

where the factorization follows because $\hat{m}_{\pm,j}^{(\ell)}(x_j)$ is constant with respect to the expectation over $X_{(-j)}$.

For the reconstruction of $\hat{m}_{\pm}^{(\ell)}$ from the partial dependence functions, define

$$\bar{m}_{\pm}^{(\ell)} := \mathbb{E}\left[\hat{m}_{\pm}^{(\ell)}(X)\right], \qquad Z_{\pm}^{(\ell)} := \mathbb{E}\left[\prod_{j=1}^{p} \mathrm{PD}_{\pm,j}^{(\ell)}(X_j)\right].$$

Inverting (25) as $\hat{m}_{\pm,j}^{(\ell)}(x_j) = \mathrm{PD}_{\pm,j}^{(\ell)}(x_j)/C_{\pm,j}^{(\ell)}$ and substituting into the definition of $\hat{m}_{\pm}^{(\ell)}$:

$$\hat{m}_{\pm}^{(\ell)}(\mathbf{x}) = \lambda_{\pm}^{(\ell)} \prod_{j=1}^{p} \hat{m}_{\pm,j}^{(\ell)}(x_j) \tag{32}$$

$$= \lambda_{\pm}^{(\ell)} \prod_{j=1}^{p} \frac{\mathrm{PD}_{\pm,j}^{(\ell)}(x_j)}{C_{\pm,j}^{(\ell)}} \tag{33}$$

$$= \frac{\lambda_{\pm}^{(\ell)}}{\prod_{j=1}^{p} C_{\pm,j}^{(\ell)}} \prod_{j=1}^{p} \mathrm{PD}_{\pm,j}^{(\ell)}(x_j) \tag{34}$$

$$= \frac{\bar{m}_{\pm}^{(\ell)}}{Z_{\pm}^{(\ell)}} \prod_{j=1}^{p} \mathrm{PD}_{\pm,j}^{(\ell)}(x_j), \tag{35}$$

where the last equality identifies the unknown prefactor by taking expectations on both sides of (33): $\bar{m}_{\pm}^{(\ell)} = \frac{\lambda_{\pm}^{(\ell)}}{\prod_j C_{\pm,j}^{(\ell)}} Z_{\pm}^{(\ell)}$, so $\frac{\lambda_{\pm}^{(\ell)}}{\prod_j C_{\pm,j}^{(\ell)}} = \bar{m}_{\pm}^{(\ell)}/Z_{\pm}^{(\ell)}$. $\square$

## F. Backbone and Tilt Reconstruction from Partial Dependence

The forward parametrization Equation (3) in the main text is a bijection between $(b_j^{(\ell)}, d_j^{(\ell)})$ and $(\hat{m}_{+,j}^{(\ell)}, \hat{m}_{-,j}^{(\ell)})$ with closed-form inverse

$$
\begin{aligned}
b_j^{(\ell)}(x_j) &= \sqrt{\hat{m}_{+,j}^{(\ell)}(x_j)\hat{m}_{-,j}^{(\ell)}(x_j)}, \\
d_j^{(\ell)}(x_j) &= \frac{1}{2}\log\left(\frac{\hat{m}_{+,j}^{(\ell)}(x_j)}{\hat{m}_{-,j}^{(\ell)}(x_j)}\right).
\end{aligned}
\tag{36}
$$

The backbone/tilt parametrization can also be recovered directly from the 1D partial dependence functions. This provides an interpretable decomposition of each stage's effect into magnitude (backbone) and signed direction (tilt).

**Proposition F.1** (Backbone and Tilt Reconstruction from partial dependence plots). *For a TSL stage $\ell$ in the two-tensor regime $(\lambda_+^{(\ell)}, \lambda_-^{(\ell)} > 0)$ and coordinate $j$, let $\mathrm{PD}_{+,j}^{(\ell)}(x_j)$ and $\mathrm{PD}_{-,j}^{(\ell)}(x_j)$ denote the 1D partial dependence functions of the positive and negative products, as defined in (6). In the normalized gauge fixed by (36), the backbone $b_j^{(\ell)}(x_j)$ and tilt $d_j^{(\ell)}(x_j)$ reconstruct as:*

$$
b_j^{(\ell)}(x_j) = \frac{1}{\sqrt{C_{+,j}^{(\ell)} C_{-,j}^{(\ell)}}}\sqrt{\mathrm{PD}_{+,j}^{(\ell)}(x_j)\,\mathrm{PD}_{-,j}^{(\ell)}(x_j)}, \qquad d_j^{(\ell)}(x_j) = \frac{1}{2}\log\left(\frac{\mathrm{PD}_{+,j}^{(\ell)}(x_j)\,C_{-,j}^{(\ell)}}{\mathrm{PD}_{-,j}^{(\ell)}(x_j)\,C_{+,j}^{(\ell)}}\right),
\tag{37}
$$

*provided that $b_j(x_j) \neq 0$.*

To interpret the result, start from the 1D partial dependence of the *stage* on coordinate $j$:

$$
\mathrm{PD}_j^{(\ell)}(x_j) = \mathrm{PD}_{+,j}^{(\ell)}(x_j) - \mathrm{PD}_{-,j}^{(\ell)}(x_j),
$$

where the last equality follows from $\hat{m}^{(\ell)} = \hat{m}_+^{(\ell)} - \hat{m}_-^{(\ell)}$ and linearity of expectation. Because $\mathrm{PD}_j^{(\ell)}$ is a signed difference of two positive products, *cancellation between $\mathrm{PD}_{+,j}^{(\ell)}$ and $\mathrm{PD}_{-,j}^{(\ell)}$* can occur: $\mathrm{PD}_j^{(\ell)}(x_j)$ may be close to zero even when both $\mathrm{PD}_{+,j}^{(\ell)}(x_j)$ and $\mathrm{PD}_{-,j}^{(\ell)}(x_j)$ are large, so the stage's PD curve can appear flat and hide substantial sensitivity in magnitude. This is one way TSL can fit data whose 1D PD has been masked by higher-order interactions (Section 7.2): the two positive products carry the magnitude that the signed difference erases. The backbone/tilt view makes this explicit: (37) shows that $b_j^{(\ell)}(x_j)$ is proportional to $\sqrt{\mathrm{PD}_{+,j}^{(\ell)}(x_j)\,\mathrm{PD}_{-,j}^{(\ell)}(x_j)}$, a positive quantity that cannot cancel and therefore reflects *magnitude* even when $\mathrm{PD}_j^{(\ell)}$ is near zero. The tilt, recovered from the ratio of the partial dependence curves, captures the signed imbalance between the two products and provides directionality.

*Proof of Theorem F.1.* From (25), we have:

$$
\mathrm{PD}_{+,j}^{(\ell)}(x_j) = \lambda_+^{(\ell)}\,\hat{m}_{+,j}^{(\ell)}(x_j) \cdot c_{+,j}^{(\ell)},
\tag{38}
$$

$$
\mathrm{PD}_{-,j}^{(\ell)}(x_j) = \lambda_-^{(\ell)}\,\hat{m}_{-,j}^{(\ell)}(x_j) \cdot c_{-,j}^{(\ell)}.
\tag{39}
$$

Multiplying these two expressions and taking the square root:

$$
\sqrt{\mathrm{PD}_{+,j}^{(\ell)}(x_j)\,\mathrm{PD}_{-,j}^{(\ell)}(x_j)} = \sqrt{\lambda_+^{(\ell)}\lambda_-^{(\ell)}\,\hat{m}_{+,j}^{(\ell)}(x_j)\hat{m}_{-,j}^{(\ell)}(x_j)\,c_{+,j}^{(\ell)}c_{-,j}^{(\ell)}}
\tag{40}
$$

$$
= \sqrt{\lambda_+^{(\ell)}\lambda_-^{(\ell)}\,c_{+,j}^{(\ell)}c_{-,j}^{(\ell)}} \cdot \sqrt{\hat{m}_{+,j}^{(\ell)}(x_j)\hat{m}_{-,j}^{(\ell)}(x_j)}.
\tag{41}
$$

From the definition of $c_{\pm,j}^{(\ell)} = \mathbb{E}[\prod_{k\neq j}\hat{m}_{\pm,k}^{(\ell)}(X_k)]$, we have:

$$
c_{+,j}^{(\ell)}c_{-,j}^{(\ell)} = \mathbb{E}\left[\prod_{k\neq j}\hat{m}_{+,k}^{(\ell)}(X_k)\right]\mathbb{E}\left[\prod_{k\neq j}\hat{m}_{-,k}^{(\ell)}(X_k)\right].
\tag{42}
$$

Recalling $C_{\pm,j}^{(\ell)} = \lambda_{\pm}^{(\ell)} c_{\pm,j}^{(\ell)}$, this gives

$$\sqrt{\mathrm{PD}_{+,j}^{(\ell)}(x_j)\,\mathrm{PD}_{-,j}^{(\ell)}(x_j)} = \sqrt{C_{+,j}^{(\ell)} C_{-,j}^{(\ell)}} \cdot b_j^{(\ell)}(x_j), \qquad (43)$$

where $b_j^{(\ell)}(x_j) = \sqrt{\hat{m}_{+,j}^{(\ell)}(x_j)\hat{m}_{-,j}^{(\ell)}(x_j)}$ from the backbone definition. Rearranging yields the backbone formula in (37).

For the tilt, from the backbone/tilt parametrization we have $\hat{m}_{+,j}^{(\ell)}(x_j) = b_j^{(\ell)}(x_j)e^{d_j^{(\ell)}(x_j)}$ and $\hat{m}_{-,j}^{(\ell)}(x_j) = b_j^{(\ell)}(x_j)e^{-d_j^{(\ell)}(x_j)}$. Under the assumption that $b_j^{(\ell)}(x_j) \neq 0$ (equivalently, $\hat{m}_{+,j}^{(\ell)}(x_j), \hat{m}_{-,j}^{(\ell)}(x_j) > 0$), the ratio and logarithm are well-defined, so:

$$d_j^{(\ell)}(x_j) = \frac{1}{2}\log\left(\frac{\hat{m}_{+,j}^{(\ell)}(x_j)}{\hat{m}_{-,j}^{(\ell)}(x_j)}\right). \qquad (44)$$

Substituting the partial dependence expressions from (6):

$$d_j^{(\ell)}(x_j) = \frac{1}{2}\log\left(\frac{\hat{m}_{+,j}^{(\ell)}(x_j)}{\hat{m}_{-,j}^{(\ell)}(x_j)}\right) \qquad (45)$$

$$= \frac{1}{2}\log\left(\frac{\mathrm{PD}_{+,j}^{(\ell)}(x_j)/(\lambda_{+}^{(\ell)} c_{+,j}^{(\ell)})}{\mathrm{PD}_{-,j}^{(\ell)}(x_j)/(\lambda_{-}^{(\ell)} c_{-,j}^{(\ell)})}\right) \qquad (46)$$

$$= \frac{1}{2}\log\left(\frac{\mathrm{PD}_{+,j}^{(\ell)}(x_j)\,\lambda_{-}^{(\ell)}\,c_{-,j}^{(\ell)}}{\mathrm{PD}_{-,j}^{(\ell)}(x_j)\,\lambda_{+}^{(\ell)}\,c_{+,j}^{(\ell)}}\right), \qquad (47)$$

which yields the tilt formula in (37). $\qquad\square$

## G. Synthetic Masked Interaction: Full Mathematical Treatment

**Data-generating process.** The true function is $m(x_1, x_2, x_3) = x_1^2 x_2(1 + x_3)$ with independent Gaussians $X_1 \sim \mathcal{N}(0, 1^2)$, $X_2 \sim \mathcal{N}(-1, 1.5^2)$, $X_3 \sim \mathcal{N}(-1, 0.8^2)$ and additive Gaussian noise $\varepsilon \sim \mathcal{N}(0, 0.25^2)$, so the response is $Y = m(X_1, X_2, X_3) + \varepsilon$. The choice $\mathbb{E}[1 + X_3] = 0$ produces the masking effect discussed below.

**Hyperparameter search.** For each model (TSL, EBM, XGBoost) we tune hyperparameters via Optuna's TPE sampler (Akiba et al., 2019; Bergstra et al., 2011) with 20,000 trials; each trial samples a fresh training set of 10,000 points from the DGP above and evaluates MSE on an independent 50,000-point test sample drawn from the same DGP. The search spaces for TSL ($R \leq 2$), EBM, and XGBoost are those in Table 2, 9, and 5 respectively, except that for XGBoost we fix max_depth$= 3$ and tune the number of trees in $[10, 100]$. After tuning, the best configuration for each model is refit on a fresh sample of 100,000 points and used to produce the partial-dependence and ICE figures discussed in the main text (Figure 3) and the additional ICE figures of Section I.3.

**Population masking.** By construction, the population 1D partial dependence for $x_1$ is:

$$\mathrm{PD}_1(x_1) = \mathbb{E}[m(x_1, X_2, X_3)] = \mathbb{E}[x_1^2 X_2(1 + X_3)] = x_1^2 \mathbb{E}[X_2]\mathbb{E}[1 + X_3] = 0, \qquad (48)$$

since $\mathbb{E}[X_2] = -1$ and $\mathbb{E}[1 + X_3] = 0$. Despite $x_1$ having a strong quadratic effect through the third-order interaction, the 1D marginal operator integrates the interaction signal to zero, leaving the PD identically flat. This is not cancellation between two terms in a model decomposition — it is the marginalization itself, acting on a higher-order interaction, that erases the signature of $x_1$.

The ICE curves reveal the hidden signal: for each sample $i$, the ICE curve is:

$$\mathrm{ICE}_i(x_1) = m(x_1, x_2^{(i)}, x_3^{(i)}) = x_1^2 x_2^{(i)}(1 + x_3^{(i)}) = c_i x_1^2, \qquad (49)$$

where $c_i = x_2^{(i)}(1 + x_3^{(i)})$ is the signed amplitude. The mean of these amplitudes is zero: $\mathbb{E}[c_i] = \mathbb{E}[X_2(1 + X_3)] = \mathbb{E}[X_2]\mathbb{E}[1 + X_3] = 0$, explaining why the partial dependence plot is flat.

TSL addresses this limitation by exposing per-stage partial dependence plots of $\hat{m}_+^{(\ell)}$ and $\hat{m}_-^{(\ell)}$ naturally via the backbone structure. The stage model separates positive and negative mass before subtraction: $\hat{m}^{(\ell)}(\mathbf{x}) = \hat{m}_+^{(\ell)}(\mathbf{x}) - \hat{m}_-^{(\ell)}(\mathbf{x})$ with $\hat{m}_{\pm,j}^{(\ell)} > 0$ (main eq. (2)). The backbone/tilt parametrization defines the per-axis backbone $b_j^{(\ell)} = \sqrt{\hat{m}_{+,j}^{(\ell)} \hat{m}_{-,j}^{(\ell)}}$ as a shared magnitude between the two products.

For this masked-interaction example, TSL learns to represent the quadratic effect $x_1^2$ in the backbone $b_1^{(\ell)}(x_1)$, while the signed amplitude $c_i$ is captured through the interaction with $x_2$ and $x_3$. The per-stage partial dependence plots of $\hat{m}_+$ and $\hat{m}_-$ for $x_1$ expose nontrivial structure even when the signed partial dependence is flat, demonstrating that the backbone provides a stable 1D summary of effect magnitude without requiring practitioners to invent nonstandard statistics.

## H. Proofs of Theoretical Results

### H.1. Approximation Rate of TSL

Let $\mathcal{X} = \mathcal{X}_1 \times \cdots \times \mathcal{X}_p$ be the feature space, and let $\mathcal{H} = L^2(P_X)$ be the Hilbert space of square-integrable functions on $\mathcal{X}$ with respect to the background distribution $P_X$. We first define the set of univariate piecewise constant functions for each dimension $j \in [p]$ as

$$\mathcal{S}_j = \left\{ h : \mathcal{X}_j \to \mathbb{R} \,\middle|\, h(t) = \sum_{k=1}^{K} c_k \mathbb{1}_{I_k}(t), K \in \mathbb{N}, \bigcup_{k=1}^{K} I_k = \mathcal{X}_j \right\}. \tag{50}$$

Let $\mathcal{D}_p^+ \subset L^2(P_X)$ denote the dictionary of normalized non-negative rank-one product functions

$$\mathcal{D}_p^+ = \left\{ g(x) = \prod_{j=1}^{p} h_j(x_j) \,\middle|\, h_j \in \mathcal{S}_j, \ g(x) \geq 0, \ \|g\|_{L^2(P_X)} = 1 \right\}. \tag{51}$$

In particular, each fitted positive grid tensor (Section A) is a product of non-negative univariate piecewise-constant factors, so its normalized version is *exactly* an element of $\mathcal{D}_p^+$. Note that fitted TSL components are positive by (2); the dictionary $\mathcal{D}_p^+$ is the larger closure that also includes atoms vanishing on subsets, which the OGA construction below uses via indicator functions.

Our approach follows the analysis of (Barron et al., 2008). We introduce the following class of functions which are those that admit an expansion in the positive dictionary $\mathcal{D}_p^+$ with non-negative atoms and signed coefficients.

$$\mathcal{V}_1(\mathcal{D}_p^+) = \left\{ f = \sum_{k=1}^{\infty} \lambda_k g_k \,\middle|\, g_k \in \mathcal{D}_p^+, \ \lambda_k \in \mathbb{R}, \ \sum_{k=1}^{\infty} |\lambda_k| < \infty \right\}. \tag{52}$$

Although the atoms in $\mathcal{D}_p^+$ are non-negative, the expansion coefficients are real-valued. Thus negative contributions are represented by negative coefficients, equivalently by placing the corresponding positive rank-one product as the negative product of an ordered-difference TSL stage.

In (Barron et al., 2008), this class is denoted as $\mathcal{L}_1(\mathcal{D})$, we however adopt the notation $\mathcal{V}_1(\mathcal{D}_p^+)$ to avoid confusion with the $L^1$ norm. We define the hypothesis space for the TSL functions as sums of non-negative rank-1 products with real-valued coefficients

$$\mathcal{F}_r = \left\{ f : \mathcal{X} \to \mathbb{R} \,\middle|\, f(x) = \sum_{\ell=1}^{r} \lambda_\ell g_\ell(\mathbf{x}), \lambda_\ell \in \mathbb{R}, g_\ell \in \mathcal{D}_p^+ \right\}. \tag{53}$$

We also define the variation norm

$$\|f\|_{\mathcal{V}_1(\mathcal{D}_p^+)} := \inf \left\{ \sum_k |\lambda_k| : f = \sum_k \lambda_k g_k, \ g_k \in \mathcal{D}_p^+, \ \lambda_k \in \mathbb{R} \right\}. \tag{54}$$

(Barron et al., 2008) analyzes the behavior of the orthogonal greedy algorithm (OGA) for approximating a target function $f \in \mathcal{V}_1(\mathcal{D}_p^+)$ by a sequence of functions $f_\ell \in \mathcal{F}_\ell$. In OGA, the algorithm is initialized with $f_0 := 0$, and at each step $\ell$, we define the residual $\rho_{\ell-1} := f - f_{\ell-1}$ (we use $\rho$ rather than $r$ for the residual function to avoid collision with the OGA

atom-count $r$ in Theorem H.2 and Theorem 6.1). The algorithm then selects the function $g_\ell \in \mathcal{D}_p^+$ that maximizes the correlation with the residual, i.e.,

$$g_\ell := \arg\max_{g \in \mathcal{D}_p^+} |\langle \rho_{\ell-1}, g \rangle|. \tag{55}$$

The selected atom is the one whose best signed scalar multiple gives the largest one-step reduction in residual norm:

$$g_\ell \in \arg\max_{g \in \mathcal{D}_p^+} |\langle \rho_{\ell-1}, g \rangle| = \arg\min_{g \in \mathcal{D}_p^+} \min_{\alpha \in \mathbb{R}} \|\rho_{\ell-1} - \alpha g\|_{L^2(P_X)}^2. \tag{56}$$

The approximation $f_\ell$ is then the orthogonal projection of $f$ onto $V_\ell = \text{span}(\{g_1, \ldots, g_\ell\})$. This means that $f_\ell = \sum_{k=1}^\ell \lambda_k^{(\ell)} g_k$, where the coefficients $\{\lambda_k^{(\ell)}\}_{k=1}^\ell$ are re-optimized at each step $\ell$. This backfitting step updates all previous coefficients at each iteration. Our TSL algorithm follows this OGA/backfitting viewpoint: after selecting a new component, we refit coefficients on the span of the selected components by least squares projection (in finite samples), which is the regime analyzed by the greedy-approximation guarantees cited here.

The question remains: what exactly does the function class $\mathcal{V}_1(\mathcal{D}_p^+)$ represent? The bounded variation norm constraint is saying that $f$ admits a sparse representation of the functions in $\mathcal{D}_p^+$. In other words, $f$ is well approximated by a sum of a few functions from $\mathcal{D}_p^+$. When the dimension $p$ becomes large, the condition that $f \in \mathcal{V}_1(\mathcal{D}_p^+)$ becomes more restrictive. (Barron et al., 2008) specifically mentions that in the case where we are working on $H := L^2([0,1]^p)$, and $D_p$ is the wavelet basis, then $\mathcal{V}_1(D_p)$ is a subset of the Besov space $B_1^s(L^1)$ which means that $f$ needs to have all its derivatives of order less than or equal to $p/2$ in $L^1$. In particular, it might seem that Theorem H.2 avoids the curse of dimensionality, but this is not the case, the target function class just becomes more restrictive as the dimension $p$ increases. Stated differently, the apparent $O(1/\sqrt{r})$ rate comes at the cost of a target class whose smoothness constraint strengthens with dimension: as $p$ grows we require mixed differentiability up to order $p$ (in the anchored case, integrability of the full $p$-th order mixed derivative) in $L^1$.

The class $\mathcal{V}_1(\mathcal{D}_p^+)$ seems quite abstract without a concrete example. We will argue that $\mathcal{V}_1(\mathcal{D}_p^+)$ can be used to approximate a large class of functions. We will restrict our attention to the case where $\mathcal{X} = [0,1]^p$, and argue that the Sobolev space of dominant mixed smoothness $\mathcal{W}_{\text{mix}}^{(1,1)}([0,1]^p)$ can be approximated well by $\mathcal{V}_1(\mathcal{D}_p^+)$. This space consists of functions with $L^1$ integrable mixed partial derivatives up to order $p$. We define for $\boldsymbol{\alpha} = (\alpha_1, \alpha_2, \ldots, \alpha_p) \in \mathbb{N}_0^p$, the mixed partial derivative of order $\boldsymbol{\alpha}$ is defined as

$$D_{\boldsymbol{\alpha}} f(\mathbf{x}) = \frac{\partial^{\|\boldsymbol{\alpha}\|_1} f(\mathbf{x})}{\partial x_1^{\alpha_1} \partial x_2^{\alpha_2} \cdots \partial x_p^{\alpha_p}}, \tag{57}$$

where

$$\|\boldsymbol{\alpha}\|_\infty = \max_{j \in [p]} \alpha_j \quad \text{and} \quad \|\boldsymbol{\alpha}\|_1 = \sum_{j \in [p]} \alpha_j. \tag{58}$$

We consider the following class of functions:

$$\mathcal{W}_{\text{mix}}^{(1,1)}(\mathcal{X}) = \left\{ f : \mathcal{X} \to \mathbb{R} \,\big|\, D_{\boldsymbol{\alpha}} f \in L^1(\mathcal{X}) \text{ for all } \|\boldsymbol{\alpha}\|_\infty \leq 1 \right\}. \tag{59}$$

If we restrict our attention to $\mathcal{X} = [0,1]^p$ and consider functions $f \in \mathcal{W}_{\text{mix}}^{(1,1)}(\mathcal{X})$ that are anchored at zero (i.e. $f(\mathbf{x}) = 0$ whenever $x_j = 0$ for some $j$), then the condition that $D_{\boldsymbol{\alpha}} f \in L^1(\mathcal{X})$ for all $\|\boldsymbol{\alpha}\|_\infty \leq 1$ is equivalent to the requirement that only the highest order partial derivative is absolutely integrable, i.e. that $D_{\boldsymbol{\alpha}} f \in L^1(\mathcal{X})$ for $\boldsymbol{\alpha} = (1, 1, \ldots, 1)$. This is a consequence of the multivariate fundamental theorem of calculus.

We will use the following form of Barron et al.'s OGA certificate bound.

**Theorem H.1** (Barron et al. (2008), Theorem 2.3). *Let $H$ be a Hilbert space and let $\mathcal{D} \subset H$ be a normalized dictionary. For any $f \in H$, let $f_r$ be the $r$-term OGA approximation to $f$ with respect to $\mathcal{D}$, using exact greedy selection. Then for every $h \in \mathcal{V}_1(\mathcal{D})$,*

$$\|f - f_r\|_H^2 \leq \|f - h\|_H^2 + \frac{4\|h\|_{\mathcal{V}_1(\mathcal{D})}^2}{r}. \tag{60}$$

*Here $\mathcal{V}_1(\mathcal{D})$ is Barron et al.'s $\mathcal{L}_1(\mathcal{D})$ variation class.*

We now state the main approximation result.

**Theorem H.2.** *Let $\mathcal{X} = [0,1]^p$, assume $P_X \ll \lambda_p$, and let $f \in \mathcal{W}_{\text{mix}}^{(1,1)}(\mathcal{X})$ be anchored at zero, meaning $f(x) = 0$ whenever $x_j = 0$ for at least one $j$. Let $f_r$ be the $r$-term OGA approximation with respect to the positive dictionary $\mathcal{D}_p^+$. Then*

$$\|f - f_r\|_{L^2(P_X)} \leq \frac{2\|D_{(1,\ldots,1)}f\|_{L^1(\mathcal{X})}}{\sqrt{r}}. \tag{61}$$

*Proof.* Let $f$ be as stated in the theorem. Define

$$c(t) := D_{(1,\ldots,1)}f(t). \tag{62}$$

Since $f$ is anchored at zero, the multivariate fundamental theorem of calculus gives the Sobolev representation

$$f(x) = \int_{[0,1]^p} c(t) \prod_{j=1}^p \mathbb{1}(t_j \leq x_j)\, dt. \tag{63}$$

This representation holds Lebesgue-a.e., and hence also $P_X$-a.e. by the assumption $P_X \ll \lambda_p$.

For $t \in [0,1]^p$, define

$$h_t(x) := \prod_{j=1}^p \mathbb{1}(t_j \leq x_j). \tag{64}$$

Whenever $\|h_t\|_{L^2(P_X)} > 0$, define

$$g_t := \frac{h_t}{\|h_t\|_{L^2(P_X)}}. \tag{65}$$

Then $g_t \in \mathcal{D}_p^+$. Points $t$ for which $\|h_t\|_{L^2(P_X)} = 0$ can be ignored, since they do not affect the $L^2(P_X)$ representation. Thus,

$$f(x) = \int_{[0,1]^p} c(t)\|h_t\|_{L^2(P_X)} g_t(x)\, dt. \tag{66}$$

Let $\mathcal{P}_m = \{R_{m,k}\}_k$ be a sequence of finite rectangular partitions of $[0,1]^p$, where

$$R_{m,k} = \prod_{j=1}^p [a_{m,k,j}, b_{m,k,j})$$

up to the usual convention at the right endpoint. Define the mesh size by

$$|\mathcal{P}_m| := \max_{k,j}(b_{m,k,j} - a_{m,k,j}),$$

and assume that $|\mathcal{P}_m| \to 0$ as $m \to \infty$. For each cell $R_{m,k}$, choose the lower-left representative

$$t_{m,k} := (a_{m,k,1}, \ldots, a_{m,k,p}).$$

Define $\pi_m(t) = t_{m,k}$ whenever $t \in R_{m,k}$. Then

$$\|\pi_m(t) - t\|_\infty \leq |\mathcal{P}_m| \to 0$$

for every $t \in [0,1]^p$, and moreover $\pi_m(t)_j \leq t_j$ for every coordinate $j$. Define the signed cell mass

$$\eta_{m,k} := \int_{R_{m,k}} c(t)\, dt, \tag{67}$$

and the coefficients

$$\lambda_{m,k} := \|h_{t_{m,k}}\|_{L^2(P_X)} \eta_{m,k}. \tag{68}$$

Finally, define the certificate function which we will show lies in $\mathcal{V}_1(\mathcal{D}_p^+)$ and which approximates $f$ in $L^2(P_X)$ as $m \to \infty$:

$$s_m := \sum_{k:\, \|h_{t_{m,k}}\|_{L^2(P_X)} > 0} \lambda_{m,k} g_{t_{m,k}} = \sum_k \eta_{m,k} h_{t_{m,k}}. \tag{69}$$

We will first show that $s_m \to f$ in $L^2(P_X)$. Write the construction in integral form

$$s_m = \sum_k \left( \int_{R_{m,k}} c(t)\, dt \right) h_{t_{m,k}} = \int_{\mathcal{X}} c(t) h_{\pi_m(t)}\, dt. \tag{70}$$

Hence

$$f - s_m = \int_{\mathcal{X}} c(t) \big( h_t - h_{\pi_m(t)} \big)\, dt. \tag{71}$$

By the triangle inequality for the $L^2(P_X)$-valued integral,

$$\|f - s_m\|_{L^2(P_X)} \leq \int_{\mathcal{X}} |c(t)|\, \|h_t - h_{\pi_m(t)}\|_{L^2(P_X)}\, dt. \tag{72}$$

Fix $t \in [0,1]^p$. Since $\pi_m(t)_j \leq t_j$ and $\pi_m(t)_j \to t_j$, we have for every $x_j \in [0,1]$,

$$\mathbb{1}\{\pi_m(t)_j \leq x_j\} \longrightarrow \mathbb{1}\{t_j \leq x_j\}. \tag{73}$$

Indeed, if $x_j < t_j$, then eventually $\pi_m(t)_j > x_j$, while if $x_j \geq t_j$, then $\pi_m(t)_j \leq t_j \leq x_j$ for all $m$. Therefore

$$h_{\pi_m(t)}(x) \to h_t(x) \qquad \text{for every } x \in [0,1]^p. \tag{74}$$

Since $h_t$ and $h_{\pi_m(t)}$ are $\{0,1\}$-valued,

$$|h_t(x) - h_{\pi_m(t)}(x)|^2 \leq 1. \tag{75}$$

Dominated convergence with respect to $P_X$ gives

$$\|h_t - h_{\pi_m(t)}\|^2_{L^2(P_X)} = \int_{[0,1]^p} |h_t(x) - h_{\pi_m(t)}(x)|^2\, dP_X(x) \to 0. \tag{76}$$

Thus

$$\|h_t - h_{\pi_m(t)}\|_{L^2(P_X)} \to 0 \qquad \text{for every } t. \tag{77}$$

Moreover,

$$|c(t)|\, \|h_t - h_{\pi_m(t)}\|_{L^2(P_X)} \leq |c(t)|. \tag{78}$$

Since $c \in L^1(\mathcal{X})$, dominated convergence in $t$ yields

$$\int_{\mathcal{X}} |c(t)|\, \|h_t - h_{\pi_m(t)}\|_{L^2(P_X)}\, dt \to 0. \tag{79}$$

Therefore

$$\|f - s_m\|_{L^2(P_X)} \to 0. \tag{80}$$

It remains to show that $s_m \in \mathcal{V}_1(\mathcal{D}_p^+)$, we show this by bounding the variation norm of $s_m$. Since $0 \leq h_{t_{m,k}} \leq 1$, we have

$$\|h_{t_{m,k}}\|_{L^2(P_X)} \leq 1 \tag{81}$$

Therefore

$$\sum_k |\lambda_{m,k}| = \sum_k \|h_{t_{m,k}}\|_{L^2(P_X)} |\eta_{m,k}| \leq \sum_k |\eta_{m,k}| \leq \int_{[0,1]^p} |c(t)|\, dt. \tag{82}$$

Hence

$$\|s_m\|_{\mathcal{V}_1(\mathcal{D}_p^+)} \leq \|D_{(1,\dots,1)}f\|_{L^1([0,1]^p)}. \tag{83}$$

Applying Theorem H.1 with $H = L^2(P_X)$, $\mathcal{D} = \mathcal{D}_p^+$, and $h = s_m$, we obtain

$$\|f - f_r\|^2_{L^2(P_X)} \leq \|f - s_m\|^2_{L^2(P_X)} + \frac{4\|s_m\|^2_{\mathcal{V}_1(\mathcal{D}_p^+)}}{r}. \tag{84}$$

Using the uniform variation bound above and taking $m \to \infty$ gives

$$\|f - f_r\|^2_{L^2(P_X)} \leq \frac{4\|D_{(1,\dots,1)}f\|^2_{L^1([0,1]^p)}}{r}. \tag{85}$$

Taking square roots proves the claim. $\qquad\square$

The main assumption made in the theorem is that the target function $f$ belongs to the Sobolev space of dominant mixed smoothness $\mathcal{W}_{\text{mix}}^{(1,1)}$ (anchored at 0). Physically, this requires the mixed partial derivative of order $p$, denoted $D_{(1,\dots,1)}f$, to be integrable. This is a strong assumption; as the dimension $p$ increases, the requirement of having $p$-th order mixed differentiability becomes increasingly restrictive. The result is an approximation guarantee, not a finite-sample statistical rate.

Barron et al. also provide approximation bounds for interpolation spaces between $L^2(P_X)$ and $\mathcal{V}_1(\mathcal{D})$. A concrete characterization of these spaces for the positive separable piecewise-constant dictionary $\mathcal{D}_p^+$ is left for future work.

# I. Additional Interpretability Figures

### I.1. California Housing: Additional Figures

**How local explanations are computed.** A TSL local explanation is read off the fitted model exactly, with no post-hoc attribution step (e.g., SHAP). The prediction at any point $\mathbf{x}$ already decomposes additively over stages, $\hat{m}(\mathbf{x}) = \sum_{\ell=1}^{R} \hat{m}^{(\ell)}(\mathbf{x})$, and each stage contributes the signed gap $\hat{m}^{(\ell)}(\mathbf{x}) = \hat{m}_+^{(\ell)}(\mathbf{x}) - \hat{m}_-^{(\ell)}(\mathbf{x}) = 2\, b^{(\ell)}(\mathbf{x})\sinh(d^{(\ell)}(\mathbf{x}))$ between its two positive products $\hat{m}_\pm^{(\ell)}(\mathbf{x}) = b^{(\ell)}(\mathbf{x})\, e^{\pm d^{(\ell)}(\mathbf{x})}$ (main eq. (4)). Across stages these contributions sum *exactly* to $\hat{m}(\mathbf{x})$—nothing is approximated or redistributed—so the explanation is a complete additive account of the prediction: the reader sees which stages move it, in which direction, and by how much, in the units of $y$. The figure renders this decomposition for a single point in three panels, described next. The stage quantities are obtained by direct evaluation, not optimization: for each coordinate we look up the bin of $x_j$ in the stage's tree grids to read the per-feature backbone $b_j^{(\ell)}(x_j)$ (magnitude) and tilt $d_j^{(\ell)}(x_j)$ (direction), then aggregate into the stage backbone $b^{(\ell)}(\mathbf{x}) = b_0^{(\ell)} \prod_j b_j^{(\ell)}(x_j)$ and stage tilt $d^{(\ell)}(\mathbf{x}) = d_0^{(\ell)} + \sum_j d_j^{(\ell)}(x_j)$, where $b_0^{(\ell)} = \sqrt{\lambda_+^{(\ell)}\lambda_-^{(\ell)}}$ and $d_0^{(\ell)} = \frac{1}{2}\log(\lambda_+^{(\ell)}/\lambda_-^{(\ell)})$ absorb the stage backfitting scalars $\lambda_\pm^{(\ell)}$ (which carry the scale, after unit-norm normalization of the factors). For a positive-only stage ($\lambda_-^{(\ell)} = 0$), $b_0^{(\ell)} = 0$ and $d_0^{(\ell)}$ is undefined; the stage reduces to $\hat{m}^{(\ell)}(\mathbf{x}) = \lambda_+^{(\ell)} \prod_j b_j^{(\ell)}(x_j)\, e^{d_j^{(\ell)}(x_j)}$ (matching main eq. (2)).

**Left panel (stage contributions).** Each bar is one stage's signed contribution $\hat{m}^{(\ell)}(\mathbf{x})$ to the prediction, in the units of $y$ (USD): green when positive, pink when negative, ordered by magnitude and laid out as a waterfall that accumulates from 0 to the model output $\hat{m}(\mathbf{x})$ (marked on the axis). Because the stages sum exactly to the prediction, the panel is a complete additive account of how the point's value is built up. For the desert point (top), stage 1 supplies the bulk (+\$298,745), stage 3 is the largest downward correction (−\$115,753), and the remaining stages add progressively smaller signed adjustments, netting \$149,491.

**Middle panel (relative backbone magnitude).** For each stage, a single bar is split into stacked per-feature segments giving each feature's share of the stage's backbone *magnitude*—how strongly it gates the stage on or off at $\mathbf{x}$. We score feature $j$ by $|\log b_j^{(\ell)}(x_j)|$ (a backbone far from 1 gates strongly; $b_j^{(\ell)} \approx 1$ is neutral), pool features below a small tolerance into "Other," and normalize the rest to sum to $100\%$. Wider segments are the features that most set how active and how large the stage is; the panel reports *which* features drive a stage, not the direction of their effect. For the desert point, the geographic coordinates dominate the leading stages (longitude $52\%$, latitude $18\%$ in stage 1).

**Right panel (signed tilt).** For each stage, each bar is one feature's signed tilt $d_j^{(\ell)}(x_j)$, with the stage intercept $d_0^{(\ell)}$ shown as "Intercept": green is a positive push (toward the $(+)$ product), pink a negative push (toward the $(-)$ product). This supplies the direction the middle panel omits, since the sign of the stage is governed by the total tilt $d^{(\ell)}(\mathbf{x}) = d_0^{(\ell)} + \sum_j d_j^{(\ell)}(x_j)$ (main eq. (4)); a feature with a large backbone share but near-zero tilt therefore amplifies the stage without steering its sign. The intercept carries the stage's baseline $(+)/(-)$ imbalance, with the per-feature tilts as directional adjustments around it. Stage 1 is a special case: the California response (median house value) is positive everywhere, so the stage-1 target—the raw response, since no stage has yet fired—is all-positive, and stage 1 is fit in *positive-only* mode ($\lambda_-^{(1)} = 0$, no $(-)$ product; Section C). Such a stage cannot change sign—it only scales the prediction upward—so the panel renders it as a single large positive baseline (the "Intercept," $+34.82$ for the desert point) rather than a set of signed per-feature tilts, and its feature structure lives entirely in the magnitude (middle) panel. Sign-bearing direction enters only from stage 2 onward, where the residuals take both signs.

The two instances below show this three-panel decomposition for a desert and a coastal point.

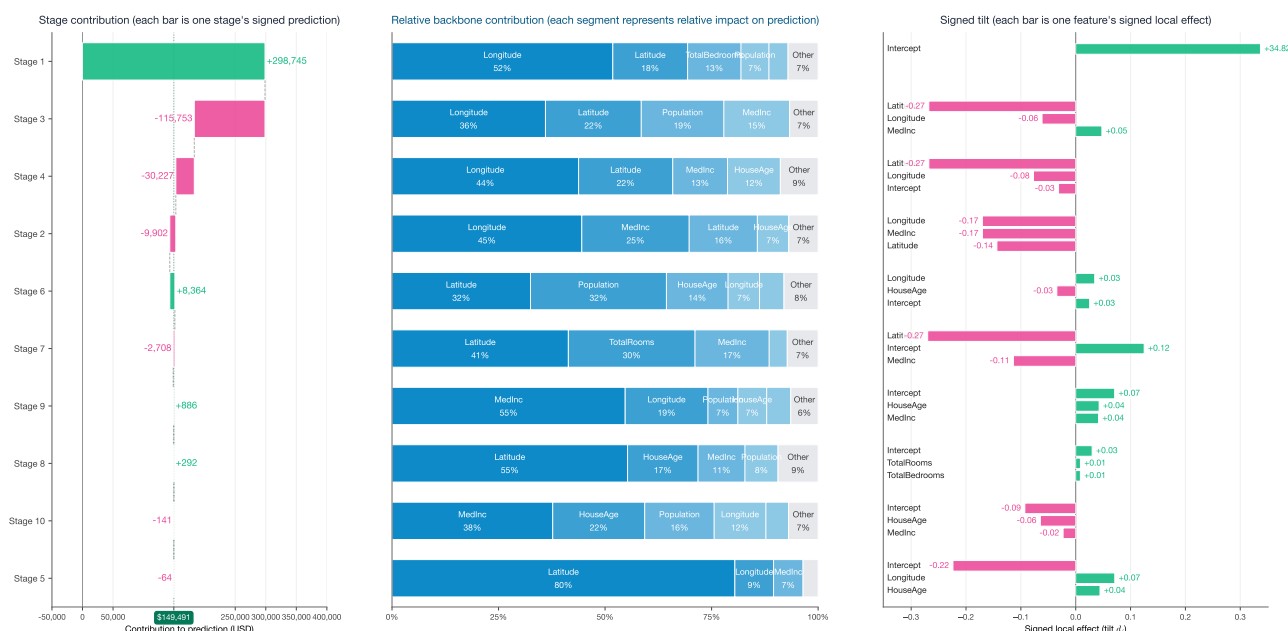

*(a)* Desert point: longitude −118.43, latitude 37.40 (eastern California, Inyo County / Owens Valley); house age 19, total rooms 2460, total bedrooms 405, population 1225, households 425, median income 4.16; median house value $141,500.

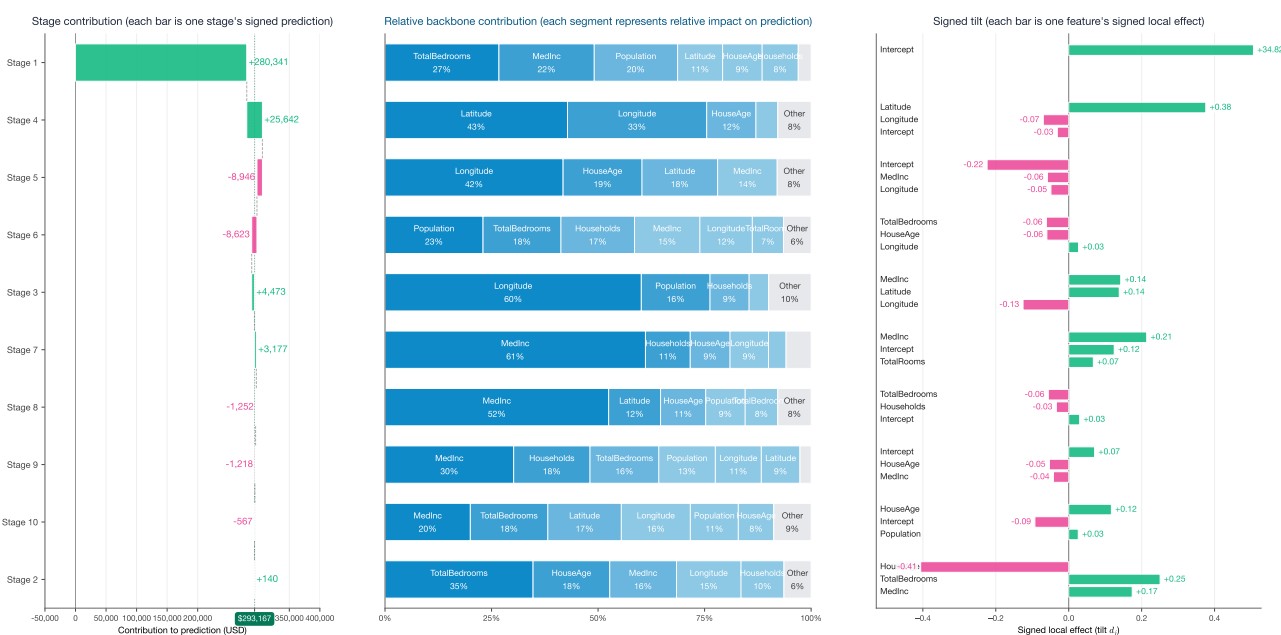

*(b)* Coastal point: longitude −118.25, latitude 34.05 (Los Angeles area, e.g. Long Beach / South Bay); house age 52, total rooms 2806, total bedrooms 1944, population 2232, households 1605, median income 0.68; median house value $350,000.

*Figure 5.* Local explanations: stages are sorted by absolute contribution to the prediction, giving an interpretation of how each stage affects the prediction. For the desert point, stage 3 is especially important as it corrects for the prediction made in stage 1; for the coastal point, the majority of the mass is already captured in stage 1, so later stages matter less. Each California Housing record summarizes a census block group, so the total rooms, total bedrooms, population, and households reported above are block-group aggregates over hundreds of dwellings rather than counts for a single home.

**Parsimonious subselection via additivity.** The stage contributions $\hat{m}^{(\ell)}(\mathbf{x})$ are additive: the prediction at $\mathbf{x}$ is $\hat{m}(\mathbf{x}) = \sum_{\ell=1}^{R} \hat{m}^{(\ell)}(\mathbf{x})$. Thus, for any subset of stages $\mathcal{L} \subseteq [R]$, the partial sum $\sum_{\ell \in \mathcal{L}} \hat{m}^{(\ell)}(\mathbf{x})$ is a well-defined component

of the prediction; in particular, one can always subselect certain stages and discard the rest, yielding a reduced model $\hat{m}_{\mathcal{L}}(\mathbf{x}) := \sum_{\ell \in \mathcal{L}} \hat{m}^{(\ell)}(\mathbf{x})$ that uses only $|\mathcal{L}|$ stages. A potential use of the stage decompositions in Figure 5 is the following: if the individuals we wish to predict in the future are known to lie in a particular feature region $\mathcal{R}$ (e.g., coastal or inland), we can first inspect which stages contribute most to points in $\mathcal{R}$ (e.g., via mean absolute contribution $\mathbb{E}_{X \in \mathcal{R}}[|\hat{m}^{(\ell)}(X)|]$ or similar), then subselect those stages and fit a parsimonious model $\hat{m}_{\mathcal{L}}$ that retains only the stages that matter for $\mathcal{R}$. This way we can achieve parsimony for that region without retraining.

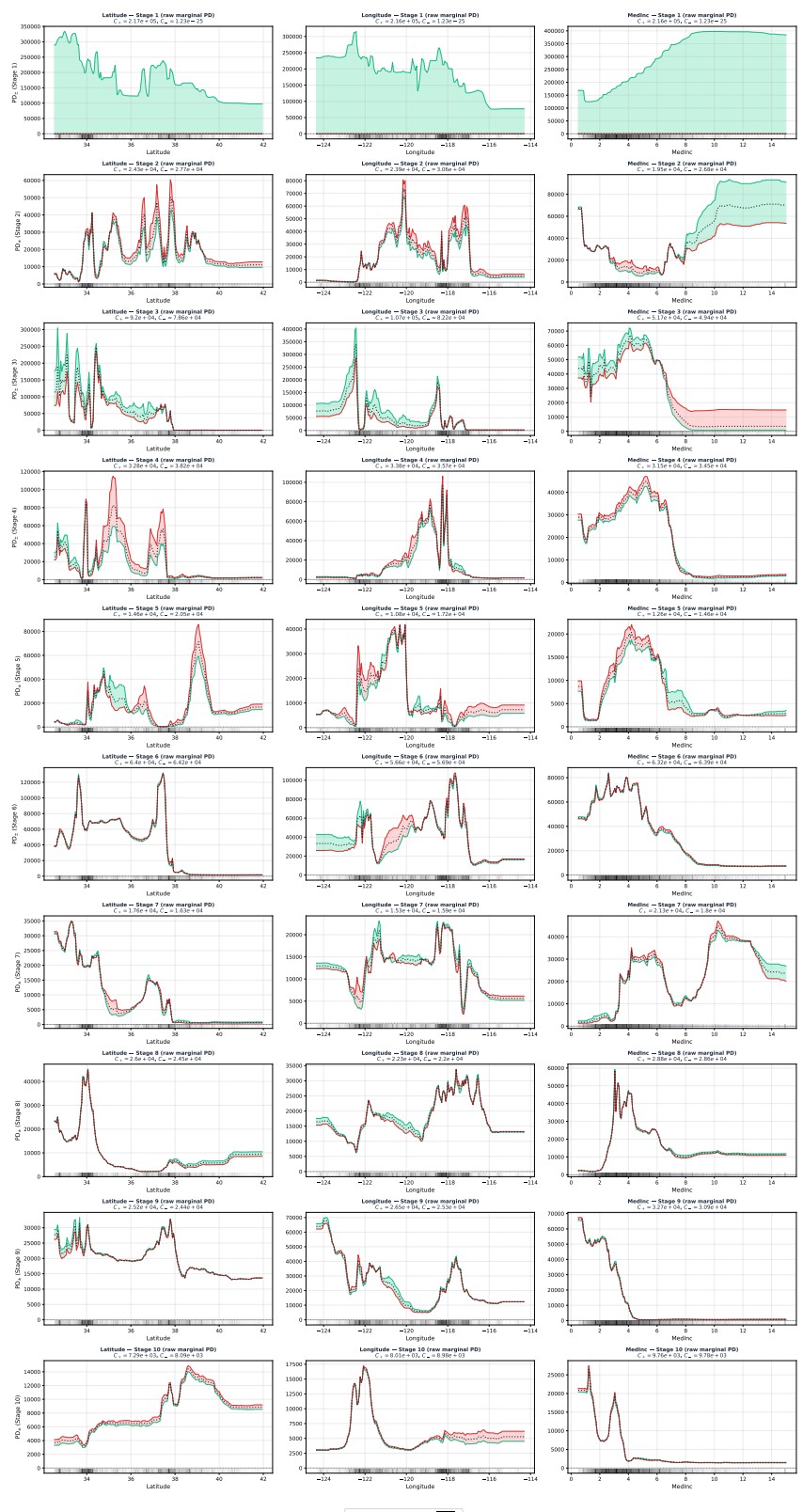

*Figure 6.* Partial dependence functions for the positive and negative products $\hat{m}_+$ and $\hat{m}_-$ over different stages for California housing: TSL with $R \leq 10$ stages (see next figure for $R \leq 2$). Features shown are the top three ranked by feature importance: latitude, longitude, and median income. The two curves in each panel are the 1D partial dependence of $\hat{m}_+$ and $\hat{m}_-$ on that feature; the shaded region between the curves represents the signed partial dependence along that axis (green positive, red negative), i.e., the net stage contribution after the ordered difference $\hat{m}_+ - \hat{m}_-$ (main text Theorem 5.1 and Figure 3). Panel titles report the per-feature, per-stage scaling constants $C_{+,j}^{(\ell)}$ and $C_{-,j}^{(\ell)}$ defined in Theorem 5.1.

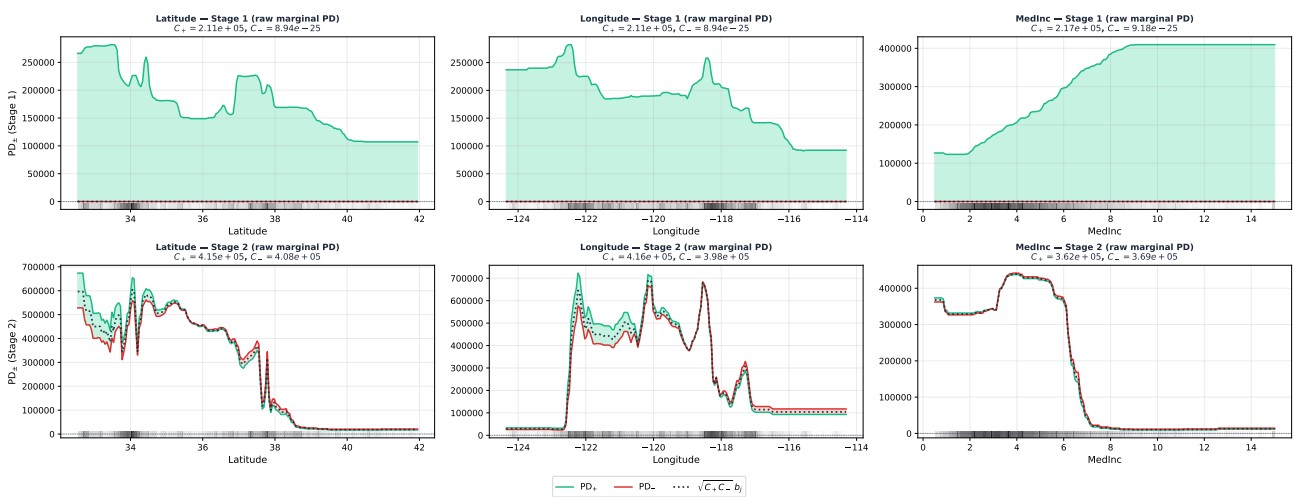

*Figure 7.* Partial dependence functions for $\hat{m}_+$ and $\hat{m}_-$ from TSL with $R \leq 2$ stages. Features shown are the top three ranked by feature importance: latitude, longitude, and median income (same order as the previous figure); fewer stages yield a more parsimonious decomposition. Shaded region: signed partial dependence (green positive, red negative). Panel titles report the per-feature, per-stage scaling constants $C_{+,j}^{(\ell)}$ and $C_{-,j}^{(\ell)}$ defined in Theorem 5.1.

**What feature importance is computing.** Let $\hat{P}_X$ denote the empirical distribution of the covariates (e.g., training data). For each stage $\ell \in [R]$ and feature $j \in [p]$, the fitted TSL model yields the univariate backbone $b_j^{(\ell)}(x_j)$ and tilt $d_j^{(\ell)}(x_j)$ (main text Equation (3)). Per-stage feature importance is the empirical variance of these quantities over $\hat{P}_X$: $I_{j,\ell}^b := \widehat{\mathrm{Var}}_{X \sim \hat{P}_X}(b_j^{(\ell)}(X_j))$ (backbone) and $I_{j,\ell}^d := \widehat{\mathrm{Var}}_{X \sim \hat{P}_X}(d_j^{(\ell)}(X_j))$ (tilt). Recall the stage-$\ell$ contribution $\hat{m}^{(\ell)}(\mathbf{x}) = \hat{m}_+^{(\ell)}(\mathbf{x}) - \hat{m}_-^{(\ell)}(\mathbf{x})$ from main eq. (2). The stage weights are $w_\ell := \mathbb{E}_{X \sim \hat{P}_X}[\hat{m}^{(\ell)}(X)^2] / \sum_{k=1}^{R} \mathbb{E}_{X \sim \hat{P}_X}[\hat{m}^{(k)}(X)^2]$, so that $w_\ell \geq 0$ and $\sum_{\ell=1}^{R} w_\ell = 1$. Aggregated importance is $I_j^b := \sum_{\ell=1}^{R} w_\ell I_{j,\ell}^b$ and $I_j^d := \sum_{\ell=1}^{R} w_\ell I_{j,\ell}^d$. The combined bar plot shows $I_j := I_j^b + \gamma I_j^d$ with $\gamma = 1$.

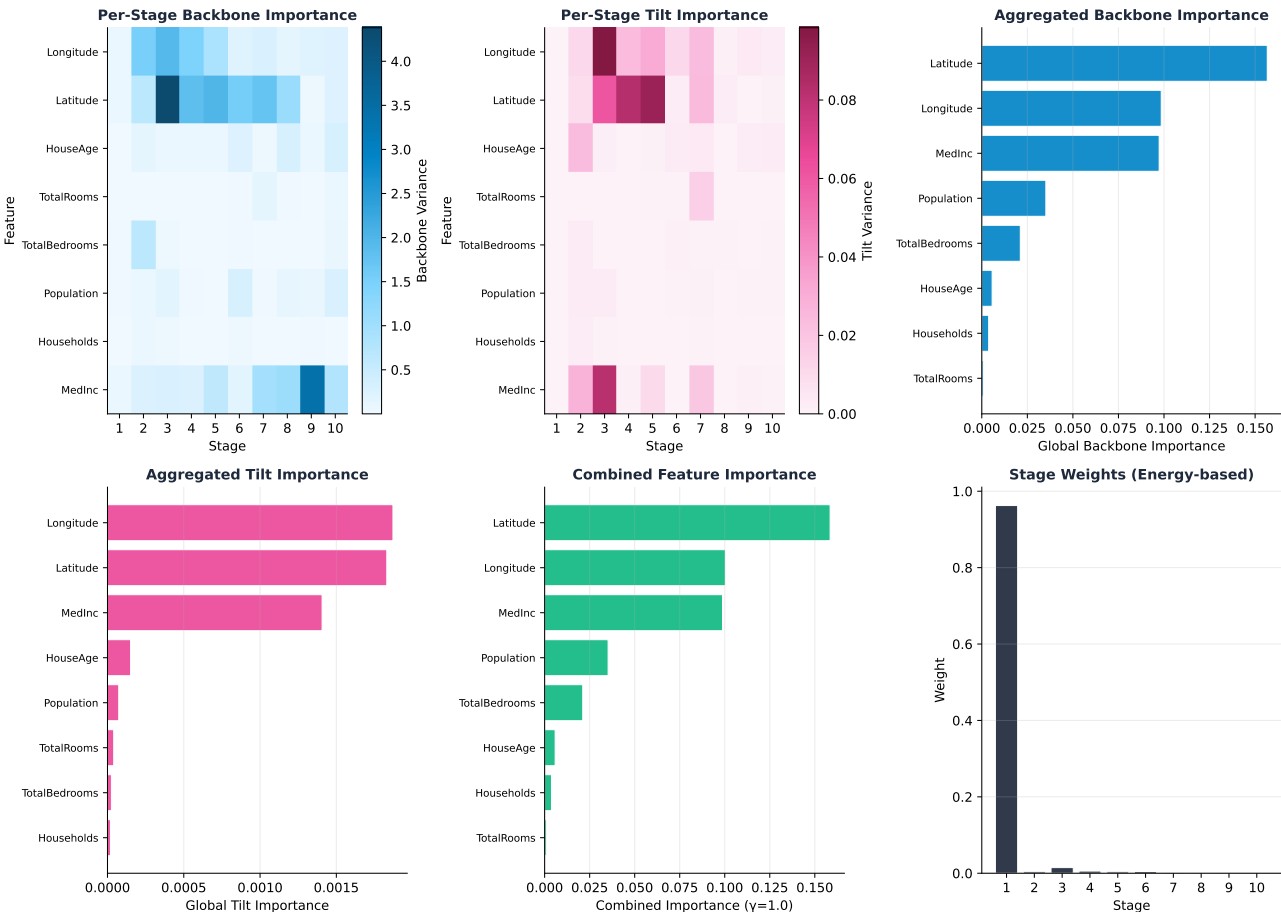

*Figure 8.* Feature importance for California housing (black-box TSL, $R \leq 10$ stages). **Row 1:** (1) Per-stage backbone importance: heatmap of $I_{j,\ell}^b$ (rows = features, columns = stages); color scale is backbone variance. (2) Per-stage tilt importance: heatmap of $I_{j,\ell}^d$ (same layout); color scale is tilt variance. (3) Aggregated backbone importance: horizontal bar plot of $I_j^b$ per feature, sorted descending. **Row 2:** (4) Aggregated tilt importance: horizontal bar plot of $I_j^d$ per feature, sorted descending. (5) Combined importance: horizontal bar plot of $I_j = I_j^b + \gamma I_j^d$ ($\gamma = 1$) per feature, sorted descending. (6) Stage weights: bar plot of $w_\ell$ by stage ($\sum_\ell w_\ell = 1$).

## I.2. Bike Sharing: Additional Figures

We use the Bike Sharing dataset (Fanaee-T, 2013) as an interpretability case study; the figures below are the primary contribution and the RMSE comparison is illustrative only. We tuned TSL ($R \leq 2$) and EBM on bike sharing the same way as for the other experiments: with Optuna and 200 iterations, using the hyperparameter search spaces in Table 2 (TSL $R \leq 2$) and Table 9 (EBM). Best test RMSE on a single 80/20 split: EBM 55.36, TSL 55.19. We do not run SepALS or the TSL (1-product) ablation here.

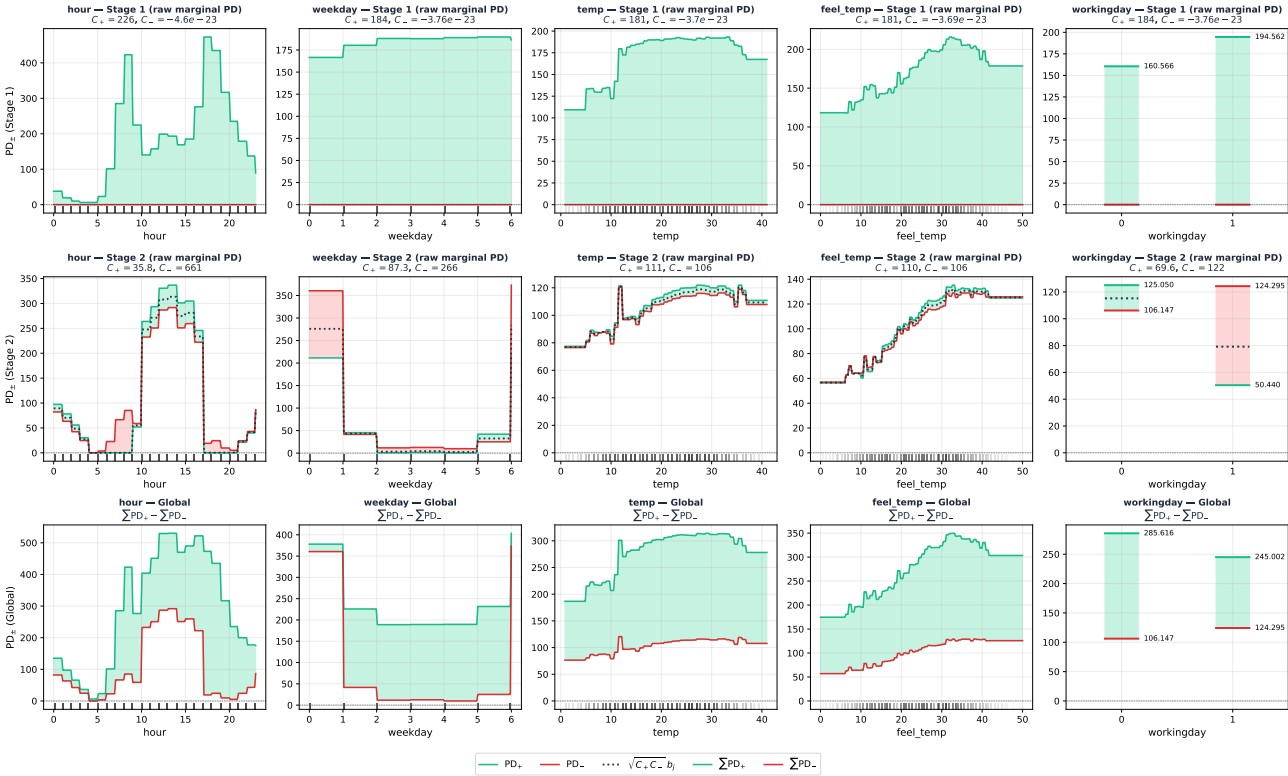

*Figure 9.* Partial dependence functions for the positive and negative products $\hat{m}_+$ and $\hat{m}_-$ for Bike Sharing: TSL with $R \leq 2$ stages. The two curves in each panel are the 1D partial dependence of $\hat{m}_+$ and $\hat{m}_-$ on that feature; the shaded region between the curves represents the signed partial dependence along that axis (green positive, red negative), i.e., the net stage contribution after the ordered difference $\hat{m}_+ - \hat{m}_-$ (main text Theorem 5.1 and Figure 3). Panel titles report the per-feature, per-stage scaling constants $C_{+,j}^{(\ell)}$ and $C_{-,j}^{(\ell)}$ defined in Theorem 5.1; the dotted backbone curve is $\sqrt{C_{+,j}^{(\ell)} C_{-,j}^{(\ell)}} \cdot b_j^{(\ell)}$.

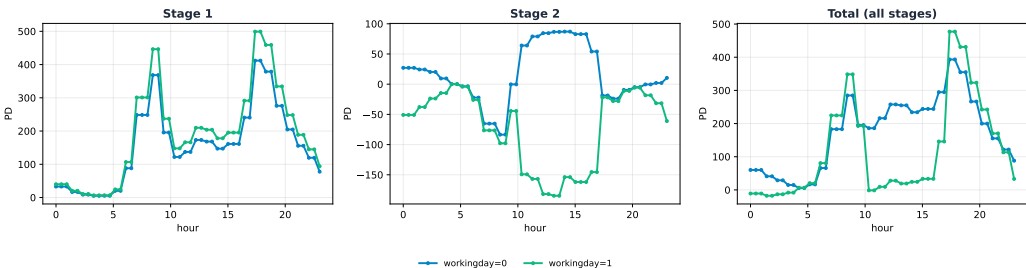

*(a)* TSL: 2D partial dependence by stage. Panels show $\text{PD}^{(1)}_{\text{hour,workingday}}$ for stage 1, $\text{PD}^{(2)}_{\text{hour,workingday}}$ for stage 2, and their sum $\text{PD}_{\text{hour,workingday}}$; in each, two curves give partial dependence versus hour for workingday = 0 (non-weekday) and workingday = 1 (weekday).

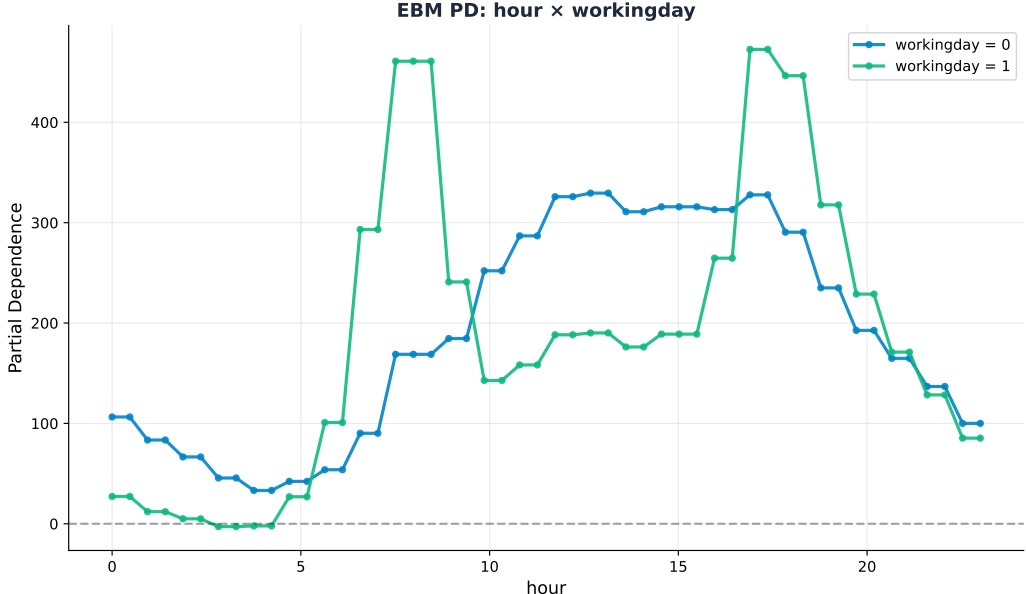

*(b)* EBM: $\text{PD}_{\text{hour,workingday}}$. Two curves show partial dependence versus hour for workingday = 0 (non-weekday) and workingday = 1 (weekday).

*Figure 10.* 2D partial dependence $\text{PD}_{\text{hour,workingday}}$ for the (hour, workingday) interaction on Bike Sharing: contribution to predicted demand versus hour of day, by weekday vs. non-weekday. For TSL, $\text{PD}_{\text{hour,workingday}} = \text{PD}^{(1)}_{\text{hour,workingday}} + \text{PD}^{(2)}_{\text{hour,workingday}}$, where $\text{PD}^{(\ell)}_{\text{hour,workingday}}$ is the 2D partial dependence function of stage $\ell$. In (a), stage 1 shows a similar profile for weekday and weekend (little separation). Stage 2 corrects by shifting mass toward weekend midday and away from weekday midday, sharpening the weekday commute peaks and raising the weekend midday bulge. This correction is only partial: in the final sum (rightmost panel), the weekend curve still exhibits morning and late-afternoon peaks, whereas in (b) the EBM weekend profile is flatter in those hours, with demand concentrated more clearly in midday.

## I.3. Synthetic Masked Interaction: Additional Figures

### I.3.1. ICE CURVES

Individual Conditional Expectation (ICE) curves are a standard visualization tool: for each sample $i$, one plots the slice $x_1 \mapsto \hat{f}(x_1, x^{(i)}_{(-1)})$ as $x_1$ varies while holding other features fixed at that sample's values. ICE can reveal heterogeneity that partial dependence averages away (e.g., quadratic curves with signed amplitudes whose mean is zero), but it is purely a post-hoc visualization—it does not expose the model's structure. For any fitted predictor (TSL, EBM, or XGBoost), ICE applies the same recipe and returns a family of curves; it does not indicate whether the underlying function is additive, multiplicative, or how the interaction has been masked. Turning ICE into a single interpretable summary requires introducing and justifying an additional aggregation (e.g., quantiles, variance bands, or magnitude functionals), which is exactly what TSL's backbone provides naturally via its positive factorization. Figure 11 shows ICE curves for $x_1$ across all three models.

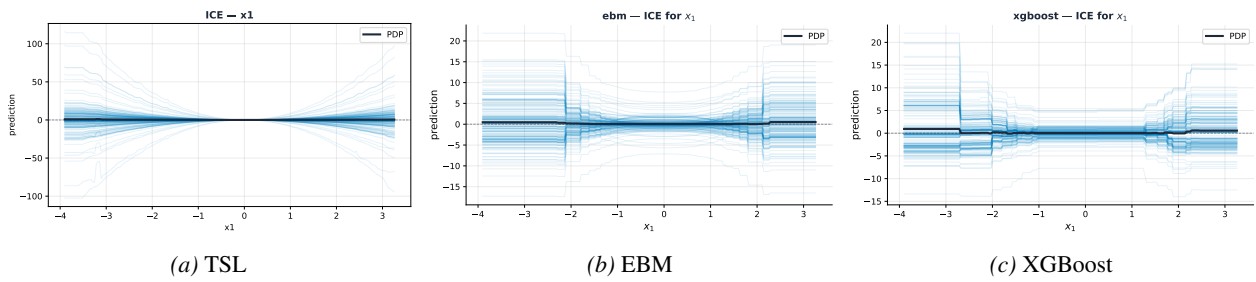

*(a)* TSL      *(b)* EBM      *(c)* XGBoost

*Figure 11.* ICE curves for $x_1$ across all models, revealing heterogeneous quadratic patterns with signed amplitudes whose mean is zero. The wide spread of ICE curves with near-zero partial dependence overlay demonstrates the "spaghetti plot" problem: ICE reveals hidden signal but introduces interpretability challenges.

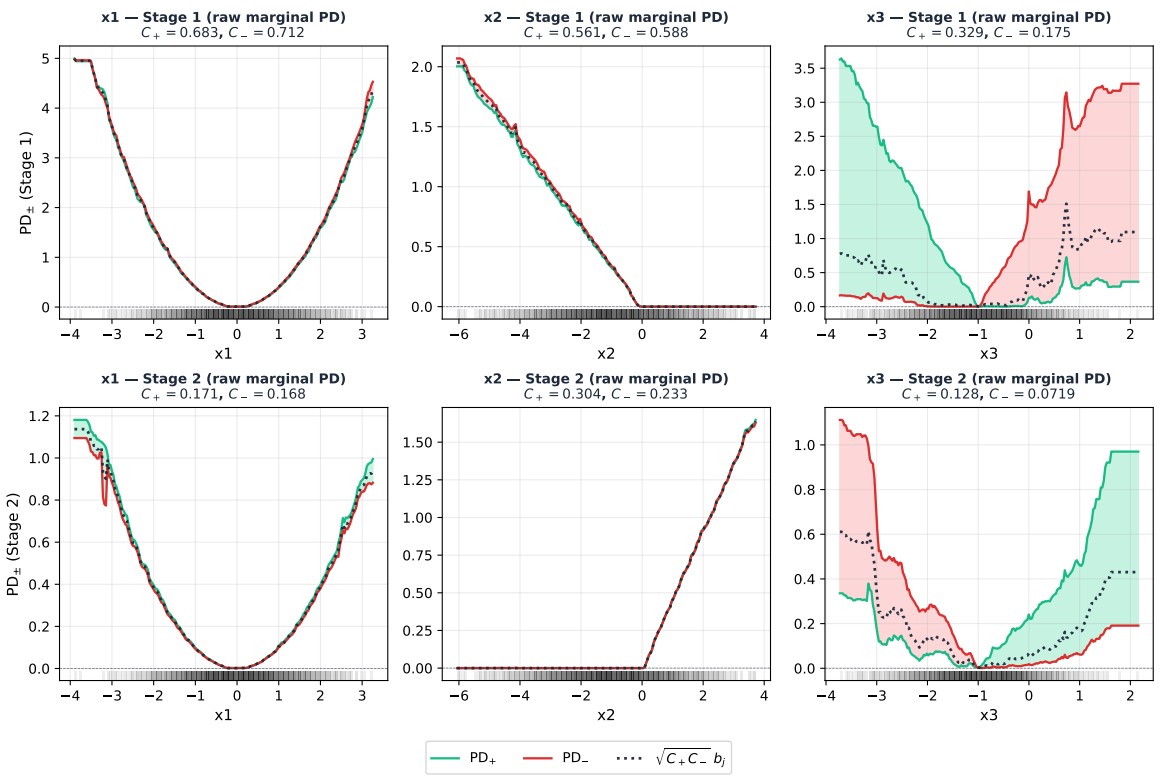

*Figure 12.* First-order partial dependence (TSL, synthetic setup): stage 1 (top row) and stage 2 (bottom row) for $x_1$, $x_2$, $x_3$. Each panel shows $\mathrm{PD}^{(\ell)}_{+,j}$ and $\mathrm{PD}^{(\ell)}_{-,j}$; panel titles report the per-feature, per-stage scaling constants $C^{(\ell)}_{+,j}$ and $C^{(\ell)}_{-,j}$ defined in Theorem 5.1 (main text).

### I.3.2. SepALS Baseline and Sign Non-Identifiability

For completeness we fit the SepALS baseline (Section J) on the same synthetic problem under the same Optuna protocol (200 trials, 10-fold CV, search space of Table 4 with $r_{\max} = 2$ and the `tent` basis excluded so only `monomial` is used). Cross-validation selects rank $r = 1$ with monomial degree 2; the fitted model writes, using the notation of (1), as $\hat{m}_{\mathrm{SepALS}}(\mathbf{x}) = s_1\, g_1^{(1)}(x_1)\, g_2^{(1)}(x_2)\, g_3^{(1)}(x_3)$ with $s_1 \approx -11.84$. Figure 13 plots the fitted univariate factors $g_j^{(1)}(x_j)$, one per feature.

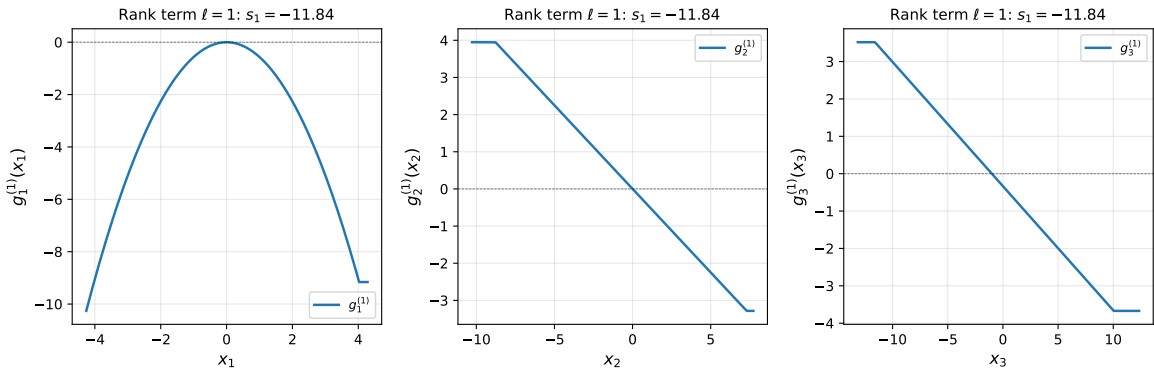

*Figure 13.* SepALS fitted separated factors $g_j^{(1)}(x_j)$ on the synthetic masked-interaction problem ($r = 1$, monomial basis, degree 2, $s_1 \approx -11.84$; test RMSE 0.2612, close to the irreducible noise 0.25). Despite SepALS being an unconstrained separable estimator, the rank-1 fit is interpretable here because the true function is itself exactly rank-1: each factor recovers a univariate component of $x_1^2 x_2(1 + x_3)$ up to a multiplicative constant absorbed in $s_1$. Reading off the panels gives $g_1^{(1)}(x_1) \approx -0.57\, x_1^2$ (downward parabola, zero at $x_1 = 0$), $g_2^{(1)}(x_2) \approx -0.45\, x_2$ (linear, zero at $x_2 = 0$), and $g_3^{(1)}(x_3) \approx -0.33\,(1 + x_3)$ (linear, zero at $x_3 = -1$); the product $s_1 \prod_j g_j^{(1)}(x_j) \approx (-11.84)(-0.57)(-0.45)(-0.33)\, x_1^2 x_2(1 + x_3) \approx x_1^2 x_2(1 + x_3)$ recovers the data-generating expression, with the four minus signs from $s_1$, $g_1^{(1)}$, $g_2^{(1)}$, and $g_3^{(1)}$ cancelling out.

SepALS recovers an equivalent rank-1 representation of the data-generating function only up to the gauge invariances of the model in (89): the prediction is unchanged under any rescaling $(g_j^{(\ell)}, s_\ell) \mapsto (\alpha\, g_j^{(\ell)}, s_\ell/\alpha)$ for $\alpha \in \mathbb{R} \setminus \{0\}$ and, in particular, under any pairwise sign flip $(g_j^{(\ell)}, g_{j'}^{(\ell)}) \mapsto (-g_j^{(\ell)}, -g_{j'}^{(\ell)})$ for $j \neq j'$. Consequently, the sign of an individual factor $g_j^{(\ell)}$ is not identified; only the joint sign $\mathrm{sign}\big(s_\ell \prod_j g_j^{(\ell)}(x_j)\big)$ is. For the rank-1 fit of Figure 13, this means that the panel $g_2^{(1)}(x_2)$ alone is uninformative about the sign of the prediction: even though $g_2^{(1)}(x_2) > 0$ for $x_2 \leq 0$ in the plotted gauge, a sign flip of two among $\{s_1, g_1^{(1)}, g_2^{(1)}, g_3^{(1)}\}$ yields an observationally equivalent model in which $g_2^{(1)}(x_2) < 0$ on the same region. Reading off the effect of $x_2$ in isolation is therefore impossible.

### I.3.3. DIRECTIONALITY IN TSL: BACKBONE AND TILT

TSL avoids this ambiguity by representing the stage as the ordered difference of two positive products (Theorem 5.1 and Equation (2)) and reading directionality through the backbone–tilt parametrization of Equation (3). From the PD identities of Theorem F.1, the backbone is recovered pointwise as $b_j^{(\ell)}(x_j) = \sqrt{\mathrm{PD}_{+,j}^{(\ell)}(x_j)\,\mathrm{PD}_{-,j}^{(\ell)}(x_j)}\big/\sqrt{C_{+,j}^{(\ell)} C_{-,j}^{(\ell)}}$, while the tilt is

$$d_j^{(\ell)}(x_j) \;=\; \tfrac{1}{2}\log\frac{\mathrm{PD}_{+,j}^{(\ell)}(x_j)}{\mathrm{PD}_{-,j}^{(\ell)}(x_j)} \;+\; \tfrac{1}{2}\log\frac{C_{-,j}^{(\ell)}}{C_{+,j}^{(\ell)}}. \tag{86}$$

The second, $x_j$-independent term reflects a gauge in the stage decomposition: rescaling $\hat{m}_{\pm,j}^{(\ell)} \mapsto \alpha_{\pm,j}\hat{m}_{\pm,j}^{(\ell)}$ together with $\lambda_{\pm}^{(\ell)} \mapsto \lambda_{\pm}^{(\ell)}/\prod_j \alpha_{\pm,j}$ leaves the stage prediction unchanged but shifts $d_j^{(\ell)}$ by $\tfrac{1}{2}\log(\alpha_{+,j}/\alpha_{-,j})$. Only the *variation* of $d_j^{(\ell)}$ across $x_j$ is gauge-invariant; any constant offset in $d_j^{(\ell)}$ can be absorbed into a corresponding rescaling of $\lambda_{\pm}^{(\ell)}$. The gauge-invariant diagnostic for whether coordinate $j$ carries directional information at stage $\ell$ is therefore not "$d_j^{(\ell)} \approx 0$" but "$d_j^{(\ell)}$ approximately constant in $x_j$," which from (86) is equivalent to $\mathrm{PD}_{+,j}^{(\ell)}/\mathrm{PD}_{-,j}^{(\ell)}$ being approximately constant in $x_j$, i.e. the two PD curves being approximately *proportional* as functions of $x_j$. This gauge is internal to the TSL stage parametrization and is distinct from the sign-flip gauge of SepALS in (89): a constant tilt offset can be harmlessly absorbed into $\lambda_{\pm}^{(\ell)}$, whereas a SepALS sign flip propagates across factors and cannot be fixed by positivity.

Before turning to the figure, we note that a negative value of $d_j^{(\ell)}(x_j)$ at a point does *not* by itself indicate that the variable contributes negatively to the stage's prediction there. There are two reasons. First, the zero point of $d_j^{(\ell)}$ is not identifiable: by the gauge above, a constant can be shifted between $d_j^{(\ell)}$ and the tilt intercept $d_0^{(\ell)}$ (equivalently, rescaled into $\lambda_{\pm}^{(\ell)}$) without changing any prediction, so the sign of $d_j^{(\ell)}$ at a point carries no absolute meaning. Second, even with the gauge fixed, the

stage factorizes as

$$\hat{m}^{(\ell)}(\mathbf{x}) = b^{(\ell)}(\mathbf{x})\big(e^{d^{(\ell)}(\mathbf{x})} - e^{-d^{(\ell)}(\mathbf{x})}\big) = 2\,b^{(\ell)}(\mathbf{x})\,\sinh\big(d^{(\ell)}(\mathbf{x})\big), \qquad d^{(\ell)}(\mathbf{x}) = d_0^{(\ell)} + \sum_j d_j^{(\ell)}(x_j), \quad (87)$$

so the sign of the stage at $\mathbf{x}$ is the sign of the total tilt $d^{(\ell)}(\mathbf{x})$ (since $b^{(\ell)} > 0$ and $\sinh$ is odd and increasing): it flips when $d^{(\ell)}(\mathbf{x})$ crosses zero—equivalently, when the accumulated per-coordinate tilt $\sum_j d_j^{(\ell)}(x_j)$ crosses $-d_0^{(\ell)} = \frac{1}{2}\log(\lambda_-^{(\ell)}/\lambda_+^{(\ell)})$—not when any individual $d_j^{(\ell)}(x_j)$ crosses zero. A negative $d_j^{(\ell)}(x_j)$ lowers $d^{(\ell)}$, but whether that nudges the stage past zero depends on the other coordinates' tilts and on the intercept $d_0^{(\ell)}$ (equivalently $\lambda_\pm^{(\ell)}$). Per-coordinate tilt curves therefore localize *where* sign-bearing variation lives (a gauge-invariant property), not the sign of the stage at any particular input.

### I.3.4. READING THE STAGE-1 FACTORIZATION OFF THE FIGURE

In stage 1 of Figure 12 this proportionality holds for $j = 1$ and $j = 2$ but not for $j = 3$: the $x_1$ and $x_2$ panels show $\mathrm{PD}_{+,j}^{(1)}$ and $\mathrm{PD}_{-,j}^{(1)}$ tracking each other up to a uniform scale (and visually coinciding, since the panel titles report $C_{+,j}^{(1)} \approx C_{-,j}^{(1)}$ for $j = 1, 2$, which pins the proportionality constant near 1), while on $x_3$ the curves cross at $x_3 = -1$ and are not proportional in any neighborhood. Equivalently, $d_1^{(1)}$ and $d_2^{(1)}$ are approximately constant in their respective coordinates, while $d_3^{(1)}$ carries the only sign-bearing variation. Absorbing the constant tilt levels of $d_1^{(1)}, d_2^{(1)}$ into a redefinition of $\lambda_\pm^{(1)}$ (which is exact, by the gauge above) lets us take $\hat{m}_{+,j}^{(1)} \approx \hat{m}_{-,j}^{(1)} =: b_j^{(1)}$ for $j = 1, 2$, and stage 1 factorizes along the sign-bearing coordinate:

$$\hat{m}^{(1)}(\mathbf{x}) \approx b_1^{(1)}(x_1)\,b_2^{(1)}(x_2)\left[\lambda_+^{(1)}\,\hat{m}_{+,3}^{(1)}(x_3) - \lambda_-^{(1)}\,\hat{m}_{-,3}^{(1)}(x_3)\right], \qquad (88)$$

where the bracketed quantity is the only signed object and is exactly the green/red shaded gap in the rightmost stage-1 panel of Figure 12. The model-level statement readable directly off the figure is therefore: *$x_1$ and $x_2$ enter the first stage only through positive magnitude factors $b_1^{(1)}, b_2^{(1)}$, and the sign of stage 1 is determined entirely by $x_3$, with the sign flip located at $x_3 = -1$.* The SepALS panels of Figure 13 admit no analogous reading because the gauge invariances of (89) prevent attributing magnitude or direction to any individual $g_j^{(\ell)}$. We therefore argue that TSL is interpretable in this example: its positivity-constrained ordered-difference parametrization (2) fixes the sign gauge so that each per-feature curve $\hat{m}_{\pm,j}^{(\ell)}$ has a uniquely-defined, positive meaning, the backbone $b_j^{(\ell)}$ summarises magnitude, the tilt $d_j^{(\ell)}$ summarises direction, and the only signed objects are the named scalars $\lambda_\pm^{(\ell)}$ and the per-feature signed PD $\mathrm{PD}_j^{(\ell)}$ visualised as the shaded gap. The reader can therefore read interpretation statements such as "$x_1, x_2$ scale; $x_3$ directs, with the flip at $x_3 = -1$" directly off Figure 12.

### I.4. Bimodal Bagged Representations: Illustrating Alignment and Filtering

This subsection illustrates the bagged-grid aggregation pipeline of Section D.1 (Algorithm 11) on a controlled example, making concrete why the normalization, anchoring, and similarity-filtering steps of Section 4.2.2 are needed. We generate $n = 5000$ observations from the data-generating process

$$f(x_1, x_2) = \exp\big(\sin(x_1)\cos(x_2)\big) + x_1, \qquad x_1, x_2 \overset{\text{iid}}{\sim} \text{Uniform}[-4, 4],$$

and fit a single TSL stage with $n_{\text{grids}} = 389$ bagged grids using the hyperparameters tuned for this problem. Figure 14 shows the resulting stage-1 backbone curves $b_1^{(1)}(x_1)$ and $b_2^{(1)}(x_2)$ across the bagged grids, organized in three rows to expose the bimodality, the role of the reference grid, and the population view.

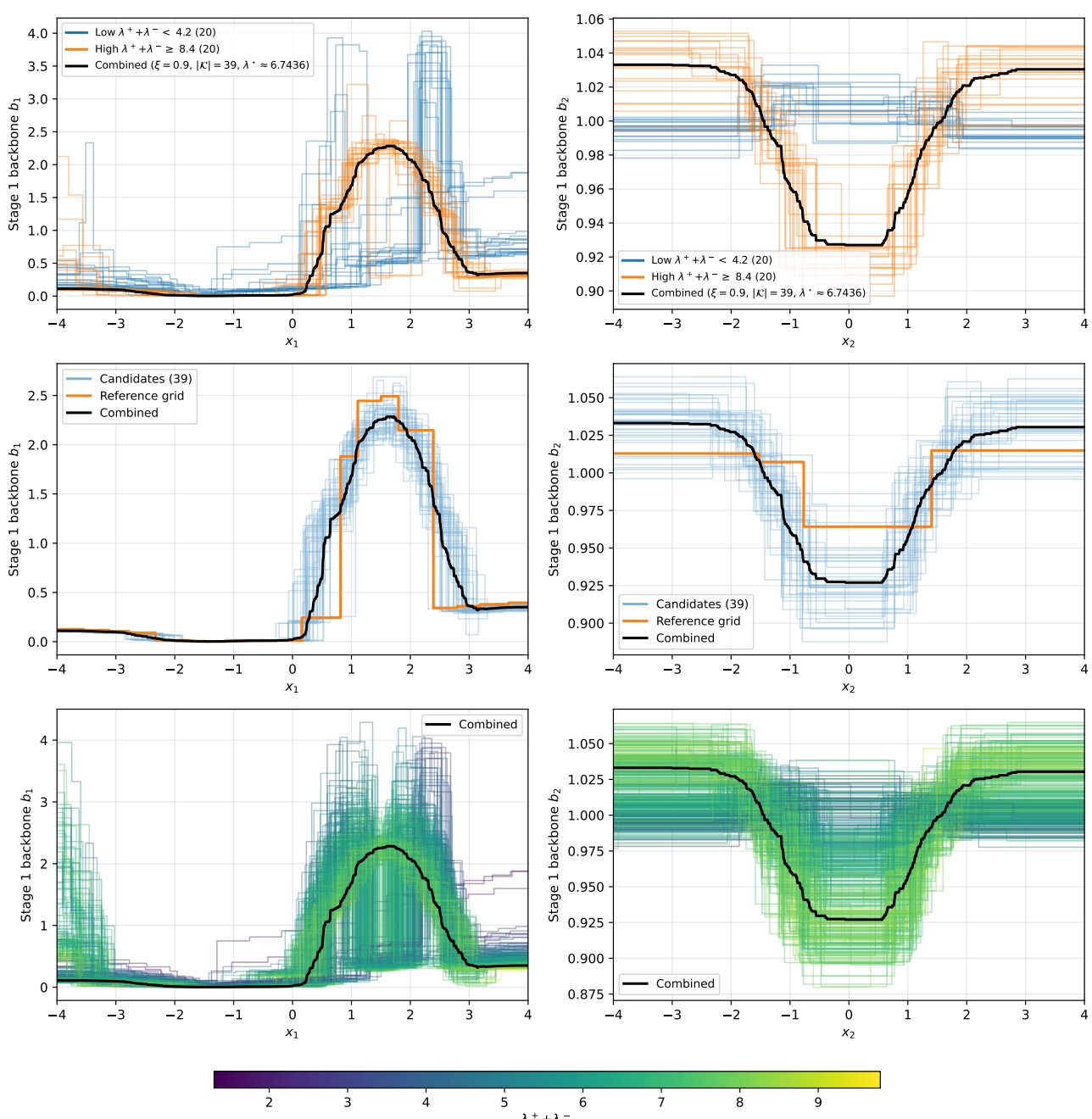

*Figure 14.* Bimodal bagged representations on $f(x_1, x_2) = \exp(\sin(x_1)\cos(x_2)) + x_1$ ($n = 5000$, $n_{\text{grids}} = 389$, trimming parameter $\xi = 0.9$); $\lambda_{\pm}$ denote the stage scales $\lambda_{\pm}^{(\ell)}$ from the two-tensor parametrization $\lambda_+ \prod_j \hat{m}_{+,j} - \lambda_- \prod_j \hat{m}_{-,j}$ of (2). Columns: $b_1^{(1)}(x_1)$ (left), $b_2^{(1)}(x_2)$ (right). **Row 1 (bimodality):** the 20 grids with smallest total scale $\lambda_+ + \lambda_- < 4.2$ (blue) and the 20 with largest $\lambda_+ + \lambda_- \geq 8.4$ (orange). Depending on the random seed, the bagged grids converge to one of two qualitatively distinct backbone representations of the *same* fitted function: the high-$\lambda$ group concentrates the $x_1$ bump near $x_1 \in [1, 2.5]$, while the low-$\lambda$ group distributes the signal differently and the $x_2$ curve picks up a reversed orientation in part of the range. The black curve is the combined backbone produced by Algorithm 11. **Row 2 (selection):** the kept set $\mathcal{K}$ of $\lceil (1 - \xi)n_{\text{grids}} \rceil = 39$ top-ranked candidates by combined cosine similarity to the reference grid (closest to the $(\lambda_+, \lambda_-)$ centroid, orange); the competing representation is filtered out, the kept candidates are all close to the same shape, and the combined backbone (black) is a stable average. **Row 3 (population):** all 389 candidates, colored continuously by $\lambda_+ + \lambda_-$, confirming that the two converged representations populate opposite ends of the $\lambda$ spectrum.

**Why naive averaging fails.** Row 1 shows that independently bagged grids do not all converge to a single representation.

The two product factors $b_1^{(1)}, b_2^{(1)}$ admit multiple componentwise allocations of the same fitted stage: at large $\lambda_+ + \lambda_-$ the $x_1$ factor carries a sharp bump and $x_2$ is nearly flat, while at small $\lambda_+ + \lambda_-$ the mass is redistributed across coordinates and the $x_2$ curve picks up a reversed orientation in part of the range. Both branches fit the data comparably well at the stage level—the predictions $\lambda_+^{(1)} \prod_j b_j^{(1)} - \lambda_-^{(1)} \prod_j b_j^{(1)}$ are similar—but the per-feature backbones differ substantially. Componentwise averaging is not component-aware: it cannot account for the fact that each branch compensates differently across features, and averaging incompatible representations produces a curve that minimizes neither the residual nor the per-feature shape, destroying the interpretable structure TSL is designed to provide.

**How similarity filtering resolves the issue.** Row 2 shows the output of Algorithm 11: the reference grid $\mathcal{G}^\star$ (orange) is picked as the $(\lambda_+, \lambda_-)$-centroid medoid, and only the top $\lceil (1 - \xi) n_{\text{grids}} \rceil = 39$ candidates ranked by the combined cosine similarity of (23) to the reference are retained. Restricting the average to $\mathcal{K}$ yields a coherent set whose shapes all agree with $\mathcal{G}^\star$, so the combined backbone (black) is a sharp and stable estimate. The competing representation from row 1 is filtered out entirely. The trimming parameter $\xi$ is tuned by cross-validation alongside the other hyperparameters.

**Population view.** Row 3 plots all 389 candidates colored continuously by $\lambda_+ + \lambda_-$. The color gradient separates by $\lambda$, confirming that the two converged backbone shapes populate opposite ends of the $\lambda$ spectrum: this is the non-rectangular-support / multiplicative-symmetry phenomenon discussed in Section 4.2.3 made visible at the level of fitted grids.

**Connection to the limitations.** The non-uniqueness exhibited in row 1 is the source of TSL's principal current limitation: because the bagged grids may converge to different separable representations of the same function, the similarity filter must discard grids not aligned with $\mathcal{G}^\star$, so only a fraction of the fitted bags contribute to the final stage. Developing an aggregation method that uses all bags—e.g. via Riemannian optimization on the positive rank-1 manifold (Liu & Boumal, 2020)—is a natural next step.

## J. SepALS Baseline

As a separable-model baseline we implement *SepALS*, a fixed-rank separable alternating least-squares estimator following the multivariate regression formulation of Beylkin et al. (2009) (see also Beylkin & Mohlenkamp (2005); Garcke (2010)). SepALS fits a sum of unconstrained separable rank-1 products

$$\hat{m}_{\text{SepALS}}(\mathbf{x}) \;=\; \sum_{\ell=1}^{r} s_\ell \prod_{j=1}^{p} g_j^{(\ell)}(x_j), \tag{89}$$

where $r$ is the *separation rank* in the sense of Beylkin et al. (2009) (i.e., the total number of separable products in the sum), each univariate factor $g_j^{(\ell)}$ is expanded in a fixed univariate basis (tent functions or monomials in our implementation) of configurable degree, and the stage scales $s_\ell \in \mathbb{R}$ are unconstrained. The model class in (89) coincides with the separable model of (1) and is exactly the setting studied by Beylkin et al. (2009): there are no positivity, ordered-difference, or stagewise constraints, and all $rp$ univariate factors and $r$ scales are jointly optimized. Note that the separation rank $r$ used here for SepALS is distinct from the TSL stage count $R$ used in the main paper: because each TSL stage contributes two separable products $(\hat{m}_+^{(\ell)} - \hat{m}_-^{(\ell)})$, a TSL run with $R$ stages has total separation rank at most $2R$ in the sense of (89) (with $2R-$ number of positive-only stages, since a positive-only stage contributes one product not two).

Following Beylkin et al. (2009), fitting proceeds by alternating least-squares (ALS) over the factor matrices: at each sweep, one mode $j$ is updated while the others are held fixed, reducing the problem to a linear least-squares system in the basis coefficients of $\{g_j^{(\ell)}\}_{\ell=1}^{r}$ with a Tikhonov (ridge) penalty and an optional smoothness penalty on the basis coefficients to regularize ill-conditioned modes. Stage scales $s_\ell$ are refit after every sweep. Multiple random initializations are used and the run with the lowest training residual is retained. The full hyperparameter search space (basis, degree, rank, ridge, smoothness, sweep budget, number of restarts) is reported in Table 4; we use the same Optuna protocol as for the other baselines.

We use SepALS in two complexity regimes, mirroring our TSL conditions: **SepALS** ($r \leq 2$), with rank sampled from $\{1, 2\}$, and **SepALS** ($r \leq 10$), with rank sampled from $\{1, \ldots, 10\}$. These two settings let us compare TSL against the closest unconstrained separable baseline at matched rank budgets.

# K. Hyperparameter Search Spaces

All models were tuned via Optuna's TPE sampler with 200 trials; each configuration was evaluated by 10-fold cross-validation on the training split. Below we report the hyperparameter search space for each model. For TSL, both variants are interpretable: TSL with at most 2 stages ($R \leq 2$) uses the search space in Table 2, and TSL with $R \leq 10$ stages uses the search space in Table 3; throughout, $R$ denotes the number of TSL stages, distinct from the SepALS separation rank $r$. The TSL (1-product) ablation uses the same TSL ($R \leq 2$) search space (Table 2) with two flags disabled—positivity (component positivity constraint dropped) and the ordered difference (one product per stage, $\hat{m}^{(\ell)} = \hat{m}_+^{(\ell)}$)—and $R$ extended to $\leq 4$ so the total separation rank matches SepALS at $r \leq 4$. SepALS (described in detail in Section J) uses the search space in Table 4, with the rank cap set to either $r \leq 2$ or $r \leq 10$. Ranges are given as distribution type and bounds; categorical choices are listed explicitly; fixed values are shown as "fixed."

*Table 2.* TSL ($R \leq 2$): hyperparameter search space. Implementation names and paper notation (where used) are given; $R$ is the number of TSL stages (each stage fits an ordered-difference pair $\hat{m}_+^{(\ell)} - \hat{m}_-^{(\ell)}$, so the total separation rank is at most $2R$), and $n_{\text{grids}}$ the number of bagged grid learners per stage.

| Hyperparameter | Description | Search space |
|---|---|---|
| $\alpha$ | Ridge regularization in the $2 \times 2$ least-squares system for split scoring (Alg. 1). | log-uniform($10^{-6}$, 1) |
| colsample_bytree | Fraction of features sampled when constructing each grid tensor (column subsampling). | uniform(0.3, 1) |
| decay | Decay rate for the number of splits per stage (later stages get fewer splits when $< 1$). | uniform(0.2, 1) |
| epochs ($R$) | Number of TSL stages. | $\{1, 2\}$ |
| min_interval_samples | Minimum observations per resulting subinterval; only split positions with at least this many on both sides are valid. If no valid positions exist, no split is proposed for that step. | integer in [1, 100) |
| n_iter | Max. grid refinement iterations (splits) per single grid tensor. | integer in [10, 251) |
| refinement_strategy | Loss for split scoring: L2 (squared error) or Huber. | l2, huber |
| similarity_threshold | Threshold for merging similar components when combining bagged grids. | uniform(0, 1) |
| split_try | Number of candidate split points tried per feature at each refinement step. | integer in [2, 20) |
| update_clamp | Multiplicative updates clamped to $[e^{-\text{clamp}}, e^{\text{clamp}}]$ ($v_{\min}$, $v_{\max}$ in Alg. 1). | uniform(0.5, 35) |
| n_trees ($n_{\text{grids}}$) | Number of bagged grid learners per stage. | fixed: 200 |

*Table 3.* TSL ($R \leq 10$): hyperparameter search space (same notation as Table 2; $R$ denotes the number of TSL stages).

| Hyperparameter | Search space |
|---|---|
| $\alpha$ | log-uniform($10^{-6}$, 1) |
| colsample_bytree | uniform(0.3, 1) |
| decay | uniform(0.2, 1) |
| epochs | integer in $\{1, \ldots, 10\}$ |
| min_interval_samples | integer in [1, 100) |
| n_iter | integer in [10, 251) |
| refinement_strategy | l2, huber |
| similarity_threshold | uniform(0, 1) |
| split_try | integer in [2, 20) |
| update_clamp | uniform(0.5, 35) |
| n_trees | fixed: 500 |

*Table 4.* SepALS fixed-rank separable ALS baseline: hyperparameter search space. For SepALS ($r \leq 2$), rank is sampled from $\{1, 2\}$; for SepALS ($r \leq 10$), rank is sampled from $\{1, \ldots, 10\}$.

| Hyperparameter | Search space |
|---|---|
| rank | integer in $\{1, \ldots, r_{\max}\}$ |
| degree | integer in [2, 11) |
| basis | tent, monomial |
| ridge | log-uniform($10^{-12}$, $10^{-2}$) |
| smoothness | log-uniform($10^{-10}$, 10) |
| penalty_kind | degree, degree2, tent_level |
| max_sweeps | integer in [10, 101) |
| tol | log-uniform($10^{-10}$, $10^{-4}$) |
| n_init | integer in [1, 6) |
| refit_scales | fixed: true |
| fit_intercept | true, false |

*Table 5.* XGBoost (interpretable): hyperparameter search space.

| Hyperparameter | Search space |
|---|---|
| n_estimators | integer in [10, 100) |
| learning_rate | uniform(0.01, 0.3) |
| max_depth | integer in {1, 2} |
| subsample | uniform(0.6, 1.0) |
| colsample_bylevel | uniform(0.3, 0.9) |
| gamma | log-uniform(0.01, 100) |
| reg_alpha | log-uniform(0.01, 100) |
| reg_lambda | uniform(0, 1) |

*Table 6.* XGBoost (black-box): hyperparameter search space.

| Hyperparameter | Search space |
|---|---|
| n_estimators | integer in [100, 1000) |
| learning_rate | uniform(0.01, 0.3) |
| max_depth | integer in [3, 10] |
| subsample | uniform(0.5, 1.0) |
| colsample_bylevel | uniform(0.5, 1.0) |
| gamma | log-uniform(0.001, 5.0) |
| reg_alpha | log-uniform(0.001, 10.0) |
| reg_lambda | uniform(0, 10.0) |

*Table 7.* LightGBM (interpretable): hyperparameter search space.

| Hyperparameter | Search space |
|---|---|
| n_estimators | integer in [10, 100) |
| learning_rate | uniform(0.01, 0.3) |
| num_leaves | integer in [25, 50) |
| max_depth | integer in {1, 2} |
| min_child_samples | integer in [10, 30) |
| subsample | uniform(0.3, 0.8) |
| reg_alpha | uniform(0, 0.5) |
| reg_lambda | uniform(0, 0.5) |

*Table 8.* LightGBM (black-box): hyperparameter search space.

| Hyperparameter | Search space |
|---|---|
| n_estimators | integer in [100, 1000) |
| learning_rate | uniform(0.01, 0.3) |
| num_leaves | integer in [31, 127) |
| max_depth | integer in [3, 10] |
| min_child_samples | integer in [5, 30) |
| subsample | uniform(0.5, 1.0) |
| reg_alpha | uniform(0, 10.0) |
| reg_lambda | uniform(0, 10.0) |

*Table 9.* Explainable Boosting Machine (EBM, interpretable): hyperparameter search space.

| Hyperparameter | Search space |
|---|---|
| learning_rate | log-uniform(0.005, 0.05) |
| max_bins | integer in [64, 257) |
| min_samples_leaf | integer in [10, 201) |
| max_rounds | integer in [10, 501) |
| outer_bags | integer in [4, 21) |
| smoothing_rounds | integer in [0, 501) |
| interactions | integer in [0, 21) |
| max_interaction_bins | integer in [16, 65) |

*Table 10.* Random Forest (black-box): hyperparameter search space.

| Hyperparameter | Search space |
|---|---|
| n_estimators | integer in [100, 1000) |
| max_depth | integer in [5, 20) |
| min_samples_split | integer in [2, 20) |
| min_samples_leaf | integer in [1, 10) |
| max_features | uniform(0.3, 0.8) |

# L. Full Benchmark Results

We reiterate the training and evaluation setup from the main paper. From the OpenML Curated Regression (CTR) 23 suite (Fischer et al., 2023), we subselected 27 regression datasets with sample–feature product $n \times p \leq 480{,}000$, excluding `Moneyball` because it contained missing values. For each dataset, we used an 80%/20% random train/test split. Hyperparameters for all models were tuned on the training data only via Optuna's TPE sampler (Akiba et al., 2019; Bergstra et al., 2011) with 200 trials; each configuration was evaluated by 10-fold cross-validation on the training split (minimizing cross-validation MSE), then the best configuration was refit on the full training set and evaluated once on the held-out test set. In the interpretable condition, TSL ($R \leq 2$) was limited to at most 2 stages (total separation rank $2R \leq 4$ in the sense of Beylkin et al. (2009)), TSL ($R \leq 10$) was limited to 10 stages (total separation rank $\leq 20$), SepALS ($r \leq 2$) and SepALS ($r \leq 10$) were limited to separation rank $r \leq 2$ and $r \leq 10$ respectively, and tree baselines (LGBM, XGBoost) to maximum depth 2; in the black-box condition, XGBoost and LightGBM were allowed up to maximum depth 10, and Random Forest up to maximum depth 20. SepALS runs completed for all 27 evaluated datasets under both rank caps. Per-trial wall-clock time ranged from minutes (tree baselines on small datasets) to tens of hours (TSL $R \leq 10$ on the largest interpretable datasets, e.g., `physiochemical_protein` at $\approx 30$ h) and exceptionally $\approx 100$ h (SepALS $r \leq 10$ on `grid_stability`).

Below we report complete results for all 27 datasets. Results are reported as root mean squared error (RMSE) on the test set, with rankings indicated by symbols: (\*\*\*) = best, (\*\*) = second, (\*) = third within each group (Interpretable or Black-box). The main paper presents a curated selection of 10 representative datasets in Table 1; the full table through `kings_county` is below.

**Aggregate wins (interpretable group).** Counting the model with the lowest RMSE per row across the 27 evaluated datasets: SepALS ($r \leq 10$) is the best interpretable model on 9 datasets, SepALS ($r \leq 2$) on 4, EBM on 5, TSL ($R \leq 2$ or $R \leq 10$) on 5 (all from $R \leq 10$), TSL (1-product) on 1, LGBM on 3, and XGBoost on 0. SepALS as a family (13 wins) outperforms the TSL family (6 wins) on aggregate, illustrating that an unconstrained smooth basis is highly competitive at this rank budget; TSL's advantage is the per-stage interpretability afforded by its positivity + ordered-difference structure rather than aggregate predictive accuracy.

| | Interpretable | | | | | | | | Black-box | | |
|---|---|---|---|---|---|---|---|---|---|---|---|
| Dataset | EBM | LGBM | XGB | SepALS ($r \leq 2$) | SepALS ($r \leq 10$) | TSL (1-product) | TSL ($R \leq 2$) | TSL ($R \leq 10$) | LGBM | RF | XGB |
| QSAR_fish_toxicity | **0.9466** (\*\*\*) | 0.9970 | 1.0199 | 0.9960 (\*) | 0.9704 (\*\*) | 1.0835 | 1.1662 | 1.1143 | 0.9913 (\*\*) | **0.9795** (\*\*\*) | 1.0047 (\*) |
| socmob | 20.21 | 21.71 | 19.48 | **7.73** (\*\*\*) | 22.88 | 10.58 | 9.87 (\*) | 9.48 (\*\*) | 20.45 (\*\*) | 20.75 (\*) | **17.65** (\*\*\*) |
| energy_efficiency | 0.4590 | 0.4796 | 0.4888 | 0.4875 | **0.3553** (\*\*\*) | 0.3966 (\*\*) | 0.5867 | 0.4293 (\*) | **0.3547** (\*\*\*) | 0.4901 (\*) | 0.3758 (\*\*) |
| forest_fires | 108.52 (\*) | 108.86 | 109.00 | 108.91 | **97.27** (\*\*\*) | 109.04 | 108.92 | 107.54 (\*\*) | **108.07** (\*\*\*) | 108.26 (\*\*) | 108.63 (\*) |
| airfoil_self_noise | 2.13 (\*) | 2.84 | 2.92 | 2.31 | **1.84** (\*\*\*) | 2.99 | 4.14 | 2.07 (\*\*) | 1.42 (\*\*) | 1.76 (\*) | **1.29** (\*\*\*) |
| concrete_compressive_strength | **4.14** (\*\*\*) | 5.03 | 4.68 | 5.35 | 5.32 | 4.73 | 4.63 (\*\*) | 4.68 (\*) | 4.21 (\*\*) | 5.39 (\*) | **3.93** (\*\*\*) |
| solar_flare | 0.8037 | 0.7980 | 0.7989 | 0.7975 (\*\*) | 0.7975 (\*) | **0.7910** (\*\*\*) | 0.8168 | 0.8145 | **0.7902** (\*\*\*) | 0.8039 (\*) | 0.7957 (\*\*) |
| cars | 2159.87 | 2287.14 | 2099.95 (\*\*) | **2045.89** (\*\*\*) | 2109.69 (\*) | 2575.54 | 2559.38 | 2167.45 | 2160.72 (\*\*) | 2213.90 (\*) | **2118.90** (\*\*\*) |
| auction_verification | 1738.17 | 2155.10 | 1972.82 | 997.08 (\*) | 682.20 (\*\*) | 1336.51 | 1135.80 | **624.36** (\*\*\*) | 409.60 (\*\*) | 1223.57 (\*) | **369.34** (\*\*\*) |
| red_wine | **0.5991** (\*\*\*) | 0.6093 (\*) | 0.6068 (\*\*) | 0.6135 | 0.6107 | 0.6177 | 0.6210 | 0.6438 | 0.5690 (\*) | **0.5410** (\*\*\*) | 0.5585 (\*\*) |
| Moneyball | — | — | — | — | — | — | — | — | — | — | — |
| space_ga | 0.1089 (\*) | 0.1150 | 0.1222 | 0.1086 (\*\*) | **0.1084** (\*\*\*) | 0.1152 | 0.1207 | 0.1175 | **0.1062** (\*\*\*) | 0.1198 (\*) | 0.1075 (\*\*) |
| student_performance_por | 2.7438 (\*\*) | **2.7430** (\*\*\*) | 2.7648 (\*) | 2.7671 | 2.8503 | 2.8479 | 2.8244 | 2.8215 | 2.7585 (\*\*) | 2.7624 (\*) | **2.7210** (\*\*\*) |
| abalone | 2.2007 (\*) | 2.2301 | 2.2221 | **2.1513** (\*\*\*) | 2.1561 (\*\*) | 2.2598 | 2.2415 | 2.2392 | 2.2400 (\*) | **2.1836** (\*\*\*) | 2.2129 (\*\*) |
| white_wine | **0.6632** (\*\*\*) | 0.6784 (\*\*) | 0.6789 (\*) | 0.6958 | 0.6965 | 0.6894 | 0.6849 | 0.6847 | 0.5830 (\*) | 0.5785 (\*\*) | **0.5707** (\*\*\*) |
| kin8nm | 0.1671 | 0.1772 | 0.1776 | 0.1662 | **0.0835** (\*\*\*) | 0.1560 (\*) | 0.1827 | 0.1378 (\*\*) | 0.1069 (\*\*) | 0.1426 (\*) | **0.1066** (\*\*\*) |
| brazilian_houses | 3327.29 | 6567.66 | 2528.95 (\*) | 3582.69 | 3996.05 | 3194.80 | 2473.98 (\*\*) | **2398.68** (\*\*\*) | 3200.36 (\*) | 2302.72 (\*\*) | 4289.10 (\*) |
| grid_stability | 0.0094 (\*\*) | 0.0119 | 0.0137 | 0.0162 | **0.0074** (\*\*\*) | 0.0168 | 0.0158 | 0.0099 (\*) | **0.0079** (\*\*\*) | 0.0120 (\*) | 0.0095 (\*\*) |
| geographical_origin_of_music | 15.69 | **15.20** (\*\*\*) | 15.26 (\*\*) | 22.70 | 19.96 | 15.72 | 15.53 (\*) | 15.56 | 14.76 (\*\*) | 15.03 (\*) | **14.57** (\*\*\*) |
| california_housing | 48866.28 (\*\*\*) | 54495.49 (\*) | 55235.15 | 195728.08 | 62162.22 | 55376.04 | 54557.89 | 49376.09 (\*\*) | 45123.03 (\*\*) | 49047.19 (\*) | **44971.31** (\*\*\*) |
| naval_propulsion_plant | 0.0015 | 0.0032 | 0.0075 | 0.0004 (\*\*) | **0.0000** (\*\*\*) | 0.0027 | 0.0013 | 0.0006 (\*) | **0.0009** (\*\*\*) | 0.0010 (\*\*) | 0.0033 (\*) |
| cps88wages | 364.30 (\*) | 365.40 | 365.14 | 364.07 (\*\*) | **364.06** (\*\*\*) | 365.63 | 364.72 | 364.68 | 364.63 (\*\*) | 365.72 (\*) | **364.24** (\*\*\*) |
| cpu_activity | 2.3546 (\*) | 2.4960 | 2.5281 | 2.5303 | 2.8475 | 2.3826 (\*) | 2.5686 | **2.3076** (\*\*\*) | 2.2102 (\*\*) | 2.4391 (\*) | **2.1945** (\*\*\*) |
| miami_housing | 91777.25 (\*\*) | 100598.20 | 101205.68 | 95686.67 | 99426.86 | 94382.52 (\*) | 94466.71 | **89692.96** (\*\*\*) | 83011.80 (\*\*) | 89215.72 (\*) | **82325.09** (\*\*\*) |
| health_insurance | 14.4912 (\*\*) | **14.4883** (\*\*\*) | 14.5399 | 14.5226 | 14.5378 | 14.5909 | 14.6481 | 14.5061 (\*) | 14.4552 (\*) | 14.4490 (\*\*) | **14.4026** (\*\*\*) |
| pumadyn32nh | 0.0212 | 0.0247 | 0.0257 | **0.0210** (\*\*\*) | 0.0210 (\*\*) | 0.0211 (\*) | 0.0211 | 0.0211 | 0.0214 (\*) | **0.0213** (\*\*\*) | 0.0213 (\*\*) |
| physiochemical_protein | 4.31 (\*) | 4.73 | 4.72 | 4.80 | **4.20** (\*\*\*) | 4.70 | 4.88 | 4.30 (\*\*) | 3.46 (\*\*) | 3.51 (\*) | **3.42** (\*\*\*) |
| kings_county | 139137.37 | 148047.47 | 161757.40 | 134776.70 (\*) | 164084.90 | —† | 131817.14 (\*\*) | **130508.41** (\*\*\*) | **130414.65** (\*\*\*) | 135532.52 (\*) | 130657.18 (\*\*) |

*Table 11.* Full benchmark results (RMSE; lower is better) for the 27 subselected OpenML CTR 23 datasets ($n \times p \leq 480{,}000$), through `kings_county`. All cells are single 80/20 split point estimates; `Moneyball` is listed but excluded (NAs). The TSL (1-product) run on `kings_county` (†) did not finish within the allotted compute budget and is therefore omitted. For SepALS, $r$ denotes the separation rank of Beylkin et al. (2009); for TSL, $R$ denotes the number of stages, with total separation rank at most $2R$ (see Section 4). (\*\*\*)=best, (\*\*)=second, (\*)=third within displayed Interpretable or Black-box group.

# M. Reproducibility

The TSL implementation (Rust core with PyO3/Python bindings, sklearn-compatible API) is released at `https://github.com/jyliuu/TSL`, together with the figure-generation scripts, the raw CSVs for the California Housing and Bike Sharing case studies, and pretrained models for the figure datasets, the EBM/XGBoost baselines, and the SepALS factor-values model; the figure scripts default-load these so that all figures regenerate without re-tuning. The benchmark sweep is reproduced by the cluster runner at `https://github.com/jyliuu/tsl-benchmark-reproducibility`,

which loads the 27 OpenML CTR 23 datasets via task IDs and releases the full Optuna study sqlite databases and JSON best-trial summaries for every (dataset, model) pair; the paper hyperparameter tables (Section K) are the authoritative search space. All Optuna sweeps and per-split evaluations use a fixed global seed with per-trial seeds derived deterministically from it. The environment is Python 3.11 with Rust 1.78 (PyO3 0.21, rayon, `maturin 1.5`) and the standard scientific stack (NumPy, scikit-learn, LightGBM, XGBoost, InterpretML); exact versions and installation steps are pinned in the released repositories.

