# OpenReview forum: "Beyond Additive Decompositions: Interpretability Through Separability"
_ICML.cc/2026/Conference — ICML 2026 regular_

### Official Review · Reviewer_1WXA · 2026-03-02

**Soundness:** 4
**Presentation:** 3
**Significance:** 4
**Originality:** 4
**Overall Recommendation:** 5
**Confidence:** 3

**Summary:**

The paper proposes Tensor Separation Learning, a regression model that aims to overcome the limitations of additive models in terms of interpreation. By enforcing separability, the model avoids the information loss that can be present in additive projections by marginalizing high-order interactions and partial dependence contamination. The authors provide extensive evaluations using the OpenML CTR 23 regression suite, comparing their methods against both black-box and white-box approaches.

**Compliance With Llm Reviewing Policy:**

Affirmed.

**Final Justification:**

Overall, I maintain a positive evaluation of this paper and support an accept recommendation.

The paper introduces Tensor Separation Learning (TSL), a novel regression framework that aims to address known limitations of additive models in interpretability, particularly the loss of information due to marginalization of higher-order interactions. I find the motivation well grounded and the proposed approach both original and technically interesting. The manuscript is generally well written, and I was able to follow both the intuition and the mathematical formulation.

In terms of soundness, the work is strong. The theoretical analysis provides approximation guarantees for mixed-smoothness function classes, and the empirical evaluation on the OpenML CTR-23 benchmark is extensive and demonstrates that TSL achieves competitive predictive performance relative to both white-box and black-box methods.

Regarding originality, this is one of the key strengths of the paper. The move from additive to separable representations, together with the explicit connection to interpretability via partial dependence structures, represents a meaningful contribution beyond existing GAM-based approaches. The method is well positioned within the broader literature on interpretable models.

In terms of significance, the work has clear potential impact. It addresses an important limitation of additive models and proposes a framework that could be valuable for practitioners seeking more expressive yet interpretable models. The empirical results further support its practical relevance.

My main concern in the original review was related to identifiability, which I considered a critical aspect for interpretability. Specifically, I noted that TSL is only approximately identifiable and that different runs may yield different factorizations, which could limit the reliability of the learned explanations. The authors’ rebuttal addresses this concern in a thorough and thoughtful manner. In particular:

- They clarify that identifiability in TSL differs from classical tensor decomposition settings due to the supervised-learning regime and non-rectangular support.
- They explain the inherent ambiguities in multiplicative factorizations (e.g., scaling and sign indeterminacies), and provide concrete examples illustrating non-identifiability.
- They argue convincingly that exact identifiability is not strictly required for interpretability, and that interpretability in TSL is instead grounded in the direct correspondence between model components and partial dependence functions, as well as the backbone–tilt parametrization.

I find this clarification satisfactory and it improves my understanding of the intended notion of interpretability in the paper. Additionally, the authors commit to expanding the discussion of limitations, including identifiability, in the revised manuscript, which is an important improvement.

The rebuttal also constructively addresses my other comments:
- They acknowledge the value of additional controlled simulations and outline possible extensions.
- They provide a clearer positioning relative to separability-based methods.
- They agree to improve the presentation in the introduction and add a dedicated limitations subsection.

Overall, the rebuttal fully addressed my main concerns and strengthened my confidence in the contribution. While some limitations remain (particularly regarding identifiability and the need for broader simulation studies) they are now clearly acknowledged and do not undermine the core contribution.

Taking all aspects into account, I believe the paper makes a solid and interesting contribution, and I support acceptance.

**Key Questions For Authors:**

1. Controlled simulation experiments: while the paper provides extensive experiments on the OpenML CTR-23 benchmark, have the authors considered including controlled simulation studies to better characterize the behavior of the proposed method? For example, simulations with varying response distributions or different noise/variance conditions could help analyze the robustness of the approach. Additionally, such experiments would allow evaluation of how accurately TSL can recover the true functional form of each covariate, which is central to the interpretability claims of the method.

2. Comparison with separability-based methods: the empirical comparison focuses mainly on gradient-boosting methods and Random Forests. Since the related work discusses approaches based on separability rather than additivity (e.g., Beylkin et al., 2009; Beylkin & Mohlenkamp, 2005; Garcke, 2010), could the authors comment on how the proposed method compares with these approaches? If empirical comparisons are not feasible, a more detailed discussion of the expected advantages or limitations relative to these methods would help clarify the positioning of the contribution.

3. Presentation of the model in the introduction: in the Introduction, the regression model formula is introduced perhaps too early, before the notation has been fully defined. Would the authors consider replacing this with a more narrative description of the model at that stage, and leaving the formal mathematical specification for the methodological section? This might improve readability for readers encountering the framework for the first time.

**Limitations:**

I think the limitations (in particular in terms of identifiability) are not adequately addressed. The authors expend an entire section on detailing the limitations of additive decomposition, and barely use a few lines to describe the limitations of their own work. Given that the results also don't show the PDP in simulation scenarios, at this point, it is not possible to fully assess the potential of their proposal.

**Strengths And Weaknesses:**

**Strenghts**

- The paper is well written and interesting, addressing a known issue of white-box approaches based on additivity constraints such as Generalized Additive Models and its modern variants (ie EBMs, NAM).
- Overall, I was able to follow the novelty and the math even without a formal methods background.
- TSL achieves competitive predictive performance when compared to black-box alternatives.

**Limitations**

- The theoretical analysis covers an approximation-rate guarantee for a mixed-smoothness function class, but does not address additional aspects. A key limitation is that TSL is only approximately identifiable, and different runs under different seeds yield different factorizations. I believe this limitation is underestimated by the authors, since identifiability is a key requirement for interpretability. I think that a deeper discussion of this limitation is needed to completely assess the suitabilty of the proposed method.

---

> ### Author Rebuttal · Authors · 2026-03-29
>
> We thank the reviewer for the positive assessment and thoughtful comments.
>
> ---
>
> ## W1: Identifiability — Expanded Treatment
>
> We appreciate this concern and agree that identifiability deserves a fuller discussion. A useful clarification is that identifiability in TSL does not map directly onto classical PARAFAC/CANDECOMP. In that literature, the object of study is typically an observed tensor, and uniqueness is characterized by algebraic conditions such as Kruskal-rank assumptions. TSL operates in the supervised-learning regime, where observations are noisy and the joint covariate support may be non-rectangular. In this setting, classical tensor-identifiability results do not transfer directly, because what can be identified depends also on the geometry of the observed support.
>
> **Identifiability in multiplicative factorizations** For a product of univariate functions $\prod_j h_j(x_j)$, the factors are at best identifiable up to sign changes and scaling; for example, $-h_1 \cdot -h_2 = c h_1 \cdot h_2/c$. This ambiguity is not specific to TSL, but is inherent to multiplicative factorizations more generally; classical CP identifiability results likewise guarantee uniqueness only up to scaling, sign, and permutation. Moreover, with non-rectangular support, non-identifiability can arise in ways that go beyond these basic symmetries.
>
> As a concrete example, let $\mathcal X = A \cup B$, where $A=[-1,0]\times[-1,0]$ and $B=[0,1]\times[0,1]$, and define
> $$
> m(x_1,x_2)=\mathbf 1_A f_1(x_1)f_2(x_2)+\mathbf 1_B h_1(x_1)h_2(x_2).
> $$
> Then the following three distinct rank-1 functions all agree with $m$ on $\mathcal X$:
> $$
> m_1=\big(\mathbf 1_{[-1,0]}f_1+\mathbf 1_{[0,1]}h_1\big)\big(\mathbf 1_{[-1,0]}f_2+\mathbf 1_{[0,1]}h_2\big),
> $$
> $$
> m_2=\big(-\mathbf 1_{[-1,0]}f_1+\mathbf 1_{[0,1]}h_1\big)\big(-\mathbf 1_{[-1,0]}f_2+\mathbf 1_{[0,1]}h_2\big),
> $$
> $$
> m_3=\big(-\mathbf 1_{[-1,0]}f_1-\mathbf 1_{[0,1]}h_1\big)\big(-\mathbf 1_{[-1,0]}f_2-\mathbf 1_{[0,1]}h_2\big).
> $$
> Thus, even within the rank-1 class, the same fitted function on a non-rectangular support may admit multiple distinct factorizations. In our early experiments with unconstrained signed components on California Housing (non-rectangular support across latitude and longitude), we observed the same behavior: different grid tensors realized equivalent signed factorizations of the same fitted function.
>
> **Interpretability does not require exact identifiability.** Exact identifiability is not necessary for interpretability of a fitted model. One does not require identification of weights in a neural network or leaf regions in a tree ensemble, indeed, PD-based explanations of these models are understood as properties of the fitted predictor. The same applies to TSL: in our case, interpretability comes from the fact that partial dependence objects are directly represented by the model structure: Proposition 5.1 shows that the PD functions can be read directly from the fitted components, providing a transparent link between the model representation and an interpretability object. In addition, the backbone–tilt parametrization gives an interpretable view of feature effects, as illustrated in Figure 1: the backbone acts as a gating function, since regions where it is zero contribute zero to the model. Together, the PD correspondence and the backbone–tilt visualization form the basis of our interpretability claim; we elaborate in our response to Reviewer EsMe.
>
> ---
>
> ## W2: Controlled Simulations
>
> We agree that a broader simulation study across multiple data-generating mechanisms would be valuable. The current paper already includes control simulations (Figure 3), but this can be expanded. In our response to Reviewer zSd4, we provide an additional simulation (Figure R1) illustrating how independently fitted grids can converge to different representations and how similarity filtering resolves this. Another natural benchmark is to consider a data generating process in which the stagewise $L^2$ tensor projection can be computed analytically at each step, allowing a cleaner comparison between the population update and the practical fitted procedure.
>
> ---
>
> ## W3: Comparison with Separability-Based Methods
>
> Existing tensor-factorization methods typically assume data on a grid and fit a fixed-rank model via ALS, so components are estimated jointly rather than added sequentially. By contrast, TSL is stagewise and greedy: early stages absorb most of the explained variance, while later stages mainly act as refinements. Moreover, basis-function tensor methods are often less natural for sparse, noisy tabular data with outliers or non-rectangular support, which is the regime TSL is designed to address.
>
> ---
>
> ## W4: Limitations and presentation
>
> We will add an explicit Limitations subsection with a clearer discussion of identifiability constraints. To improve readability, we will replace the early regression-model formula in the Introduction with a more narrative description.

---

> > ### Author Rebuttal · Reviewer_1WXA · 2026-04-02
> >
> > Thanks for your detailed answer, my concerns have been resolved.

---

### Official Review · Reviewer_Dztf · 2026-03-03

**Soundness:** 2
**Presentation:** 2
**Significance:** 2
**Originality:** 1
**Overall Recommendation:** 2
**Confidence:** 4

**Summary:**

This paper proposes using a particular kind of low-rank sum of separable functions model for interpretable supervised learning tasks. The particular model proposed here is a difference of two such models with nonnegativity constraints on the separable factors. Each component 1D function is further assumed to be piecewise constant with some initial breakpoints which are adaptively refined using a CART-like procedure during model fitting. The proposed model fitting process is incremental, proceeding to update one pair (difference) of separable nonnegative components at a time, and then only updating the weights of previously extracted components.

**Compliance With Llm Reviewing Policy:**

Affirmed.

**Final Justification:**

I think that the authors argue that they can interpret a given model that is computed by their method. If there exists another model that is equivalent from the input-output point of view though, and that other model has a different "visualization"-based interpretation, then what? Moreover, if we have a single product of positive 1-D functions, then yes, I can interpret that increasing a variable by a factor of 1.5 increases the product by a factor of 1.5. But what if I have a difference of sums of such products? What happens with "visualization" then?

**Key Questions For Authors:**

See main weaknesses in review above.

1) Does the proposed method offer uniqueness guarantees? If not, how can it be interpretable?
2) Deflation-type approaches such as the one proposed here are known to be very suboptimal in tensor decomposition because, unlike SVD, the rank-one factors are not orthogonal and thus no orthogonality can be imposed. Why did the authors opt for a deflation-based approach?
3) Why piecewise constant versus smooth / splines?

**Limitations:**

I like the fact that in their numerical results the authors have presented cases where their algorithm does not win. This is a good sign.

**Strengths And Weaknesses:**

Strengths:

+ Interpretable supervised learning is an important problem.
+ Low-rank tensor models of this kind are useful tools towards interpretable learning as they have simpler structure and uniqueness guarantees.

Weaknesses:

- Why model the 1D functions as piecewise constant? Most of the relevant literature uses smooth 1D functions, such as polynomials, splines, or Fourier bases, because real-world regression functions are usually smooth due to physical considerations. Second, and more important, why use a sum of differences of non-negative separable functions to model real-valued functions? Why not fit a standard unconstrained sign low-rank model?
- Note that by decomposing a real-valued tensor in this way you may lose identifiability because you over-parameterize: you are using twice as many degrees of freedom than needed for exact decomposition. Think of the difference between matrix rank and positive rank which can exceed rank even for positive matrices.
- The authors suggest that they do this splitting for interpretability purposes, and cite Donoho and Stoden for this. But if the parameters of the model are not even unique due to over-parametrization, how can they ever be interpreted?  Also, alluding to NMF interpretability is very vague: NMF entails no minus sign whatsoever. NMF "constructively" decomposes a matrix that is positive into positive parts. But if the function itself is not positive this "interpretation" does not make sense. Would we use the difference of two NMFs to model a real-valued matrix? Would that make sense or make the decomposition "interpretable"?
- I also have concerns about the literature review. There are many additional references reporting work on sum-of-separable function models in the tensor literature, e.g., using Fourier or spline bases, and these are not cited. When based on the canonical polyadic decomposition, these models are essentially unique -- except for permutation and scaling ambiguity. They are not necessarily interpretable, but they are at least essentially unique.
- The incremental / greedy forward decomposition proposed in this paper may further sacrifice uniqueness and even increase the rank of the function due to initial errors. There is no way to get an optimal model this way, and that will almost inevitably lead to over-parametrization and loss of uniqueness.

The above concerns make the rest of the claims hard to parse -- especially because they are not well-explained in the manuscript. Also, the novelty is low relative to other literature that is not properly cited here, for example

Govindarajan N, Vervliet N, De Lathauwer L. Regression and Classification With Spline-Based Separable Expansions. Front Big Data. 2022 Feb 11;5:688496. doi: 10.3389/fdata.2022.688496. PMID: 35224482; PMCID: PMC8874272.

N. Kargas and N. D. Sidiropoulos, "Supervised Learning and Canonical Decomposition of Multivariate Functions," in IEEE Transactions on Signal Processing, vol. 69, pp. 1097-1107, 2021, doi: 10.1109/TSP.2021.3055000.

E. M. Stoudenmire and David J. Schwab. 2016. Supervised learning with tensor networks. In Proceedings of the 30th International Conference on Neural Information Processing Systems (NIPS'16). Curran Associates Inc., Red Hook, NY, USA, 4806–4814.

---

> ### Author Rebuttal · Authors · 2026-03-29
>
> We thank the reviewer for the careful reading and thoughtful comments.
>
> ## D1: Identifiability and Uniqueness
>
> To address identifiability and its relation to interpretability, it is useful to distinguish two goals.
>
> **Structural or latent-factor interpretability.** In many tensor applications, the observed tensor is itself the object of study, and the goal is to recover real latent sources. Here, strong algebraic identifiability matters because the recovered factors correspond to real underlying sources, not one of several equivalent parametrizations, and uniqueness results are directly relevant for scientific interpretation.
>
> **Fitted-model interpretability.** In supervised learning, one observes outcome-feature pairs $(Y_i, X_i)$ from an unknown distribution $P$, and the goal is to fit a predictor $\hat m$ that approximates the regression function $E_P[Y \mid X=x]$. In our setting, this predictor is built from products of component functions, $\prod_j \hat m_j(x_j)$. A natural question is whether the fitted components $\hat m_j$ recover corresponding components of the true regression function, but without additional assumptions on $P$ such identification cannot generally be expected and may be impossible; see Point W1 in our response to Reviewer 1WXA.
>
> We argue that $\hat m$ is nonetheless interpretable because its components can be easily visualized. Furthermore, by imposing positivity constraints on the components, we are able to improve stability and predictive accuracy. At each stage, $n_{\mathrm{grid}}$ grid tensors are fitted and aggregated by averaging their components. If the sign of the components is unconstrained, averaging will induce cancellation and produce a noisier aggregate. Furthermore, when $P$ has non-rectangular support, the sign ambiguity may occur simultaneously across multiple regions (see again Point W1). Positivity removes this instability and also aids interpretation: by fitting an ordered difference of two grid tensors, the fitted components have a consistently positive or negative effect regardless of the other coordinates.
>
> ## D2: Missing Related Work
>
> We will cite the omitted works and discuss Kargas and Sidiropoulos (2021) in the revised identifiability section. Their framework differs from ours in two important respects. First, they study functions $f:[0,1]^N \to \mathbb{R}$, over a compact rectangular support — in contrast to the non-rectangular support often encountered in the general regression setting that is considered here. Second, their identifiability result concerns the Fourier coefficient tensor induced by $f$, rather than the fitted predictor itself. In particular, if $f$ admits a finite multidimensional Fourier expansion, then conditions on that coefficient tensor can guarantee uniqueness of the corresponding decomposition. They also note that noise in estimation may destroy exact low-rank structure in practice, so the result applies under idealized assumptions rather than directly to fitted regression models.
>
> ## D3: Piecewise Constant vs. Smooth Bases
>
> The piecewise constant structure in TSL arises naturally from the greedy CART-style splitting procedure, motivated by the strong empirical success of tree ensembles on tabular data. We do not view this as sacrificing accuracy, as partition estimators are universal approximators and the tree-style refinement concentrates resolution where the fitted function changes most. This is consistent with the empirical results in Table 1, where TSL is competitive with strong tree-based baselines. Moreover, tabular data often contain threshold effects, discretization, and censoring, so piecewise behavior is frequently a feature of the signal rather than an artifact of the model. Adaptive sparsity also comes naturally since variables or regions within variables with little contribution are simply left unsplit.
>
> ## D4: Greedy forward-additive modeling (deflation)
>
> Boosting methods have achieved wide success from employing greedy forward-additive modeling, and we adapt it here by using separable tensors as base learners. While such a stagewise strategy is not optimal for tensor decomposition, TSL is not intended to recover a ground-truth tensor from an exact low-rank model. In classical tensor decomposition, one fixes the rank $r$ and studies rank-$r$ representations of a given tensor; the usual identifiability guarantees apply when the tensor truly admits a rank-$r$ decomposition. In regression, by contrast, the underlying function is not directly observable and the effective rank is unknown. We therefore use a stagewise procedure in which $r$ is tuned, adding one rank-1 component at a time until further stages explain little residual variation. Empirically, this is reasonable: Appendix Figure 4 shows later stages contribute very little, and Appendix Table 10 shows only small gains from increasing $r$ from 2 to 10 on datasets such as socmob and QSAR.

---

> > ### Author Rebuttal · Reviewer_Dztf · 2026-04-02
> >
> > I would like to acknowledge the authors responses.
> >
> > Regarding interpretability, the authors' responses are very unclear: "We argue that $\hat m$ is nonetheless interpretable because its components can be easily visualized." - what does this mean?  "Furthermore, by imposing positivity constraints on the components, we are able to improve stability and predictive accuracy." -- why and how would the authors substantiate this assertion?  "At each stage, $n_{\mathrm{grid}}$ grid tensors are fitted and aggregated by averaging their components. If the sign of the components is unconstrained, averaging will induce cancellation and produce a noisier aggregate." Why?
> >
> > Why not model a low rank real-valued matrix as the difference of two non-negative matrices then? Would that improve interpretability and robustness?
> >
> > I understand that what we care about is identifiability of the regression function itself, not necessarily its latent factors. If the latent factorization is unique then the function is unique, but not necessarily the other way around. However, can the authors establish identifiability of their regression function without going through a unique latent factorization?
> >
> > D2: Identifiability of the FS coefficient tensor implies identifiability of the learned function and vice-versa, assuming that the sought function is smooth in the FS sense. Identifiability assumes a certain rank and bandlimit, which are idealized assumptions, but it also has a practical implication: suppose you fix the residuals (errors) in your fitted model, and you ask whether your fitted model is unique *for the given residuals*. If your fitted model is not unique, then even conditioned on the presumed errors you can find another model which is as good as your fitted one. Which is a serious blow to interpretability.
> >
> > For these reasons, I maintain my original evaluation.

---

> > > ### Author Response · Authors · 2026-04-05
> > >
> > > We thank the reviewer for the continued engagement. The follow-up questions largely concern material already addressed in our original response (Point D1–D2); we reiterate the key points below with precise citations.
> > >
> > > **Definition of interpretability.**
> > >
> > > Our paper uses the standard notion from the Interpretable ML community: a model $\hat m$ is interpretable if a human can understand how $\hat m$ computes its predictions. This is a statement about the fitted predictor, not about uniquely recovering latent components of the true regression function $m$ (Point D1).
> > >
> > > In general, two different fitted predictors $\hat m_1$ and $\hat m_2$ can fit finite noisy data equally well. This is not specific to TSL; it is fundamental to all nonparametric regression — neural networks, random forests, and boosted trees all share this property. Regarding the concern that non-uniqueness is "a serious blow to interpretability," our interpretability claim does not require unique recovery of the true regression function. It is conditional on the fitted predictor $\hat m$: once $\hat m$ is fixed, TSL provides an exact low-dimensional representation of what that fitted predictor computes (Proposition 5.1). This is a form of transparency that black-box methods do not offer. See also our response to Reviewer 1WXA (Point W1) for a concrete example.
> > >
> > > **Why TSL is interpretable.**
> > >
> > > For a given fitted model — regardless of identifiability concerns or whether its internal structure maps to the true data-generating process — visualization is a primary tool for interpretability: if the model's internal structure can be faithfully depicted in low-dimensional plots, a human can inspect and reason about its behavior. For example, a 1D plot showing that a component $\hat m^{(\ell)}_{+,k}(x_k)$ is monotonically increasing immediately tells the practitioner that larger values of $x_k$ amplify the positive contribution of that stage — no further analysis is needed. For most black-box models, objects such as PD plots or SHAP values are post-hoc summaries that do not in general determine $\hat m$. In TSL, the stagewise 1D partial dependence plots are sufficient statistics of each fitted stage up to a scalar normalizer — the visualization is a faithful, exact representation of $\hat m$, not a surrogate. See also our response to Reviewer EsMe (Point W2) for the backbone/tilt interpretation.
> > >
> > > **Role of positivity in interpretability.**
> > >
> > >
> > > Each side of a TSL stage is a product $\hat{m}\_{\pm}^{(\ell)}(x) = \prod_{j=1}^{p} \hat{m}_{\pm,j}^{(\ell)}(x_j)$.
> > >
> > > Positivity makes each component's contribution readable in isolation: if all factors are positive, then increasing $\hat{m}\_{\pm,k}^{(\ell)}(x_k)$ always increases the product, regardless of the other coordinates $X_{-k}$. Without positivity, whether a factor increases or decreases the stage output depends on the signs of the remaining factors. With positivity, each factor's effect is unambiguous — a value of $1.5$ always means a 50\% increase on that side of the stage.
> > >
> > > **Role of positivity in aggregation.**
> > >
> > > In supervised learning, tree-based ensembles have shown very strong empirical performance on tabular data, and their success relies on averaging many fitted learners to improve stability and accuracy. TSL follows this ensemble logic at the stage level, fitting multiple bootstrap-based grid tensors and aggregating them componentwise. Positivity plays a distinct role in stabilizing this aggregation (Point D1). Without positivity, $h_1 h_2 = (-h_1)(-h_2)$, so different bootstrap samples may have opposite sign orientations, and componentwise averaging gives $\tfrac{1}{2}(h_1 + (-h_1)) = 0$, destroying signal. Positivity ensures a shared orientation, preventing cancellation. Remaining permutation/rescaling ambiguities are addressed via similarity fitting (see our response to Reviewer zSd4).
> > >
> > > **Low-rank matrix decomposition.**
> > >
> > > Whether this is useful depends on the application; we do not have a general opinion on matrix decomposition strategies. However, this question does not directly apply to TSL, as TSL does not decompose an observed matrix or tensor. TSL is a supervised learning method that models the regression function from scattered noisy outcome–feature pairs. The positivity constraint is motivated by stabilizing bootstrap aggregation and from an interpretability perspective (see above), not by matrix factorization considerations.
> > >
> > > We hope these clarifications are helpful and remain available for any further questions.

---

### Official Review · Reviewer_zSd4 · 2026-03-07

**Soundness:** 3
**Presentation:** 3
**Significance:** 3
**Originality:** 3
**Overall Recommendation:** 4
**Confidence:** 3

**Summary:**

This paper proposed a Tensor Separation Learning framework to solve the limitations in current additive structured interpretable methods.

**Compliance With Llm Reviewing Policy:**

Affirmed.

**Key Questions For Authors:**

1. On page 5 section 4, what is the measurement of the similarity?
2. Also, for the top k candidate you selected, how did you select $\xi$? Is there any statistical inferential guarantee for reducing the variance?
3. It is slightly surprising that the table 1result suggests TSL is weaker when the true model is additive. It might be helpful if there is any tested simple additive structured simulations to verify it.
4. How robust is the TSL when the working model is misspecified (especially in the high-dimensional setting)?

**Limitations:**

Yes

**Strengths And Weaknesses:**

Pros: The motivation of the work is clear and well-written, and the limitations of the current work related to the current explainable tools is clearly stated.
Cons: See questions below. Also, the author(s) may consider adding a flowchat to visualize the framework of TSL.

---

> ### Author Rebuttal · Authors · 2026-03-29
>
> We thank the reviewer for the positive assessment and the thoughtful technical questions. The reviewer raises the following points:
>
> - **[Q1]** What is the similarity metric used in Section 4?
> - **[Q2]** How is the trimming parameter $\xi$ selected, and are there statistical guarantees for the variance reduction?
> - **[Q3]** TSL appears weaker when the true model is additive; are there simple additive simulations to verify this?
>
> We believe these questions are best addressed together, by first explaining how the combined grid tensor is assembled — because the similarity/top-$k$ filtering step becomes clear only in that light.
>
> ---
>
> ## TSL fitting process
>
> At each stage, TSL fits $n_{\text{grids}}$ independent grid tensors in parallel on bootstrap samples (Algorithm 10 in the appendix), then combines them into a single stage tensor (Algorithm 11). A key subtlety — discussed further in our response to Reviewer 1WXA — is that grid tensors are **not uniquely identified**: the tensor product structure admits multiple backbone configurations that explain the same data equally well, so independent grids initialized with different random seeds can converge to different but similarly fitted representations.
>
> **An illustrative example.** We illustrate this in the following figure (https://www.dropbox.com/scl/fi/8a64c4gv7gk10hvdestfz/backbone_bimodal_epoch0.png?rlkey=ibsv2gohzarr4b1bp2b3abf9b&e=1&st=79wf0p05&dl=0), which shows the backbone curves $b_j^{(i)}$ for TSL that has been tuned with 389 grid tensors on the DGP $f(x_1, x_2) = \exp(\sin(x_1)\cos(x_2)) + x_1$, trained on $n = 5000$ observations with $x_1, x_2 \overset{\text{iid}}{\sim} \text{Uniform}[-4, 4]$. The figure has three rows. In the first, we split the 389 tensors into two extreme groups of 17 by their total scale $\lambda := \lambda^+ + \lambda^-$. Depending on their random seed, the backbones appear to have converged to one of two representations: the high-$\lambda$ group places a pronounced bump in Feature 0, while the low-$\lambda$ group distributes the signal differently and, in Feature 1, moves in the opposite direction. Without filtering, averaging the backbones across both groups will produce a combined curve that does not correspond to either representations, and is unlikely to minimize the error — backbone averaging is not component-aware, so it cannot account for the fact that each group compensates differently across features; the result destroys the per-feature interpretable structure that TSL is designed to provide. The third row shows all 389 tensors colored continuously by $\lambda$, confirming that the shape of the two converged representations.
>
> **How similarity filtering resolves this.** The second row shows the output of the combination step (Algorithm 11). We first select a reference grid $g^\star$ as the medoid in $(\lambda_+, \lambda_-)$ space. We then score each candidate by its cosine similarity to $g^\star$, (see Appendix Algorithm 11 for the full definition including the tilt component and distance-to-medoid factor). Restricting to the 39 tensors with combined similarity $\geq \xi := 0.9$ to the medoid reference tensor yields a coherent set that all share the Feature 0 bump, with the competing representation filtered out entirely, so that the averaged backbone is a sharp and stable estimate. We do not currently have a formal statistical guarantee for the variance reduction from this filtering step; the threshold $\xi$ is tuned via cross-validation alongside the other hyperparameters. Developing a principled aggregation scheme — e.g., via Riemannian optimization on the positive rank-1 manifold — is an important direction for future work.
>
> ---
>
> ## Q3. On TSL performance when the true model is additive
>
> We will clarify this point in the revision and add appendix simulations covering a broader range of settings to show when TSL does not give high/competitive accuracy.
> We suspect that certain additive signals may require more stages to fit well
> making it less competitive compared to models that better leverage the additive structure.
> TSL  is constrained by the stability of fitting: instability arises when a grid tensor admits multiple dominant representations, such as when the observed support is non-rectangular (See W1 response to Reviewer 1WXA), or in later stages (that are often used for additive models) where the signal-to-noise ratio is low. In these settings, independently bagged grid tensors may converge to different but similarly performant componentwise representations, making alignment more difficult and causing naive aggregation to degrade the combined backbone.
>
> ---
>
> ## Flowchart suggestion
>
> > "the author(s) may consider adding a flowchart to visualize the framework of TSL."
>
> We agree and will add an illustration of the fitting process to the revision.

---

> > ### Author Rebuttal · Reviewer_zSd4 · 2026-04-02
> >
> > Thanks for your response.

---

### Official Review · Reviewer_EsMe · 2026-03-13

**Soundness:** 3
**Presentation:** 2
**Significance:** 3
**Originality:** 3
**Overall Recommendation:** 4
**Confidence:** 3

**Summary:**

This paper proposes Tensor Separation Learning (TSL), a glass-box regression model that represents a function as a sum of separable (rank-1) tensor products. The motivation is to address limitations of additive decompositions commonly used in interpretable machine learning, such as signal cancellation in local explanations and contamination of partial dependence plots by higher-order interactions.
The model is trained using a stagewise greedy procedure with orthogonal refitting. The authors argue that the separable structure allows partial dependence plots to correspond exactly to model components, enabling faithful interpretability without post-hoc decomposition.
The experimental section includes both predictive evaluations and interpretability case studies. In particular, the California Housing dataset is used as an interpretability example, where partial dependence plots are compared across models. The results illustrate how the proposed method produces different partial dependence behavior compared with existing approaches.

**Compliance With Llm Reviewing Policy:**

Affirmed.

**Final Justification:**

**Soundness**: The rebuttal adequately addressed my technical concerns. The justification of ordered-difference as sign-ambiguity removal (W1), the computational complexity analysis with embarrassingly parallel bagged grids (W4), and the explanation of PD curve shift due to sparse boundary data (W5) were all satisfactory.

**Originality**: The combination of separable tensor decomposition with stagewise greedy learning as a glass-box model is novel and provides a concrete alternative to additive decomposition that avoids signal cancellation and PD plot contamination.

**Significance**: The paper makes a meaningful contribution to interpretable ML, though the limitation under additive data structures should be clearly stated in the revision.

**Clarity**: The backbone-tilt interpretation (W2) was initially difficult to parse, but the authors' response referencing Proposition 5.2 and the promise to add examples and visualizations in the revision are adequate.

The Broader Impact Statement requested in my acknowledgement has been provided.

I raise my score from 3 to 4 (Weak accept).

**Key Questions For Authors:**

* In the empirical example, why does the TSL partial dependence curve shift toward negative values in the corresponding figure?

**Limitations:**

The authors do not appear to explicitly discuss limitations or potential negative societal impacts of their work.

**Strengths And Weaknesses:**

## Strengths

* The property that partial dependence functions correspond exactly to the fitted model structure is a compelling aspect of the proposed approach. This connection between model structure and interpretability diagnostics is clearly articulated and represents a meaningful design goal for interpretable models.
* The motivation of the paper, highlighting limitations of additive decompositions and proposing a separable alternative, is clearly presented. The contrast between additive models and separable decompositions provides a coherent narrative for the proposed method.
* The paper addresses important issues in explainable machine learning, including signal cancellation in additive explanations and contamination of partial dependence plots.
Addressing these limitations is relevant for improving the reliability of interpretability tools.
* The proposed approach combines separable tensor decompositions with stagewise greedy learning in a glass-box model designed specifically for interpretability. This combination appears to offer a novel perspective compared with existing additive interpretability models.

## Weaknesses
* Several modeling choices would benefit from additional justification or explanation. In particular, the rationale for the ordered-difference parameterization is not fully clear. Providing more intuition or motivation for this formulation could improve understanding.
* Some sections introduce substantial technical detail without clearly explaining how these elements influence the model’s behavior. For example, the backbone and exponential-tilt parameterization is mathematically involved, and it can be difficult to develop an intuitive understanding of how these components affect the learned model.
* The empirical results suggest that predictive performance can vary depending on whether the data generating process matches the separable structure assumed by the model. In particular, the method may perform less strongly when the underlying signal is largely additive. This limitation could be discussed more explicitly.
* The paper does not provide a discussion of the computational complexity of the training algorithm. Since the method involves stagewise fitting and grid refinement procedures, a complexity analysis would help readers assess its scalability and practical applicability.

---

> ### Author Rebuttal · Authors · 2026-03-29
>
> We thank the reviewer for the positive assessment and constructive suggestions. We address each point in turn.
>
> ---
>
> ## W1. Ordered-difference parameterization
>
> The ordered-difference parameterization removes the sign ambiguity of unconstrained multiplicative factorizations in the aggregation step of Appendix Algorithm 10. At each stage, TSL averages feature-wise components across $n_{\mathrm{grid}}$ fitted grids and then forms their product. In an unconstrained rank-1 product $\prod_j f_j(x_j)$, flipping the sign of any even subset of components leaves the product unchanged, so nearly equivalent fitted grids can have opposite component signs. Averaging then becomes unstable, since similar components with opposite signs cancel. We therefore constrain components to be nonnegative and use the ordered difference to recover arbitrary signed stage functions; see Appendix Algorithm 11 for details. If the support is non-rectangular, sign-flips can even occur simultaneously within a component (See W1 response to Reviewer 1WXA). We found empirically that imposing positivity and using the ordered difference is the most stable fitting procedure. See also response D1 to Reviewer Dztf on the interpretability advantage and our response to Reviewer zSd4 on the TSL fitting process.
>
> ---
>
> ## W2. Backbone and tilt interpretation
> We appreciate this comment. The intended intuition is meant to come from the examples and visualizations, and we will make this clearer in the revision. In practice, the signed PD already conveys most of a stage's direction, so the tilt need not be read in isolation; by Proposition 5.2, the backbone-tilt representation can be recovered from stage-wise PD objects (see also our response to Reviewer 1WXA, Point W1, for why PD-based interpretability does not require exact identifiability). The backbone adds the new information, since it remains nonzero under cancellation and therefore distinguishes truly inactive regions from regions where the stage is active but the signed PD is small because positive and negative parts nearly cancel.
>
> Figure 1 illustrates this on California Housing. Since TSL is fitted greedily, stage 2 corrects what stage 1 misses, and its backbone for latitude and longitude concentrates on coastal and inland regions where stage 1 underfits. The signed PD then gives the direction of that correction: stage 2 pushes predictions upward for coastal cities and downward for inland cities. Appendix Figure 5 shows the second part of the intuition. In later stages such as stage 8, the shaded signed-PD region is small for most features, but the backbone remains nonzero. These features are therefore not inactive; they still act mainly as magnitude multipliers, while the signed direction is carried by features with larger imbalance. We will revise the text to make this active-vs-balanced distinction clearer.
>
> ---
>
> ## W3. Performance on additive data.
>  We will make this connection clearer in the paper. For the broader discussion of why additive structure can be more challenging for TSL, we refer to our response to Reviewer zSd4.
> An illustrative example  where TSL performs substantially better than the best tree-based baseline is socmob. In the original paper for that dataset, the authors use a log-linear model; if the signal is approximately linear on the log scale, then it is multiplicative on the original scale, which naturally favors TSL.
>
> ---
>
> ## W4. Computational complexity.
>
> TSL has overall training cost $O(r \cdot B \cdot T \cdot n \cdot p)$, where $r$ is the number of boosting stages, $B$ the number of bagged grids per stage, $T$ the number of split iterations per grid, $n$ the sample size, and $p$ the number of features. Each split iteration has CART-like cost $O(np)$, since prefix-sum caches of sufficient statistics allow $O(1)$ evaluation per candidate split. Relative to standard gradient boosting, the main extra multiplier is $B$, though these fits are embarrassingly parallel. Our current implementation uses exact split points; histogram binning as in LightGBM/XGBoost is directly applicable and would reduce the per-iteration cost from $O(np)$ to $O(Hp)$ with $H \ll n$. We will add this discussion in the revision.
>
> ---
>
> ## W5. Why does the TSL partial dependence curve shift toward negative values?
>
>  In the Figure 3(a), the fitted negative-side scaling $C_-$ slightly exceeds $C_+$, shifting the signed PD downward. This is visible in Figure 3(b) where the $PD_{-,1}$ curve has slightly larger magnitude than ${PD}_{+,1}$. This occurs because sparse boundary data reduces **fit** stability.
>
> ---
>
> ## Limitations and societal impact.
> We will add an explicit limitations subsection and societal-impact discussion. We view TSL's impact as positive: it provides a glassbox alternative whose internal structure is directly linked to interpretability targets such as PD functions.

---

> > ### Author Rebuttal · Reviewer_EsMe · 2026-04-02
> >
> > I thank the authors for the thorough rebuttal. The responses adequately address my technical concerns (W1–W5).
> >
> > However, the manuscript is missing a Broader Impact Statement, which is mandatory per the ICML 2026 Call for Papers: "Authors are required to include a statement of the potential broader impact of their work, including its ethical aspects and future societal consequences." I request that the authors provide the concrete text of this statement in their next response.

---

> > > ### Author Response · Authors · 2026-04-05
> > >
> > > Thank you for pointing this out. We had inadvertently omitted the impact statement. We propose to add the following text to the revision:
> > >
> > > **Limitations**
> > >
> > > TSL's separable structure is both its strength and its main limitation. When the underlying regression function is well-approximated by a sum of rank-1 products, TSL produces accurate, interpretable fits; when the true signal is predominantly additive or requires high tensor rank, TSL may need many stages to match the accuracy of methods whose inductive bias better aligns with the data. Different random seeds can yield fitted regressors with different separable representations, since separable factorizations are inherently non-unique — scaling can be redistributed among factors without changing the product, stages can be permuted, and the nonconvex fitting procedure can converge to different representations. Each fitted model is still exactly readable through its stagewise PD components (Proposition 5.1), but practitioners should be aware that the decomposition describes what the fitted model computes, not a unique latent structure of the data-generating process. The largest current drawback is sample inefficiency: because independently fitted grid tensors may converge to different representations of the same function, the similarity filtering step must discard grids that are not aligned with the reference, so only a fraction of the fitted grids contribute to the final stage. The filtering threshold is currently a tuned hyperparameter; developing an aggregation method that utilizes all fitted grids — for example, via alignment on the positive rank-1 manifold — is an important direction for future work.
> > >
> > > **Impact Statement**
> > >
> > > TSL is a methodological contribution to interpretable machine learning. By producing a glass-box regression model whose partial dependence plots are exact representations of fitted components rather than post-hoc summaries, it can help practitioners inspect interaction structure, identify which features act as multipliers versus directional drivers, and detect regions where the model is active versus balanced. That said, interpretability does not by itself guarantee correctness, fairness, or safety. Like any regression model, TSL should be paired with domain expertise, robustness checks, and fairness analyses before informing consequential decisions.

---

### Decision · Program_Chairs · 2026-04-30

**Decision:**

Accept (regular)

**Comment:**

This paper presents a new low-rank tensor decomposition for functions of multiple variables which separates positive and negative components of each rank components, and propose an algorithm for recovering the best approximation of a given function. The algorithm is based on greedy least squares following [1] but also incorporates adaptable grids for the components, which unlike other works are assumed to be piecewise constant functions. The authors show an approximation error bound of $O(1/\sqrt{r})$ for functions with derivatives in $L^1$, show decent performance on a broad range of datasets, and demonstrate the interpretability properties of the model on the California Housing dataset.



**Most reviewers are positive** about the paper, especially Reviewer 1WXA who not only has a high score but is quite vocal about their support. The reviewers praise the “good motivation”, “novel perspective” (Rev. EsMe) and the fact that  the paper is “well written and interesting”, and  “accessible even without particularly strong background” (rev. 1WXA). Both reviewers 1WXA and Dztf mention the lack of true **identifiability** as a **limitation**.

Going more into details, Reviewer EsMe complains that “the backbone and exponential tilt could be better explained”, a point where based on my own reading of the paper I must side with the reviewer and encourage the authors to improve the presentation.
zSd4 finds it surprising that the model underperforms when the ground truth is additive. Authors mention that the stability of the algorithm may be in affected in this case. Personally, I don’t find it too surprising that TSL can underperform NAM when the ground truth is additive/NAM type, so this is not an issue and I side with the authors on that.


Reviewer 1WXA also **requests for separable baselines** such as [2,3]. The authors retort that “basis-function tensor methods are often less natural for sparse data” and describe the difference between their algorithm and the baselines. Personally, I am not particularly satisfied by that answer. If the use of basis functions makes the baselines worse, why not evaluate them?

**Reviewer Dztf** argues without concrete evidence that the “*novelty is low*” compared to certain works which they argue were not “*properly cited*”. At the same time, the reviewer also attacks the fact that the authors are differentiating themselves from the said references by using piecewise constant functions instead of splines or Fourier basis functions because the latter would be more “*usual*” or proper. The reviewer also attacks the interpretability of the method based on the idea that the summation of rank 1 factors obscures the interpretability of the method, a blanket criticism which applies equally to all the references they have provided. **I have largely ignored this review in my decision**.





=================


Overall, I find the method very solid and the experimental results very convincing. I am surprised the authors have not mentioned the **breadth of the experimental evaluation** in terms of datasets and the pleasant **interpretability plots** including for the synthetic case, which I especially appreciate. The notion of partial dependency curve is also pleasantly explained. However, I am a bit bothered the fact that the authors **only** provide **approximation guarantees, without** incorporating **sample complexity guarantees**. Similarly, **only** the **PARAFAC** case is **covered**, **not Tucker**. Both are especially problematic given that the new model is relatively simple and the depth of the contribution is not especially strong for a top tier publication such as ICML (this is arguably a bit thinly sliced). The components of the algorithm should also be better explained. Lastly, as mentioned in more detail below, I am bothered by some possible imprecisions in the proof of the main approximation result. However, the contribution is still a solid first step and a **meaningful addition to the literature**. Taking all these into account, I **lean towards acceptance**.



For the revision, the actionable items are:

1.	Add some separable baselines
2.	Improve the clarity of the exposition of the exponential tilt
3.	Decompress the proof of proposition H.2. to better clarify how to incorporate the positive constraints. I am not satisfied by the rigor of the proof in its current form as Remark H.3 is too vague. Cite the appropriate theorem 2.3 from [1] when using it in the proof
4.	Improve the description of the components of the algorithm: there are many details which can only be understood by reading the algorithms themselves in the appendix, as they are not really described in text.
5.	In future work, the authors are strongly encouraged to incorporate sample complexity guarantees into the analysis.
6.	In future work, the Tucker decomposition variant should also be studied.


====================

[*More detailed AC comments*]

It seems that proposition H.2, corresponding to Proposition 6.1 in the main text, is your main result. I find it a little unintuitive that the authors are first stating Proposition H.1, which corresponds to Theorem 2.1 in [1] before the proof. Indeed, inside the proof of Proposition H.2, it seems to me that at line 1326 when the authors write “by (Barron et al. 2008), …”, they actually mean to use not Theorem 2.1 from [1] but Theorem 2.3 from [1], which indeed states that the $L^2$ error of the greedy approximation is bounded by $\frac{4 \|h\|_{\mathcal{V}_1}}{\sqrt{r}}+\|f-h\|^2$ (using the notation $\mathcal{V}_1$ as in the present submission) where $h$ is an arbitrary function. The result from [1] itself is not entirely intuitive in that sense and it is only after digging through the reference that one understands that the present submission’s proof is actually correct and that the constructed function $s_m$ is *just a certificate* and is never meant to be the actual OGA approximation. The authors are strongly encouraged to make the proof more reader friendly in that regard, at the absolute minimum by quoting and stating the precise result from [1] they are using.

Going further, the proof of Proposition 6.1 is not completely covered by Proposition H.2, due to the positivity constraints. In fact, a very significant part of the proof of the actual result 6.1 is buried away inside the relatively imprecise Remark H.3.

 **I cannot fully make sense of how to turn this into a full proof**. Indeed,  whilst I agree with the authors that the elements $g$ in the proof of Proposition H.2 are positive, but I cannot really see how the argument that “the OGA procedure does not depend on the sign of the dictionary elements” works. For the proof provided here to be correct, it must be the case that the raw result from [1] (i.e. Theorem 2.3 from there) is enough to finish the proof without dissecting its proof within the reference [1]. I can’t see how that is possible since the residuals $r_N$ in Theorem 2.3 may not be positive.

Given how professional the authors appear in the rest of the paper, I am inclined to give the benefit of the doubt and believe that the result is in fact correct, but it doesn’t alter the fact that the proof is not reader friendly.

Another comment, the discussion from lines 1343 to line 1364 on page 25 doesn’t seem to pertain to any results actually present in the main paper, which makes the appendix appear a little unfinished.



===============


**References**

[1] Andrew R. Barron, Albert Cohen, Wolfgang Dahmen, Ronald A. DeVore. Approximation and learning by greedy algorithms. The annals of statistics, 2008.

[2] Beylkin, G. and Mohlenkamp, M. J. Algorithms for numerical analysis in high dimensions. SIAM J. Opt. 2005.

[3] Beylkin, G., Garcke, J., and Mohlenkamp, M. J. Multi- variate regression and machine learning with sums of separable functions. SIAM J. Opt. 2009